# BRANCHED SCHRÖDINGER BRIDGE MATCHING

**Sophia Tang**[1], **Yinuo Zhang**[2], **Alexander Tong**[3], **Pranam Chatterjee**[1,4,†]

[1]Department of Computer and Information Science, University of Pennsylvania
[2]Center of Computational Biology, Duke-NUS Medical School, [3]AITHYRA
[4]Department of Bioengineering, University of Pennsylvania

[†]Corresponding author: pranam@seas.upenn.edu

## ABSTRACT

Predicting the intermediate trajectories between an initial and target distribution is a central problem in generative modeling. Existing approaches, such as flow matching and Schrödinger bridge matching, effectively learn mappings between two distributions by modeling a single stochastic path. However, these methods are inherently limited to unimodal transitions and cannot capture *branched* or *divergent* evolution from a common origin to multiple distinct modes. To address this, we introduce **Branched Schrödinger Bridge Matching (BranchSBM)**, a novel framework that learns branched Schrödinger bridges. BranchSBM parameterizes multiple time-dependent velocity fields and growth processes, enabling the representation of population-level divergence into multiple terminal distributions. We show that BranchSBM is not only more expressive but also essential for tasks involving multi-path surface navigation, modeling cell fate bifurcations from homogeneous progenitor states, and simulating diverging cellular responses to perturbations.

## 1 INTRODUCTION

Tasks like crowd navigation and modeling cell-state transitions under perturbation involve learning a transport map between two empirically observed endpoint distributions, rather than a noisy prior required for denoising diffusion (Austin et al., 2021) and flow matching (Lipman et al., 2023). The Schrödinger Bridge (SB) (Schrödinger, 1931) problem seeks to identify an optimal stochastic map between a pair of endpoint distributions that minimizes the Kullback-Leibler (KL) divergence to an underlying reference process. Schrödinger Bridge Matching (SBM) solves the SB problem by parameterizing a drift field that matches a mixture of conditional stochastic bridges between endpoint pairs that each minimize the KL divergence from a known reference process. Typically, SBM assumes conservation of mass from the initial to the target distribution, which fails to capture dynamical population behaviors such as growth and destruction of mass, commonly seen in single-cell population data. Furthermore, prior methods focus on transporting samples from an initial distribution to the target distributions via *independent stochastic trajectories*, without accounting for *branching* dynamics (Liu et al., 2023a; Tong et al., 2024b; Theodoropoulos et al., 2024; Liu et al., 2022; De Bortoli et al., 2021), which are susceptible to mode collapse when the system follows a branched trajectory that diverges toward multiple distinct target modes.

The notion of branching is central to many real-world systems. For example, when a homogeneous cell population undergoes a perturbation such as gene knockouts or drug treatments, it frequently induces fate bifurcation as the cell population splits into multiple phenotypically distinct outcomes or commits to divergent cell fates (Shalem et al., 2014; Zhang et al., 2025a). These trajectories are observable in single-cell RNA sequencing (scRNA-seq) data, where each subpopulation independently evolves and undergoes growth or contraction along its trajectory toward a distinct terminal state (Dixit et al., 2016). In this work, we introduce **Branched Schrödinger Bridge Matching (BranchSBM)**, a framework for solving the branched Schrödinger bridge problem by parameterizing diverging velocity fields and branch-specific growth rates, which together define a set of conditional stochastic bridges from the common source to multiple terminal modes.

Our **main contributions** can be summarized in the following points:

1. We define the Branched Generalized Schrödinger Bridge problem and introduce **BranchSBM**, a novel matching framework that learns optimal branched trajectories from an initial distribution $\pi_0$ to multiple target distributions $\{\pi_{t,k}\}$.

2. We derive the Branched Conditional Stochastic Optimal Control (CondSOC) problem as the sum of Unbalanced CondSOC objectives and leverage a multi-stage training algorithm to learn the optimal branching drift and growth fields that transport mass along a branched trajectory.

3. We demonstrate the unique capability of BranchSBM to model dynamic branching trajectories while matching multiple target distributions across various problems, including 3D navigation over LiDAR manifolds (Section 5.1), modeling differentiating single-cell population dynamics (Section 5.2), and predicting heterogeneous cell states after perturbation (Section 5.3).

## 2 PRELIMINARIES

**Schrödinger Bridge Problem** Given a reference probability path measure $\mathbb{Q}$, the Schrödinger Bridge (SB) problem aims to find an optimal path measure $\mathbb{P}^{\text{SB}}$ that minimizes the Kullback-Leibler (KL) divergence with $\mathbb{Q}$ while satisfying the boundary distributions $\mathbb{P}_0 = \pi_0$ and $\mathbb{P}_1 = \pi_1$.

$$\mathbb{P}^{\text{SB}} = \min_{\mathbb{P}}\{\text{KL}(\mathbb{P}\|\mathbb{Q}) : \mathbb{P}_0 = \pi_0, \mathbb{P}_1 = \pi_1\} \tag{1}$$

where $\mathbb{Q}$ is commonly defined as standard Brownian motion. For an extended background and formal definition of Schrödinger Bridges, refer to Definition A.3 and Appendix A.1.

**Generalized Schrödinger Bridge Problem** The solution to the standard SB problem minimizes the kinetic energy of the conditional drift term $u_t(X_t)$ that preserves the endpoints drawn from the coupling $(\boldsymbol{x}_0, \boldsymbol{x}_1) \sim \pi_{0,1}$ defined as

$$\min_{u_t} \int_0^1 \mathbb{E}_{p_t}\|u_t(X_t)\|_2^2 dt \quad \text{s.t.} \quad \begin{cases} dX_t = u_t(X_t)dt + \sigma dB_t \\ X_0 \sim \pi_0, \quad X_1 \sim \pi_1 \end{cases} \tag{2}$$

where $dB_t$ is standard $d$-dimensional Brownian motion. To define more complex systems where the *optimal* dynamics cannot be accurately captured by minimizing the standard squared-Euclidean cost in entropic OT (Vargas et al., 2021), the Generalized Schrödinger Bridge (GSB) problem introduces an additional non-linear state-cost $V_t(X_t)$ (Chen et al., 2021b; Chen & Georgiou, 2016; Liu et al., 2022). The minimization objective becomes

$$\min_{u_t} \int_0^1 \mathbb{E}_{p_t} \left[\frac{1}{2}\|u_t(X_t)\|_2^2 + V_t(X_t)\right] dt \quad \text{s.t.} \quad \begin{cases} dX_t = u_t(X_t)dt + \sigma dB_t \\ X_0 \sim \pi_0, \quad X_1 \sim \pi_1 \end{cases} \tag{3}$$

where the state cost can also be interpreted as the *potential energy* of the system at state $X_t$.

## 3 BRANCHED SCHRÖDINGER BRIDGE MATCHING

A **key challenge** in trajectory matching is reconstructing multi-modal marginals, particularly when modes diverge along distinct dynamical paths. Existing Schrödinger bridge and flow matching frameworks approximate multi-modal distributions by simulating many independent particle trajectories, which are susceptible to mode collapse, with particles concentrating on dominant high-density modes or traversing only low-energy intermediate paths. To address this challenge, we introduce **Branched Schrödinger Bridge Matching (BranchSBM)**, a framework that learns a set of diverging velocity fields to reconstruct multi-modal target distributions while simultaneously learning growth networks that allocate mass across branches. Guided by a time-dependent potential $V_t$, BranchSBM captures diverging, energy-minimizing dynamics without requiring intermediate-time supervision and can generate the full branched evolution from a single initial sample.

### 3.1 UNBALANCED CONDITIONAL STOCHASTIC OPTIMAL CONTROL

**Unbalanced Generalized Schrödinger Bridge Problem** Extending the definition of the Generalized Schrödinger Bridge (GSB) problem in Equation 3, we define the Unbalanced GSB problem by

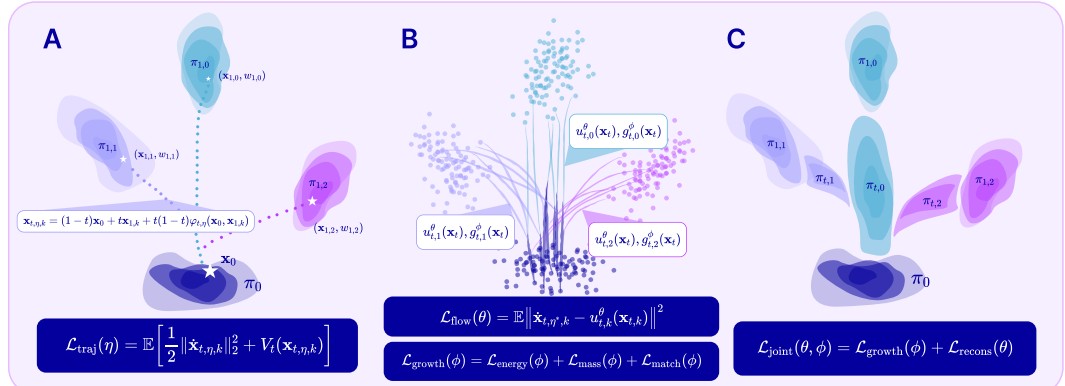

Figure 1: **Branched Schrödinger Bridge Matching (A)** Stage 1 trains a correction term that learns the optimal interpolant conditioned on endpoints **(B)** Stage 2 and 3 trains a separate flow and growth network for each branch independently **(C)** Stage 4 jointly optimizes the flow and growth networks to minimize the energy, mass, and matching loss.

scaling the minimization objective by a time-dependent weight $w_t = w_0 + \int_0^t g_s(X_s)ds$ that evolves according to a time-varying growth rate $g_t(X_t) : \mathbb{R}^d \times [0,1] \to \mathbb{R}$.

$$\min_{u_t, g_t} \int_0^1 \mathbb{E}_{p_t} \left[ \frac{1}{2} \|u_t(X_t)\|_2^2 + V_t(X_t) \right] w_t(X_t)dt \quad \text{s.t.} \quad \begin{cases} dX_t = u_t(X_t)dt + \sigma dB_t \\ X_0 \sim \pi_0, \quad X_1 \sim \pi_1 \\ w_0 = w_0^\star, \quad w_1 = w_1^\star \end{cases} \tag{4}$$

**Unbalanced Conditional Stochastic Optimal Control (CondSOC)**    Now, we show that we can solve the Unbalanced GSB problem as an Unbalanced CondSOC problem where the optimal drift $u_t$ and growth $g_t$ minimize the expectation of the objective in (4) conditioned on pairs of endpoints.

**Proposition 3.1** (Unbalanced Conditional Stochastic Optimal Control). *Suppose the marginal density can be decomposed as $p_t(X_t) = \int p_t(X_t|\boldsymbol{x}_0, \boldsymbol{x}_1)\pi_{0,1}(d\boldsymbol{x}_0, d\boldsymbol{x}_1)$, where $\pi_{0,1}$ is a fixed joint coupling of the data. Then, we can identify the optimal drift $u_t^\star$ and growth $g_t^\star$ that solves the Unbalanced GSB problem in (4) by minimizing the **Unbalanced Conditional Stochastic Optimal Control objective** given by*

$$\min_{u_t, g_t} \mathbb{E}_{(\boldsymbol{x}_0, \boldsymbol{x}_1) \sim \pi_{0,1}} \left[ \int_0^1 \mathbb{E}_{p_{t|0,1}} \left[ \left( \frac{1}{2} \|u_t(X_t|\boldsymbol{x}_0, \boldsymbol{x}_1)\|_2^2 + V_t(X_t) \right) w_t(X_t) \right] dt \right] \tag{5}$$

$$\text{s.t.} \quad dX_t = u_t(X_t|\boldsymbol{x}_0, \boldsymbol{x}_1)dt + \sigma dB_t, \quad X_0 = \boldsymbol{x}_0, \quad X_1 = \boldsymbol{x}_1 \quad w_0 = w_0^\star, \quad w_1 = w_1^\star \tag{6}$$

*where $w_t = w_0 + \int_0^t g_s(X_s)ds$ is the time-dependent weight initialized at $w_0^\star$, $u_t$ is the drift, $g_t$ is the growth rate, and $\pi_{0,1}$ is the weighted coupling of paired endpoints $(\boldsymbol{x}_0, w_0^\star, \boldsymbol{x}_1, w_1^\star) \sim \pi_{0,1}$.*

The proof is provided in Appendix C.1. This defines the objective for us to tractably solve the Unbalanced GSB problem by conditioning on a finite set of endpoint pairs in the dataset.

### 3.2    BRANCHSBM: SUM OF UNBALANCED CONDSOC PROBLEMS

**Branched Generalized Schrödinger Bridge Problem**    Given the Unbalanced GSB problem, we define the Branched GSB problem as minimizing the sum of Unbalanced GSB problems across all branches. All mass begins along a primary path indexed $k = 0$ with initial weight 1. Over $t \in [0,1]$, mass is transferred across $K$ secondary branches with initial weight 0 and target weight $w_{1,k}^\star$ such that it minimizes the objective defined as

$$\min_{\{u_{t,k}, g_{t,k}\}_{k=0}^K} \int_0^1 \left\{ \mathbb{E}_{p_{t,0}} \left[ \frac{1}{2} \|u_{t,0}(X_{t,0})\|_2^2 + V_t(X_{t,0}) \right] w_{t,0} + \sum_{k=1}^K \mathbb{E}_{p_{t,k}} \left[ \frac{1}{2} \|u_{t,k}(X_{t,k})\|_2^2 + V_t(X_{t,k}) \right] w_{t,k} \right\} dt$$

$$\text{s.t.} \quad dX_{t,k} = u_{t,k}(X_{t,k})dt + \sigma dB_t, \quad X_0 \sim \pi_0, \quad X_{1,k} \sim \pi_{1,k}, \quad w_{0,k} = \delta_{k=0}, \quad w_{1,k} = w_{1,k}^\star \tag{7}$$

When total mass across branches is conserved, we enforce $\sum_{k=0}^{K} w_{t,k} = 1$ for all $t \in [0, 1]$, which constrains the growth rates such that $g_{t,0}(X_{t,0}) + \sum_{k=1}^{K} g_{t,k}(X_{t,k}) = 0$. This ensures that mass lost from the primary branch (when $g_{t,0} < 0$) is redistributed among the secondary branches (where $g_{t,k} > 0$ for some $k \in \{1, \ldots, K\}$). The primary branch evolves from an initial weight of 1 according to $w_{t,0} = 1 + \int_0^t g_s(X_{s,0}) ds$ and the $K$ secondary branches grow from the primary branch from weight 0 according to $w_{t,k} = \int_0^t g_s(X_{s,k}) ds$.

**Branched Conditional Stochastic Optimal Control**  Following a similar procedure as shown for the Unbalanced GSB problem, we can reformulate the Branched GSB problem as solving the Branched CondSOC problem where we optimize a set of parameterized drift $\{u_{t,k}\}_{k=0}^{K}$ and growth $\{g_{t,k}\}_{k=0}^{K}$ networks by minimizing the energy of the conditional trajectories between paired samples $(\boldsymbol{x}_0, \{\boldsymbol{x}_{1,k}\}_{k=0}^{K}) \sim \{p_{0,1,k}\}_{k=0}^{K}$.

**Proposition 3.2** (Branched Conditional Stochastic Optimal Control). *For each branch, let* $p_{t,k}(X_{t,k}) = \int p_{t,k}(X_{t,k}|\boldsymbol{x}_0, \boldsymbol{x}_{1,k})\pi_{0,1,k}(d\boldsymbol{x}_0, d\boldsymbol{x}_{1,k})$, *where* $\pi_{0,1,k}$ *is the joint coupling distribution of samples* $\boldsymbol{x}_0 \sim \pi_0$ *from the initial distribution and* $\boldsymbol{x}_{1,k} \sim \pi_{1,k}$ *from the kth target distribution. Then, we can identify the set of optimal drift and growth functions* $\{u_{t,k}^\star, g_{t,k}^\star\}_{k=0}^{K}$ *that solve the Branched GSB problem in* (7) *by minimizing the **sum of Unbalanced CondSOC objectives** given by*

$$\min_{\{u_{t,k}, g_{t,k}\}_{k=0}^{K}} \mathbb{E}_{(\boldsymbol{x}_0, \boldsymbol{x}_{1,0}) \sim \pi_{0,1,0}} \int_0^1 \left\{ \mathbb{E}_{p_{t|0,1,0}} \left[ \frac{1}{2} \|u_{t,0}(X_{t,0})\|_2^2 + V_t(X_{t,0}) \right] w_{t,0} \right\} dt$$

$$+ \sum_{k=1}^{K} \mathbb{E}_{(\boldsymbol{x}_0, \boldsymbol{x}_{1,k}) \sim \pi_{0,1,k}} \int_0^1 \left\{ \mathbb{E}_{p_{t|0,1,k}} \left[ \frac{1}{2} \|u_{t,k}(X_{t,k})\|_2^2 + V_t(X_{t,k}) \right] w_{t,k} \right\} dt \quad (8)$$

$$s.t. \quad dX_{t,k} = u_{t,k}(X_{t,k})dt + \sigma dB_t, \ X_0 = \boldsymbol{x}_0, \ X_{1,k} = \boldsymbol{x}_{1,k}, \ w_{0,k} = \delta_{k=0}, \ w_{1,k} = w_{1,k}^\star \quad (9)$$

*where* $w_{t,0} = 1 + \int_0^t g_{s,0}(X_{s,0}) ds$ *is the weight of the primary paths initialized at 1 and* $w_{t,k} = \int_0^t g_{s,k}(X_{s,k}) ds$ *are the weights of the $K$ secondary branches initialized at 0.*

The proof is given in Appendix C.2. This defines the objective for us to tractably solve the Branched GSB problem in Section 4 by conditioning on a discrete set of branched endpoint pairs in the dataset.

**Remark 3.1.** *When $g_{t,0} \equiv 0$ and $g_{t,k} \equiv 0$ for all $(\boldsymbol{x}, t) \in \mathbb{R}^d \times [0, 1]$ and $k \in \{1, \ldots, K\}$, then the Branched CondSOC problem is the solution to the single path GSB problem.*

## 4 LEARNING BRANCHSBM USING NEURAL NETWORKS

Given an initial data distribution $\pi_0$ and $K + 1$ target distributions $\{\pi_{1,k}\}_{k=0}^{K}$, BranchSBM aims to learn the optimal neural drift and growth fields that approximate the solution of the Branched CondSOC problem in Proposition 3.2. Our framework leverages a multi-stage training algorithm that enables stable and scalable learning of branched dynamics from data snapshots.

### 4.1 BRANCHED NEURAL INTERPOLANT OPTIMIZATION

Since the optimal trajectory under the state cost $V_t(X_t) : \mathbb{R}^d \times [0, 1] \to \mathbb{R}$ follows a non-linear cost manifold, given a pair of endpoints $(\boldsymbol{x}_0, \boldsymbol{x}_{1,k}) \sim \pi_{0,1,k}$, we train a neural path interpolant $\varphi_{t,\eta}(\boldsymbol{x}_0, \boldsymbol{x}_{1,k}) : \mathbb{R}^d \times \mathbb{R}^d \times [0, 1] \to \mathbb{R}^d$ that defines the intermediate state $\boldsymbol{x}_{t,\eta,k}$ and velocity $\dot{\boldsymbol{x}}_{t,\eta,k} = \partial_t \boldsymbol{x}_{t,\eta,k}$ at time $t$, which minimizes (3.2). We note that the neural interpolant has been introduced previously for single-path interpolants in Kapuśniak et al. (2024); Neklyudov et al. (2023). We define $\boldsymbol{x}_{t,\eta,k}$ to be bounded at the endpoints as given by

$$\boldsymbol{x}_{t,\eta,k} = (1 - t)\boldsymbol{x}_0 + t\boldsymbol{x}_{1,k} + t(1 - t)\varphi_{t,\eta}(\boldsymbol{x}_0, \boldsymbol{x}_{1,k}) \quad (10)$$

$$\dot{\boldsymbol{x}}_{t,\eta,k} = \boldsymbol{x}_{1,k} - \boldsymbol{x}_0 + t(1 - t)\dot{\varphi}_{t,\eta}(\boldsymbol{x}_0, \boldsymbol{x}_{1,k}) + (1 - 2t)\varphi_{t,\eta}(\boldsymbol{x}_0, \boldsymbol{x}_{1,k}) \quad (11)$$

To optimize $\varphi_{t,\eta}(\boldsymbol{x}_0, \boldsymbol{x}_{1,k})$ such that it predicts the energy-minimizing trajectory, we minimize the trajectory loss $\mathcal{L}_{\text{traj}}$ defined as

$$\mathcal{L}_{\text{traj}}(\eta) = \sum_{k=0}^{K} \int_0^1 \mathbb{E}_{(\boldsymbol{x}_0, \boldsymbol{x}_{1,k}) \sim \pi_{0,1,k}} \left[ \frac{1}{2} \|\dot{\boldsymbol{x}}_{t,\eta,k}\|_2^2 + V_t(\boldsymbol{x}_{t,\eta,k}) \right] dt \qquad (12)$$

After convergence, Stage 1 returns the network $\varphi_{t,\eta}^\star(\boldsymbol{x}_0, \boldsymbol{x}_{1,k})$ that generates the optimal conditional velocity $\dot{\boldsymbol{x}}_{t,\eta,k}^\star$ which defines the matching objective in Stage 2. In Stage 2, we parameterize a set of neural drift fields $u_{t,k}^\theta(\boldsymbol{x}_{t,k}) : \mathbb{R}^d \times [0,1] \to \mathbb{R}^d$ that *generates* the *mixture of bridges* defined in Stage 1 by minimizing the conditional flow matching loss (Lipman et al., 2023; Tong et al., 2024a).

$$\mathcal{L}_{\text{flow}}(\theta) = \sum_{k=0}^{K} \int_0^1 \mathbb{E}_{(\boldsymbol{x}_0, \boldsymbol{x}_{1,k}) \sim \pi_{0,1,k}} \left\| \dot{\boldsymbol{x}}_{t,\eta,k}^\star - u_{t,k}^\theta(\boldsymbol{x}_{t,k}) \right\|_2^2 dt \qquad (13)$$

---

**Proposition 4.1** (Solving the GSB Problem with Stage 1 and 2 Training). *Stage 1 and Stage 2 training yield the optimal drift $u_t^\star(X_t)$ that generates the optimal marginal probability distribution $p_t^\star(X_t)$ that solves the GSB problem in (3).*

---

The proof is provided in Appendix C.3. Since the drift for each branch $u_{t,k}^\theta(X_{t,k})$ are trained independently in Stage 2, we can extend this result across all $K + 1$ branches and conclude that the sequential Stage 1 and Stage 2 training procedures yields the optimal set of drifts $\{u_{t,k}^\star\}_{k=0}^K$ that *generate* the optimal probability paths $\{p_{t,k}^\star\}_{k=0}^K$ that solves the GSB problem for each branch.

## 4.2 Learning the Energy-Minimizing Branching Dynamics

**Branched Energy Loss** To solve the Branched CondSOC problem defined in Proposition 3.2, we minimize a branched energy loss $\mathcal{L}_{\text{energy}}$ defined as

$$\mathcal{L}_{\text{energy}}(\theta, \phi) = \int_0^1 \mathbb{E}_{\{p_{t,k}\}_{k=0}^K} \left\{ \underbrace{\left[ \frac{1}{2} \|u_{t,0}^\theta(X_{t,0})\|_2^2 + V_t(X_{t,0}) \right] w_{t,0}^\phi}_{\text{primary trajectory}} + \underbrace{\sum_{k=1}^{K} \left[ \frac{1}{2} \|u_{t,k}^\theta(X_{t,k})\|_2^2 + V_t(X_{t,k}) \right] w_{t,k}^\phi}_{K \text{ branches}} \right\} dt$$

$$\text{s.t. } w_{t,0}^\phi = 1 + \int_0^t g_{s,0}^\phi(X_{s,0}) ds, \quad w_{t,k}^\phi = \int_0^t g_{s,k}^\phi(X_{s,k}) ds \qquad (14)$$

At time $t = 0$, the primary path has weight 1, and the $K$ branches have weights 0. Over $t \in [0,1]$, the weight of the primary path changes according to $g_{t,0}^\phi(X_{t,0})$ and supplies mass to the $K$ branches, which grow at rates $g_{t,k}^\phi(X_{t,k}) \geq 0$ (Lemma C.2). Intuitively, the branched energy loss optimizes the branching growth rates such that they are non-zero when branching is favored over the primary path.

**Weight Matching Loss** We define a weight matching loss $\mathcal{L}_{\text{match}}$ that aims to minimize the difference between the predicted weights of each branch at $t = 1$, obtained by integrating the growth function $g_{t,k}^\phi(X_{t,k})$ over $t \in [0,1]$, and the true weights of each terminal distribution $\{w_{1,k}^\star\}_{k=0}^K$.

$$\mathcal{L}_{\text{match}}(\phi) = \sum_{k=0}^{K} \mathbb{E}_{p_{1,k}} \left( w_{1,k}^\phi - w_{1,k}^\star \right)^2, \quad \text{s.t. } w_{1,k}^\phi = w_{0,k} + \int_0^1 g_{t,k}^\phi(X_{t,k}) dt \qquad (15)$$

where $w_{1,k}^\star = N_k / N_{\text{total}}$ is the fraction of the population in the $k$th target distribution.

**Mass Conservation Loss** To ensure that the growth rate satisfies conservation of total mass at all times $t \in [0,1]$, we define a mass loss $\mathcal{L}_{\text{mass}}$ that enforces the sum of the weights of all $K + 1$ branches matches the true total weight at time $t$ denoted as $w_t^{\text{total}}$.

$$\mathcal{L}_{\text{mass}}(\phi) = \int_0^1 \mathbb{E}_{\{p_{t,k}\}_{k=0}^K} \left[ \left( \sum_{k=0}^{K} w_{t,k}^\phi - w_t^{\text{total}} \right)^2 + \sum_{k=0}^{K} \max\left(0, -w_{t,k}^\phi\right) \right] dt \qquad (16)$$

where $\max(0, -w_{t,k}^\phi)$ assigns an additional linear penalty for negative weight predictions. For the balanced branched SBM problem where the total mass is conserved, we have $w_t^{\text{total}} = 1$.

**Training the Growth Networks** In Stage 3, we train the growth networks $\{g_{t,k}^{\phi}\}_{k=0}^{K}$ by fixing the weights of the flow networks $\{u_{t,k}^{\theta}\}_{k=0}^{K}$ and minimizing the weighted combined loss $\mathcal{L}_{\text{growth}}$ with an additional growth penalty term $\|g_{t,k}^{\phi}\|_{2}^{2}$ to ensure coercivity of $\mathcal{L}_{\text{growth}}$

$$\mathcal{L}_{\text{growth}}(\phi) = \lambda_{\text{energy}}\mathcal{L}_{\text{energy}}(\theta, \phi) + \lambda_{\text{match}}\mathcal{L}_{\text{match}}(\phi) + \lambda_{\text{mass}}\mathcal{L}_{\text{mass}}(\phi) + \lambda_{\text{growth}}\sum_{k=0}^{K}\|g_{t,k}^{\phi}\|_{2}^{2} \qquad (17)$$

We show in Lemma C.2 that the optimal growth rates across the $K$ secondary branches are non-decreasing; however, mass destruction can still be modeled by defining an additional branch with target weight equal to the ratio of mass lost over $t \in [0, 1]$. To ensure that the set of optimal growth functions $g^{\star} := (g_{t,0}^{\star}, \ldots, g_{t,K}^{\star})$ exists, we establish Proposition 4.2 (see proof in Appendix C.4).

**Proposition 4.2** (Existence of Optimal Growth Functions). *Assume the state space $\mathcal{X} \subseteq \mathbb{R}^{d}$ is a bounded domain within $\mathbb{R}^{d}$. Let the optimal probability density of branch $k$ be a known non-negative function bounded in $[0, 1]$, denoted as $p_{t,k}^{\star} : \mathcal{X} \times [0, 1] \to [0, 1] \in L^{\infty}(\mathcal{X} \times [0, 1])$. By Lemma C.2, we can define the set of feasible growth functions in the set of square-integrable functions $L^{2}$ as*

$$\mathcal{G} := \{g = (g_{t,0}, \ldots, g_{t,K}) \in L^{2}(\mathcal{X} \times [0, 1]; \mathbb{R}^{K+1}) \mid \forall k \geq 1, \ g_{t,k}(\boldsymbol{x}) \geq 0, \sum_{k=0}^{K} g_{t,k} = 0\} \quad (18)$$

*Let the growth loss be the functional $\mathcal{L}(g) : L^{2}(\mathcal{X} \times [0, 1]; \mathbb{R}^{K+1}) \to \mathbb{R}$. Then, there exists an optimal function $g^{\star} = (g_{t,0}^{\star}, \ldots, g_{t,K}^{\star}) \in L^{2}(\mathcal{X} \times [0, 1]; \mathbb{R}^{K+1})$ where $g^{\star} \in \mathcal{G}$ such that $\mathcal{L}(g^{\star}) = \inf_{g \in \mathcal{G}} \mathcal{L}(g)$ which can be obtained by minimizing $\mathcal{L}(g)$ over $\mathcal{G}$.*

**Final Joint Training** In the final Stage 4, we train the weights for both the flow and growth networks $\{u_{t,k}^{\theta}, g_{t,k}^{\phi}\}_{k=0}^{K}$ by minimizing $\mathcal{L}_{\text{growth}}$ from Stage 3 in addition to a reconstruction loss $\mathcal{L}_{\text{recons}}$ that ensures the endpoint distribution at time $t = 1$ is maintained.

$$\mathcal{L}_{\text{recons}}(\theta) = \sum_{k=0}^{K}\mathbb{E}_{p_{1,k}}\left[\sum_{\boldsymbol{x}_{1,k} \in \mathcal{N}_{n}(\tilde{\boldsymbol{x}}_{1,k})} \max\left(0, \|\tilde{\boldsymbol{x}}_{1,k} - \boldsymbol{x}_{1,k}\|_{2} - \epsilon\right)\right] \qquad (19)$$

where $\mathcal{N}_{n}(\tilde{\boldsymbol{x}}_{1,k})$ is the set of $n$-nearest neighbors to the reconstructed state $\tilde{\boldsymbol{x}}_{1,k} \sim p_{1,k}$ at time $t = 1$ from the data points $\boldsymbol{x}_{1,k} \sim \pi_{1,k}$ at time $t = 1$.

Our multi-stage training scheme decomposes the Branched CondSOC problem into two parts. We first independently learn an optimal drift field for each branch, which is a vector field over the state space that propagates mass flow in the direction of each target distribution. Then, we fix the drift fields and learn the growth dynamics that determine the optimal distribution of mass over the branches.

## 5 EXPERIMENTS

We evaluate BranchSBM on a variety of branched matching tasks with different state costs $V_{t}(X_{t})$, including multi-path LiDAR navigation (Section 5.1), modeling differentiating single-cell population dynamics (Section 5.2), and predicting heterogeneous cell-states after perturbation (Section 5.3).

### 5.1 BRANCHED LIDAR SURFACE NAVIGATION

First, we evaluate BranchSBM for navigating branched paths along the surface of a 3-dimensional LiDAR manifold, from an initial distribution to two distinct target distributions (Figure 3).

**Setup** We define a single initial Gaussian mixture $\pi_{0}$ and two target Gaussian mixtures $\pi_{1,0}, \pi_{1,1}$ on either side of the mountain (Figure 3). We sample 5000 points i.i.d. from each of the Gaussian mixtures and assign all endpoints a target weight of $w_{1,0}^{\star} = w_{1,1}^{\star} = 0.5$. To ensure trajectories follow the LiDAR manifold, we define the state cost $V_{t}^{\text{LAND}}(X_{t})$ as the data-dependent LAND metric (Kapuśniak et al., 2024; Arvanitidis et al., 2016), which assigns lower costs in regions near coordinates in the LiDAR dataset. Further experimental details are provided in Appendix E.3.

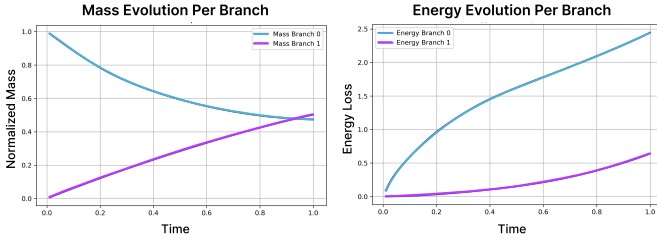

Figure 2: Plot of weight (left) and energy (right) calculated with (14) of each branch over time $t \in [0, 1]$. Mass is transferred from the primary branch to branch 1, and both converge to the target weight of 0.5 at $t = 1$. Both plots represent the average over trajectories from samples in the validation set.

Table 1: **Benchmark of BranchSBM against single-branch SBM on multi-path surface navigation.** Wasserstein distances ($\mathcal{W}_1$ and $\mathcal{W}_2$) between the reconstructed and ground-truth distributions with $N_{\text{steps}} = 100$ Euler steps at time $t = 1$ from validation samples in the initial distribution. Results are averaged over 5 independent runs.

| Model | $\mathcal{W}_1$ ($\downarrow$) | $\mathcal{W}_2$ ($\downarrow$) |
|---|---|---|
| Single Branch SBM | $0.975_{\pm 0.009}$ | $1.285_{\pm 0.007}$ |
| **BranchSBM** | $\mathbf{0.239}_{\pm 0.001}$ | $\mathbf{0.309}_{\pm 0.003}$ |

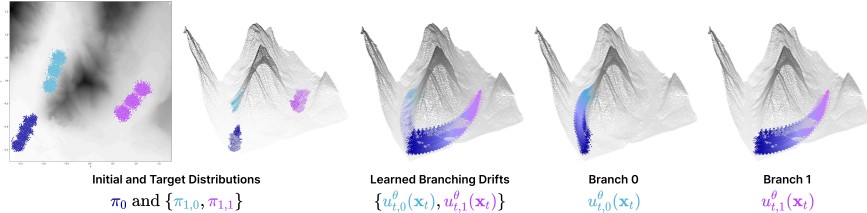

| Initial and Target Distributions | Learned Branching Drifts | Branch 0 | Branch 1 |
|---|---|---|---|
| $\pi_0$ and $\{\pi_{1,0}, \pi_{1,1}\}$ | $\{u_{t,0}^{\theta}(\mathbf{x}_t), u_{t,1}^{\theta}(\mathbf{x}_t)\}$ | $u_{t,0}^{\theta}(\mathbf{x}_t)$ | $u_{t,1}^{\theta}(\mathbf{x}_t)$ |

Figure 3: **Application of BranchSBM on Learning Branched Paths on a LiDAR Manifold.** Plots of the initial and target distributions, learned interpolants, and learned branched trajectories on the LiDAR manifold.

**Results** We show that BranchSBM can learn distinct, non-linear branched paths that curve along the 3-dimensional LiDAR manifold while minimizing the kinetic energy and state-cost. From the mass and energy curves in Figure 2, we see that mass begins in the primary branch (branch 0) and is gradually transferred to the secondary branch (branch 1) over $t \in [0, 1]$, with both curves converging to the target weight of 0.5 at $t = 1$. As mass is transferred, the slope of the cumulative energy curve decreases in branch 0 and increases in branch 1, reflecting the true energy dynamics. In Figure 3, we observe that the branching occurs at the edge of the inclined mountain, indicating that the model can determine the optimal branching time based on the paths of lowest potential energy. As shown in Table 1, BranchSBM reconstructs the endpoint distributions with significantly higher accuracy in comparison to single-branch SBM. In total, we demonstrate the capability of BranchSBM to learn branched trajectories on complex 3D manifolds.

## 5.2 Differentiating Single-Cell Population Dynamics

BranchSBM is uniquely positioned to model single-cell population dynamics where a homogeneous cell population (e.g., progenitor cells) differentiates into several distinct subpopulation branches, each of which independently undergoes growth dynamics. Here, we demonstrate this capability on mouse hematopoiesis data (Sha et al., 2023; Weinreb et al., 2020) and pancreatic $\beta$-cell differentiation data (Veres et al., 2019).

**Mouse-Hematopoiesis Results** We use a dataset consisting of mouse hematopoiesis scRNA-seq data analyzed by a lineage tracing technique from (Sha et al., 2023; Weinreb et al., 2020). This data contains three time points $t_i$ for $i \in \{0, 1, 2\}$ that are projected to two-dimensional representations $\boldsymbol{x} \in \mathbb{R}^2$ referred to as force-directed layouts or SPRING plots. From the plotted data, we can observe two clear branches that indicate the differentiation of progenitor cells into two distinct cell fates (Figure 4). We use $k$-means clustering to define two distinct target distributions $\pi_{1,0}$ and $\pi_{1,1}$ of samples at time $t_2$ and set their target weights equal to $w_{1,0} = w_{1,1} = 0.5$ due to the equal ratio of cells (Figure 10). We used samples across all time steps $t_i$ for $i \in \{0, 1, 2\}$ to define the data manifold via the LAND metric $V_{t,\eta}^{\text{LAND}}$. BranchSBM was trained on pairs sampled only from $t_0$ and $t_2$, and samples from $t_1$ were held out for evaluation. For comparison, we trained a single-branch SBM model with both clusters at $t_2$ as the target distribution.

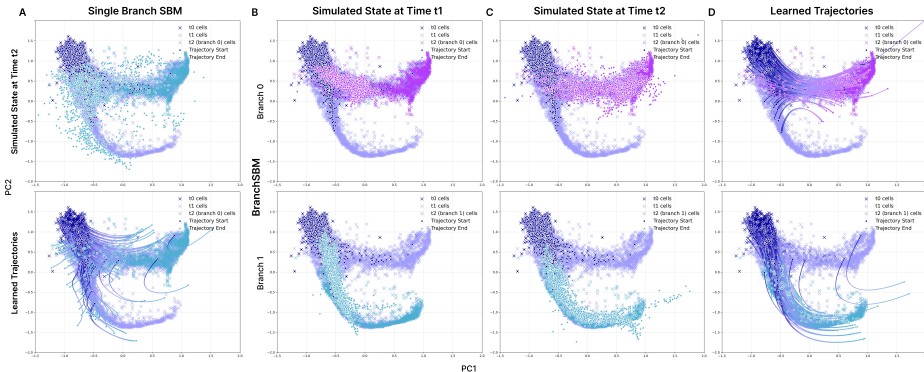

Figure 4: **Application of BranchSBM on Modeling Differentiating Single-Cell Population Dynamics.** Mouse hematopoiesis scRNA-seq data is provided for three time points $t_0, t_1, t_2$. **(A)** Simulated states (top) and trajectories (bottom) at time $t_1$ using single-branch SBM. **(B)** Simulated states with BranchSBM at $t_1$ ($t = 0.5$) and **(C)** $t_2$ ($t = 1$). **(D)** Learned trajectories over the interval $t \in [t_0, t_2]$ on validation samples.

Table 2: **Results for pancreatic $\beta$-cell differentiation experiment.** 1-Wasserstein distance ($\mathcal{W}_1$) of intermediate distributions generated by DeepRUOT (Zhang et al., 2025c), CytoBridge (Zhang et al., 2025d), and BranchSBM (Ours) at eight different time points. † denotes values taken from (Zhang et al., 2025d).

| Model | $t = 1$ $\mathcal{W}_1$ | $t = 2$ $\mathcal{W}_1$ | $t = 3$ $\mathcal{W}_1$ | $t = 4$ $\mathcal{W}_1$ | $t = 5$ $\mathcal{W}_1$ | $t = 6$ $\mathcal{W}_1$ | $t = 7$ $\mathcal{W}_1$ |
|---|---|---|---|---|---|---|---|
| DeepRUOT † | **8.0447**$_{\pm 0.0005}$ | 8.0773$_{\pm 0.0021}$ | 7.6301$_{\pm 0.0032}$ | **8.0064**$_{\pm 0.0042}$ | **7.9018**$_{\pm 0.0117}$ | 8.3977$_{\pm 0.0102}$ | 7.8346$_{\pm 0.0109}$ |
| CytoBridge † | 8.0448$_{\pm 0.0005}$ | 8.0771$_{\pm 0.0021}$ | **7.6299**$_{\pm 0.0032}$ | 8.0066$_{\pm 0.0043}$ | **7.9018**$_{\pm 0.0117}$ | **8.3974**$_{\pm 0.0102}$ | 7.8343$_{\pm 0.0109}$ |
| **BranchSBM (Ours)** | 11.9774$_{\pm 0.0000}$ | **7.4643**$_{\pm 2.4031}$ | 11.5204$_{\pm 0.0000}$ | 11.2593$_{\pm 0.0000}$ | 10.2888$_{\pm 0.0000}$ | 8.7301$_{\pm 0.0000}$ | **6.8702**$_{\pm 0.0000}$ |

After simulating validation samples from the initial distribution $x_0 \sim \pi_0$, we evaluate the reconstructed distributions at the intermediate held-out time point $t_1$ and final time point $t_2$ ($t = 1$). In Figure 4A, we observe that single-branch SBM trained with a single target distribution $p_1$ containing both terminal fates fails to learn distinct branched trajectories, and the simulated cell states at time $t_2$ do not reach either of the terminal distributions. In contrast, we show that BranchSBM simulates branched states at intermediate time steps not included in the training data while accurately reconstructing both target distributions with significantly lower 1-Wasserstein and 2-Wasserstein distances compared to the single-branch SBM model (Figure 4B-D; Table 3).

**Pancreatic $\beta$-Cell Differentiation Results** We use a dataset consisting of pancreatic $\beta$-cell differentiation data (Veres et al., 2019) containing $51,274$ cells collected over eight time points as they evolve from human pluripotent stem cells to pancreatic $\beta$-like cells. This data contains eight time points $t_i$ for $i \in \{0, \dots, 7\}$ that are projected to 30-dimensional PC representations

Table 3: **Results for Modeling Single-Cell Differentiation.** Wasserstein distances ($\mathcal{W}_1$ and $\mathcal{W}_2$) between simulated and ground-truth cell distributions at time $t_1$ and $t_2$ on the validation dataset. BranchSBM reconstructs both intermediate and terminal states significantly better than single-branch SBM. Results are averaged over 5 independent runs.

| Model | Single Branch SBM | | **BranchSBM** | |
|---|---|---|---|---|
| Time ↓ | $\mathcal{W}_1 (\downarrow)$ | $\mathcal{W}_2 (\downarrow)$ | $\mathcal{W}_1 (\downarrow)$ | $\mathcal{W}_2 (\downarrow)$ |
| $t_1$ | $0.582_{\pm 0.020}$ | $0.703_{\pm 0.008}$ | **0.366**$_{\pm 0.034}$ | **0.479**$_{\pm 0.044}$ |
| $t_2$ | $0.940_{\pm 0.075}$ | $1.037_{\pm 0.074}$ | **0.210**$_{\pm 0.042}$ | **0.265**$_{\pm 0.046}$ |

$x \in \mathbb{R}^{30}$. We use Leiden clustering to define $K = 11$ distinct target distributions of samples at time $t_7$ and set their target weights relative to the total weight at $t_0$ (App E.4). We used samples across all time steps to define the data manifold via the RBF metric $V_{t,\eta}^{\text{RBF}}$. BranchSBM was trained on pairs sampled only from $t_0$ and $t_7$, and the distributions at all intermediate time points were inferred from learning to minimize the distance from the data manifold over time. For baselines, we compared two SOTA methods for single-cell trajectory modeling: DeepRUOT (Zhang et al., 2025c) and CytoBridge (Zhang et al., 2025d), which are trained explicitly on intermediate snapshots to reconstruct the stochastic dynamics of cells that evolve independently.

Notably, BranchSBM not only reconstructs the multi-modal terminal distribution at $t_7$ with superior accuracy against all baselines, but also produces intermediate trajectories that are competitive with models trained directly on intermediate snapshots using explicit reconstruction losses (Table 2; Appendix Figure 9). Leveraging the RBF state cost $V_{t,\eta}^{\text{RBF}}$, which encourages trajectories to remain on the underlying data manifold, BranchSBM effectively captures the true differentiation dynamics through the combined influence of the neural interpolant and the path-energy objective. These results demonstrate that BranchSBM scales reliably to a large number of branches and is particularly advantageous in settings where intermediate timepoints are sparse or unavailable, allowing the model to infer biologically meaningful trajectories even with limited temporal supervision.

## 5.3 Cell-State Perturbation Modeling

Predicting the effects of perturbation on cell state dynamics is a crucial problem for therapeutic design. In this experiment, we leverage BranchSBM to model the trajectories of a single cell line from a single homogeneous state to multiple heterogeneous states after a drug-induced perturbation. We demonstrate that BranchSBM is capable of modeling high-dimensional gene expression data and learning branched trajectories that accurately reconstruct diverging perturbed cell populations.

**Setup**   For this experiment, we extract the data for a single cell line (A-549) under perturbation with Clonidine and Trametinib at 5 $\mu$L, selected based on cell abundance and response diversity from the Tahoe-100M dataset (Zhang et al., 2025a). Since both drug perturbation datasets contained over 60K genes, we selected the top 2000 highly variable genes (HVGs) based on normalized expression and performed principal component analysis (PCA) to find the top PCs that capture the variance in the data. We set the initial distribution at $t = 0$ to be a control DMSO-treated cell population and the target distributions at $t = 1$ to be distinct clusters in the drug-treated cell population. After clustering, we identified two divergent clusters in the Clonidine-perturbed population and three in the Trametinib-perturbed population (Appendix Figure 11).

To determine the weights of each branch, we take the ratio of each cluster with respect to the total perturbed cell population (Appendix Table 9). For both experiments, we simulated the top 50 PCs, which capture approximately 38% of the variance in the dataset. To further evaluate the scalability of BranchSBM on simulating trajectories in high-dimensional state spaces, we simulated the top 100 and 150 PCs for Clonidine and compared the performance across dimensions.

Table 4: **Results for Clonidine Perturbation Modeling for Increasing Principal Component Dimensions.** Maximum-mean discrepancy (MMD) across all PCs and Wasserstein distances ($\mathcal{W}_1$ and $\mathcal{W}_2$) of top 2 PCs between ground truth and reconstructed distributions at $t = 1$ simulated from the validation data at $t = 0$. Results for single-branch SBM (50 PCs) and BranchSBM (2 branches) were averaged over 5 independent runs.

| Model | RBF-MMD ($\downarrow$) | $\mathcal{W}_1$ ($\downarrow$) | $\mathcal{W}_2$ ($\downarrow$) |
|---|---|---|---|
| Single Branch SBM (50 PCs) | $0.279_{\pm 0.024}$ | $5.124_{\pm 0.509}$ | $6.149_{\pm 0.463}$ |
| **BranchSBM** | | | |
| 50 PCs | $0.065_{\pm 0.001}$ | $\mathbf{1.076}_{\pm 0.085}$ | $\mathbf{1.224}_{\pm 0.097}$ |
| 100 PCs | $\mathbf{0.053}_{\pm 0.002}$ | $1.832_{\pm 0.174}$ | $2.037_{\pm 0.174}$ |
| 150 PCs | $0.083_{\pm 0.001}$ | $1.722_{\pm 0.064}$ | $1.931_{\pm 0.035}$ |

Given that the intermediate trajectory between the control and perturbed state is unknown, we assume that the optimal trajectory both minimizes the kinetic energy of the drift field while minimizing the distance from the space of *feasible* cell states. We define the state cost $V_t(X_t)$ with the RBF metric (Kapuśniak et al., 2024; Arvanitidis et al., 2016), which pushes the intermediate trajectory to lie near states represented in the dataset. Further details are provided in Appendix E.5.

**Clonidine Perturbation Results**   After multi-stage training of BranchSBM with $d \in \{50, 100, 150\}$ PCs and two branched endpoints (Appendix Figure 6A), we simulated the final perturbed state of each branch at time $t = 1$ from the samples in the initial validation data distribution $x_0 \sim \pi_0$ corresponding to the control DMSO condition. In Appendix Figure 6C, we demonstrate that BranchSBM accurately reconstructs the ground-truth distributions of endpoint 0 (top row) and endpoint 1 (bottom row) across increasing PC dimensions, capturing the location and spread of the dataset. To prove the necessity of our branched framework, we simulate the target distribution with only a single endpoint distribution $p_1$ containing both clusters with single-branch SBM and show that it only reconstructs the population of cells in endpoint 0, which represent cells closest to the control cells along PC2, and

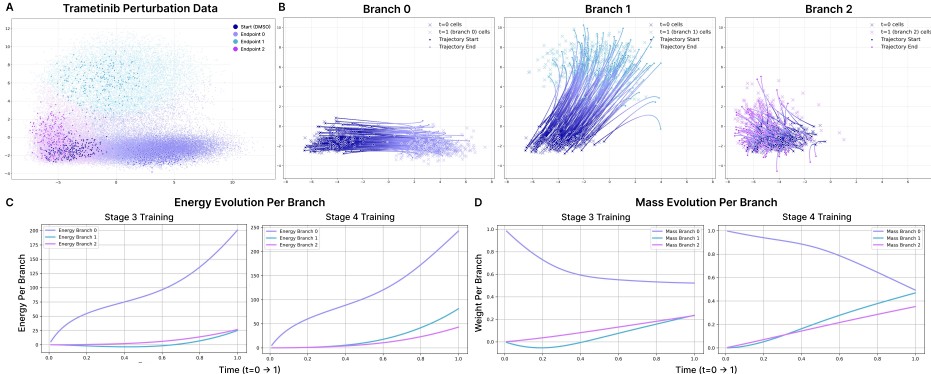

Figure 5: **Results for Trametinib Perturbation Modeling with BranchSBM. (A)** Gene expression data of DMSO control ($t = 0$) and cells after treatment with $5\mu M$ Trametinib ($t = 0$) with three distinct endpoints (purple, turquoise, and pink). **(B)** The simulated endpoints of the top 50 PCs at $t = 1$ on the validation data for each branch. **(C)** The evolution of cumulative energy across $t \in [0, 1]$ calculated as (14) along each branched trajectory after Stage 3 (growth with fixed drift) and Stage 4 (joint) training. **(D)** The evolution of mass across $t \in [0, 1]$ along each branched trajectory with target weights of $w_{1,0} = 0.603$, $w_{1,1} = 0.255$ and $w_{1,2} = 0.142$.

fails to differentiate cells in cluster 1 that differ from cluster 0 in higher-dimensional PCs (Appendix Figure 6B). Concretely, BranchSBM used across all PC dimensions outperforms single-branch SBM on only 50 PCs (Table 4), indicating that BranchSBM is required to model complex perturbation effects in high-dimensional gene expression spaces.

**Trametinib Perturbation Results**  We further show that BranchSBM can scale beyond two branches, by modeling the perturbed cell population of Trametinib-treated cells, which diverge into three distinct clusters (Figure 5A). We trained BranchSBM with three endpoints and single-branch SBM with one endpoint containing all three clusters on the top 50 PCs.

After simulating the trajectories over time $t \in [0, 1]$ on the validation cells in the control population, we show that BranchSBM generates clear trajectories to all three branched endpoints (Figure 5B) and reconstructs the overall target distribution with lower error compared to single-branch SBM (Table 5). In Figure 5C and D, we plot the evolution of cumulative energy calculated in

Table 5: **Results for Trametinib Perturbation Modeling.** Maximum-mean discrepancy (MMD) across all 50 PCs and Wasserstein distances ($\mathcal{W}_1$ and $\mathcal{W}_2$) of top 2 PCs between ground truth and reconstructed distributions at $t = 1$ simulated from the validation data at $t = 0$. Results were averaged over 5 independent runs.

| Model | RBF-MMD ($\downarrow$) | $\mathcal{W}_1$ ($\downarrow$) | $\mathcal{W}_2$ ($\downarrow$) |
|---|---|---|---|
| Single Branch SBM | $0.246_{\pm 0.013}$ | $5.428_{\pm 0.234}$ | $6.426_{\pm 0.186}$ |
| **BranchSBM** | $\mathbf{0.053}_{\pm 0.001}$ | $\mathbf{0.838}_{\pm 0.061}$ | $\mathbf{0.973}_{\pm 0.050}$ |

$\mathcal{L}_{\text{energy}}(\theta, \phi)$ (14) and weight of each branch over $t \in [0, 1]$, demonstrating that BranchSBM's multi-stage training scheme effectively learns the optimal trade-off between minimizing the energy across trajectories and matching the target weights of each branch.

## 6  CONCLUSION

In this work, we introduce **Branched Schrödinger Bridge Matching (BranchSBM)**, a novel matching framework that solves the Generalized Schrödinger Bridge (GSB) problem from an initial distribution to multiple weighted target distributions through the division of mass across learned branched trajectories. BranchSBM solves the branched Schrödinger bridge problem defined as a sum of Unbalanced Conditional Stochastic Optimal Control tasks, parameterizing branch-specific velocity and growth rates with neural networks to predict system trajectories with a single inference simulation. Through applications to nonlinear 3D navigation, cell differentiation, and perturbation-induced gene expression, we demonstrate that BranchSBM provides a unified and flexible framework for modeling complex branched dynamics across biological and physical systems.

## REPRODUCIBILITY STATEMENT

We have taken multiple steps to ensure the reproducibility of BranchSBM. All theoretical derivations are fully detailed in the main text and Appendix, with proofs of Propositions 3.1-4.2 provided in Sections C.1-C.4. Our multi-stage training procedure is described explicitly, and pseudocode is given in Algorithm 1. Experimental setups for each task are described in detail, including synthetic LiDAR navigation (Appendix E.3), differentiating single-cell dynamics (Appendix E.4), and cell-state perturbation modeling (Appendix E.5). We specify all hyperparameters and architectures in Table 11, evaluation metrics such as Wasserstein distances and MMD are defined in Equations 102, 103, and 104, and additional loss functions are given in Equations 12-17. The codebase is freely accessible to the academic community at `https://github.com/sophtang/BranchSBM` and `https://huggingface.co/ChatterjeeLab/BranchSBM`.

## ETHICS STATEMENT

This work focuses on the development of a theoretical and computational framework for learning branched Schrödinger bridges in generative modeling. All experiments are conducted on publicly available or synthetic datasets, including 3D LiDAR manifolds, mouse hematopoiesis scRNA-seq data, and the Tahoe-100M perturbation dataset, and do not involve any personally identifiable, sensitive human subject data, or animal experiments. BranchSBM provides a general-purpose algorithm for modeling stochastic processes with branching dynamics, with potential applications in biological modeling, navigation, and physics. Possible risks, such as misuse in simulating sensitive biomedical data, are mitigated by the fact that BranchSBM is a general generative framework that requires curated training datasets and does not directly generate actionable biological interventions. We believe the potential benefits of this work, in advancing generative modeling theory and enabling more accurate modeling of branching systems in biology and beyond, outweigh possible risks.

## DECLARATIONS

**Acknowledgments**    We thank Mark III Systems, Penn Computing and Educational Technology Services (CETS), and Penn Advanced Research Computing Center (PARCC), for providing database and hardware support that has contributed to the research reported within this manuscript.

**Author Contributions**    S.T. devised and developed model architectures and theoretical formulations, and trained and benchmarked models. Y.Z. advised on model design and theoretical framework, and processed data for training. S.T. drafted the manuscript and S.T. and Y.Z. designed the figures. A.T. reviewed mathematical formulations and provided advising. P.C. designed, supervised, and directed the study, and reviewed and finalized the manuscript.

**Data and Materials Availability**    The codebase is freely accessible to the academic community at `https://github.com/sophtang/BranchSBM` and `https://huggingface.co/ChatterjeeLab/BranchSBM`.

**Funding Statement**    This research was supported by NIH grant R35GM155282 to the lab of P.C.

**Competing Interests**    P.C. is a co-founder of Gameto, Inc., UbiquiTx, Inc., AtomBioworks, Inc., and Recognition Bio, Inc., and advises companies involved in biologics development and cell engineering. P.C.'s interests are reviewed and managed by the University of Pennsylvania in accordance with their conflict-of-interest policies. S.T., Y.Z., and A.T. have no conflicts of interest to declare.

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

OUTLINE OF APPENDIX

In Appendix A, we provide an extended background on the relevant theory for learning optimal stochastic bridges (A.1) and simulating trajectories on the data manifold (A.2). In Appendix B, we discuss the relationship between our proposed formulation for BranchSBM and previous related works. Appendix C provides the theoretical basis for Sections 3 and 4, including formal proofs for Proposition 3.1 (C.1), Proposition 3.2 (C.2), Proposition 4.1 (C.3), and Proposition 4.2 (C.4). We provide additional experiment results in Appendix D. In Appendix E, we describe further details on experiments and hyperparameters used including specific details for each experiment, including multi-path LiDAR navigation (E.3), modeling differentiating single-cells (E.4), and modeling cell-state perturbations (E.4). Finally, we provide the pseudo code for the multi-stage training algorithm in Appendix F.

**Notation**   We denote the state space as $\mathcal{X} \subseteq \mathbb{R}^d$ and the time interval as $t \in [0,1]$. The branches are indexed with $k \in \{0, \ldots, K\}$. We denote initial data distribution at time $t = 0$ as $\pi_0$ and the terminal data distributions at $t = 1$ as $\{\pi_{1,k}\}_{k=0}^K$. The joint data distribution is denoted $\pi_{0,1,k}$ and a pair of samples is given by $(\boldsymbol{x}_0, \boldsymbol{x}_{1,k}) \sim \pi_{0,1,k}$. For simplicity, we denote $\pi_{0,1,k} \equiv \pi(\boldsymbol{x}_0, \boldsymbol{x}_{1,k})d\boldsymbol{x}_0 d\boldsymbol{x}_{1,k} = \pi_{0,1,k}(d\boldsymbol{x}_0, d\boldsymbol{x}_{1,k})$. Let $u_{t,k}(X_{t,k})$ denote the marginal velocity field, $g_{t,k}(X_{t,k})$ denote the growth rate, and $p_{t,k}(X_{t,k})$ denote the marginal probability density for the $k$th branch, where we sometimes drop the input $X_{t,k}$ for simplicity. In addition, we denote the conditional velocity field and probability density as $u_{t|0,1,k} \equiv u_{t|0,1,k}(X_{t,k}|\boldsymbol{x}_0, \boldsymbol{x}_{1,k})$ and $p_{t|0,1,k} \equiv p_{t|0,1,k}(X_{t,k}|\boldsymbol{x}_0, \boldsymbol{x}_{1,k})$ respectively. The optimal values for any quantity are superscripted with a $\star$ symbol. We denote the parameterized flow neural networks as $u_{t,k}^\theta$ with parameters $\theta$ and the growth neural networks with $g_{t,k}^\phi$ with parameters $\phi$. In the context of unbalanced endpoint distributions, we denote the true initial weight of a sample as $w_0^\star$ and the final weight of a sample from the $k$th target distribution as $w_{1,k}^\star$. The predicted weights generated from the growth dynamics are given by $w_{t,k}$, and we seek to match the predicted weight at time $t = 1$ given by $w_{1,k}$ to the true weight $w_{1,k}^\star$. $L^2$ denotes the space of square integrable functions and $\|\cdot\|_{L^2}$ be the $L^2$-norm in function space. $L^\infty$ denotes the space of essentially bounded functions such that $\|f\|_\infty < \infty$.

# A   EXTENDED THEORETICAL BACKGROUND

## A.1   LEARNING OPTIMAL STOCHASTIC BRIDGES

**Pinned-Down Stochastic Bridges**   Let $\mathbb{Q} \in \mathcal{M}$ be a Markovian reference path measure that evolves over $t \in [0,1]$ according to a drift field $f_t(X_t) : \mathbb{R}^d \to \mathbb{R}^d$ and stochastic $d$-dimensional Brownian motion $B_t \in \mathbb{R}^d$, via the SDE

$$dX_t = f_t(X_t)dt + \sigma_t dB_t, \quad X_0 \sim \pi_0 \tag{20}$$

Given $\mathbb{Q}$, consider a stochastic process $(X_t)_{t \in [0,1]}$ over the time interval $t \in [0,1]$ *pinned-down* at the initial point $X_0 = \boldsymbol{x}_0$ and final point $X_1 = \boldsymbol{x}_1$ denoted as $\mathbb{Q}_{\cdot|0,1}(\cdot|\boldsymbol{x}_0, \boldsymbol{x}_1)$. Due to the endpoint conditions, $\mathbb{Q}_{\cdot|0,1}$ is not necessarily Markov, and evolves via the SDE

$$dX_t = \{f_t(X_t) + \sigma_t^2 \nabla_{\boldsymbol{x}} \log \mathbb{Q}_{1|t}(\boldsymbol{x}_1|X_t)\}dt + \sigma_t dB_t, \quad X_0 = \boldsymbol{x}_0 \tag{21}$$

where $\nabla_{\boldsymbol{x}} \log \mathbb{Q}_{1|t}(\boldsymbol{x}_1|X_t)$ is a non-Markovian score function that *corrects* the drift field $f_t(X_t)$ of the reference process such that it points toward the target endpoint $\boldsymbol{x}_1$. Since $\log \mathbb{Q}_{1|t}(\boldsymbol{x}_1|X_t)$ is the log-likelihood that the final state satisfies the condition $X_1 = \boldsymbol{x}_1$, the gradient defines how the log-likelihood changes with respect to the changes in the state $\boldsymbol{x}$ at time $t$. The drift moves $\boldsymbol{x}$ in the direction of the largest increase in log-likelihood given by the score function, which ensures that the process satisfies $X_1 = \boldsymbol{x}_1$ following the theory of Doob's $h$-transform (Rogers & Williams, 2000). Now, we can define the conditional probability distribution $p_t$ as a *mixture* of pinned-down stochastic bridges over pairs of endpoints in the data coupling $\pi_{0,1} = \pi_0 \otimes \pi_1$ given by

$$p_t(\boldsymbol{x}) = \pi_{0,1}\mathbb{Q}_{t|0,1}(\boldsymbol{x}|\boldsymbol{x}_0, \boldsymbol{x}_1) = \int \mathbb{Q}_{t|0,1}(\boldsymbol{x}|\boldsymbol{x}_0, \boldsymbol{x}_1)\pi_{0,1}(d\boldsymbol{x}_0, d\boldsymbol{x}_1) \tag{22}$$

To simplify notation, we denote each conditional bridge as $\mathbb{Q}_{t|0,1}(X_t|\boldsymbol{x}_0, \boldsymbol{x}_1) = p_{t|0,1}(X_t|\boldsymbol{x}_0, \boldsymbol{x}_1)$ and the joint distribution $\pi_{0,1}(\boldsymbol{x}_0, \boldsymbol{x}_1)$. Now, we can rewrite the marginal $p_t$ as

$$p_t(X_t) = \int p_{t|0,1}(X_t|\boldsymbol{x}_0, \boldsymbol{x}_1)\pi_{0,1}(d\boldsymbol{x}_0, d\boldsymbol{x}_1) = \mathbb{E}_{\pi_{0,1}}\left[p_{t|0,1}(X_t|\boldsymbol{x}_0, \boldsymbol{x}_1)\right] \tag{23}$$

Furthermore, we denote $u_{t|0,1} \equiv u_{t|0,1}(X_t|\boldsymbol{x}_0, \boldsymbol{x}_1) = \{f_t(X_t) + \sigma_t^2 \nabla_{\boldsymbol{x}} \log \mathbb{Q}_{1|t}(\boldsymbol{x}_1|X_t)\}$ as the conditional drift that *generates* $p_{t|0,1} \equiv p_{t|0,1}(X_t|\boldsymbol{x}_0, \boldsymbol{x}_1)$, that satisfies the conditional Fokker-Planck equation

$$\frac{\partial}{\partial t}p_{t|0,1} = -\nabla \cdot (u_{t|0,1}p_{t|0,1}) + \frac{1}{2}\sigma^2 \Delta p_{t|0,1} \tag{24}$$

**Definition A.1** (Reciprocal Class). *Given our definition of the conditional bridge $\mathbb{Q}_{\cdot|0,1}$, we can define the reciprocal class, denoted $\mathcal{R}(\mathbb{Q})$, of the reference measure $\mathbb{Q}$ as the class of path measures that share the same bridge as $\mathbb{Q}$, defined as $\mathcal{R}(\mathbb{Q}) = \{\Pi \mid \Pi(X_t|\boldsymbol{x}_0, \boldsymbol{x}_1) = \mathbb{Q}(X_t|\boldsymbol{x}_0, \boldsymbol{x}_1)\}$.*

**Markovian Projections**    Given a mixture of conditional stochastic bridges $\Pi = \Pi_{0,1}\mathbb{Q}_{\cdot|0,1}$ under the reference measure $\mathbb{Q}$ that require knowledge of the joint distribution $\pi_{0,1}$, we aim to project $\Pi$ to the space of Markovian measures $\mathcal{M}$, where the drift dynamics $u_t(X_t)$ is only dependent on the current state $X_t$ and require no knowledge on the endpoints. This allows us to parameterize a Markovian drift $u_t^\theta(X_t)$ that can transport samples from the initial distribution $\boldsymbol{x}_0 \sim \pi_0$ to samples from the target distribution $\boldsymbol{x}_1 \sim \pi_1$. To do this, we define the *Markovian projection* of $\Pi$ (Shi et al., 2023; Liu et al., 2023b).

**Definition A.2** (Markovian Projection). *Given a conditional bridge $\mathbb{Q}_{\cdot|0,1}$ that evolves via the SDE in (21), we define a Markovian projection of the mixture of bridges $\Pi = \Pi_{0,1}\mathbb{Q}_{\cdot|0,1}$ as a Markov process $\mathbb{M}^\star = proj_{\mathcal{M}}(\Pi) \in \mathcal{M}$ with the same marginals as $\Pi$ such that $X_t \sim \Pi_t$ for all $t \in [0, 1]$, $X_1 \sim \pi_1$ and evolves via the SDE*

$$dX_t = \{f_t(X_t) + v_t^\star(X_t)\}dt + \sigma_t dB_t \tag{25}$$

$$v_t^\star(X_t) = \sigma^2 \mathbb{E}_{\Pi_{1|t}}\left[\nabla_{\boldsymbol{x}_t} \log \mathbb{Q}_{1|t}(X_1|X_t)|X_t = \boldsymbol{x}_t\right] \tag{26}$$

*where $\Pi_{1|t}$ is the conditional distribution of $X_1$ under the mixture of bridges $\Pi$ and $\nabla_{\boldsymbol{x}_t} \log \mathbb{Q}_{1|t}(X_1|X_t)$ points in the direction of greatest increase in the log-likelihood of the target endpoint $X_1 \sim \pi_1$ under the reference process $\mathbb{Q}$. In addition, the Markov projection $\mathbb{M}^\star$ minimizes the KL-divergence with the mixture of bridges $\mathbb{M}^\star = \arg\min_{\mathbb{M}\in\mathcal{M}} KL(\Pi\|\mathbb{M})$ and can be obtained by parameterizing $v_t^\theta(X_t)$ and minimizing the dynamic formulation given by*

$$KL(\Pi\|\mathbb{M}) = \mathbb{E}_{(\boldsymbol{x}_0, \boldsymbol{x}_1)\sim\Pi_{0,1}}\mathbb{E}_{\boldsymbol{x}_t\sim\Pi_{t|0,1}} \int_0^1 \frac{1}{2\sigma_t^2}\left[\left\|\sigma_t^2 \nabla_{\boldsymbol{x}_t} \log \mathbb{Q}_{1|t}(\boldsymbol{x}_1|\boldsymbol{x}_t) - v_t^\theta(\boldsymbol{x}_t)\right\|_2^2\right]dt \tag{27}$$

In general, the Markovian projection of a reference measure $\mathbb{Q}$ does not preserve the conditional bridge and is not in the reciprocal class $\mathcal{R}(\mathbb{Q})$. The unique path measure $\mathbb{P}$ that is the Markovian projection of $\mathbb{Q}$, is in the reciprocal class $\mathcal{R}(\mathbb{Q})$, *and* preserves the endpoint distributions, is called the *Schrödinger Bridge*.

**Definition A.3** (Schrödinger Bridge). *Given a reference measure $\mathbb{Q}$, a initial distribution $\pi_0$, and final distribution $\pi_1$. A path measure $\mathbb{P}$ is the unique Schrödinger bridge if it satisfies*

1. *$\mathbb{P}$ is the Markovian projection of $\mathbb{Q}$, such that $\mathbb{P} = proj_{\mathcal{M}}(\mathbb{Q})$.*
2. *$\mathbb{P}$ is in the reciprocal class of $\mathbb{Q}$, i.e. $\mathbb{P} \in \mathcal{R}(\mathbb{Q})$, such that it preserves the conditional bridge $\mathbb{P}(X_t|\boldsymbol{x}_0, \boldsymbol{x}_1) = \mathbb{Q}(X_t|\boldsymbol{x}_0, \boldsymbol{x}_1)$.*
3. *$\mathbb{P}$ preserves the endpoint distributions $\mathbb{P}_0 = \pi_0$ and $\mathbb{P}_1 = \pi_1$.*

The goal of Schrödinger Bridge Matching (SBM) is to estimate the Schrödinger Bridge that transports samples from an initial distribution $\pi_0$ to a final distribution $\pi_1$ given the optimal reference dynamics. We further discuss previous approaches to solving the SB problem in Appendix B.

### A.2 SIMULATING TRAJECTORIES ON THE DATA MANIFOLD

**Riemannian Manifolds and Metrics** Since the interpolant $\boldsymbol{x}_{t,\eta}$ learned in Stage 1 is defined by minimizing a non-linear state cost $V_t(X_t)$, the resulting velocity field $u_t^\theta(X_t)$ lies on the tangent bundle $\mathcal{T}_{\boldsymbol{x}}\Omega$ of a smooth $d$-dimensional manifold $\Omega \in \mathbb{R}^d$ called a *Riemannian manifold*. Intuitively, a Riemannian manifold can be thought of as a smooth surface where the local curvature around a point $\boldsymbol{x} \in \Omega$ can be approximated by a tangent space $\mathcal{T}_{\boldsymbol{x}}\Omega$ that defines the set of directions in which $\boldsymbol{x}$ can move along the manifold. These *directions* are defined by a set of tangent vectors $u \in \mathcal{T}_{\boldsymbol{x}}\Omega$, which *push* the point $\boldsymbol{x}$ along the manifold.

In Stage 2, we seek to parameterize a vector field $u_t^\theta(X_t)$ that minimizes the *angle* from the tangent vector $\dot{\boldsymbol{x}}_{t,\eta} = \partial_t \boldsymbol{x}_{t,\eta}$ at each point $\boldsymbol{x}$ on the manifold optimized in Stage 1. To do this, we must define the concepts of *length* and *angles* on Riemannian manifolds. First, we define a *location-dependent inner product* in Riemannian manifolds known as the *Riemannian metric* $g_{\boldsymbol{x}} : \mathcal{T}_{\boldsymbol{x}}\Omega \times \mathcal{T}_{\boldsymbol{x}}\Omega \to \mathbb{R}$ that maps two vectors $u, v \in \mathcal{T}_{\boldsymbol{x}}\Omega$ to a scalar that describes the relative direction and length of the two vectors. Formally, the Riemannian metric can be written as the *billinear* and *positive definite* function

$$g_{\boldsymbol{x}}(u,v) = u^\top \mathbf{G}(\boldsymbol{x})v = \langle u, \mathbf{G}v \rangle \ \text{ s.t. } \ \begin{cases} \forall u \neq 0 \ \ g_{\boldsymbol{x}}(u,u) > 0 \\ \mathbf{G} \succ 0 \end{cases} \tag{28}$$

which defines the norm of a tangent vector as $\|u\|_{g_{\boldsymbol{x}}} = \sqrt{g_{\boldsymbol{x}}(u,u)}$. Now, we can decompose the Riemannian norm of the tangent vector $\|\dot{\boldsymbol{x}}_{t,\eta}\|_{g_{\boldsymbol{x}}}$ to get the loss defined in (12) as follows

$$\mathcal{L}_{\text{traj}}(\eta) = \mathbb{E}_{t,(\boldsymbol{x}_0,\boldsymbol{x}_1)\sim\pi_{0,1}} \left[ \|\dot{\boldsymbol{x}}_{t,\eta}\|_{g_{\boldsymbol{x}}}^2 \right] \tag{29}$$

$$= \mathbb{E}_{t,(\boldsymbol{x}_0,\boldsymbol{x}_1)\sim\pi_{0,1}} \langle \dot{\boldsymbol{x}}_{t,\eta}, \mathbf{G}(\boldsymbol{x}_{t,\eta})\dot{\boldsymbol{x}}_{t,\eta} \rangle \tag{30}$$

$$= \mathbb{E}_{t,(\boldsymbol{x}_0,\boldsymbol{x}_1)\sim\pi_{0,1}} \left[ \|\dot{\boldsymbol{x}}_{t,\eta}\|_2^2 + \langle \dot{\boldsymbol{x}}_{t,\eta}, (\mathbf{G}(\boldsymbol{x}_{t,\eta}) - \mathbf{I})\dot{\boldsymbol{x}}_{t,\eta} \rangle \right] \tag{31}$$

$$= \mathbb{E}_{t,(\boldsymbol{x}_0,\boldsymbol{x}_1)\sim\pi_{0,1}} \left[ \|\dot{\boldsymbol{x}}_{t,\eta}\|_2^2 + V_t(\boldsymbol{x}_{t,\eta}) \right] \tag{32}$$

where the state cost is defined as $V_t(\boldsymbol{x}_{t,\eta}) = \langle \dot{\boldsymbol{x}}_{t,\eta}, (\mathbf{G}(\boldsymbol{x}_{t,\eta}) - \mathbf{I})\dot{\boldsymbol{x}}_{t,\eta} \rangle$.

**Data-Dependent State Cost** Following Kapuśniak et al. (2024), we define the metric matrix $\mathbf{G}(\boldsymbol{x}_{t,\eta})$ described previously as the data-dependent LAND and RBF metrics of the form $\mathbf{G}_{\text{LAND}}(\boldsymbol{x},\mathcal{D}) = \mathbf{G}_{\text{RBF}}(\boldsymbol{x},\mathcal{D}) = (\text{diag}(\mathbf{h}(\boldsymbol{x})) - \varepsilon\mathbf{I})^{-1}$ which assigns higher cost (i.e. $\|\mathbf{G}(\boldsymbol{x})\|$ is larger) when $\boldsymbol{x}$ moves away from the support of the dataset $\mathcal{D}$. Specifically, given a dataset $\mathcal{D} = \{\boldsymbol{x}_i\}_{i=1}^N$, we define the elements $\mathbf{h}^{\text{LAND}}(\boldsymbol{x}) \in \mathbb{R}^d$ that scales down each dimension $j \in \{1, \ldots, d\}$ in the LAND metric as

$$h_j^{\text{LAND}}(\boldsymbol{x}) = \sum_{i=1}^N (x_i^j - x^j)^2 \exp\left( -\frac{\|\boldsymbol{x} - \boldsymbol{x}_i\|_2^2}{2\sigma^2} \right) \tag{33}$$

where the $\exp$ term is positive when there is a high concentration of data points around the point $\boldsymbol{x}$ (i.e. $\|\boldsymbol{x} - \boldsymbol{x}_i\|$ is small) and approaches 0 as the concentration of data around $\boldsymbol{x}$ decreases (i.e. $\|\boldsymbol{x} - \boldsymbol{x}_i\|$ is large). Writing $\langle \dot{\boldsymbol{x}}_{t,\eta}, \mathbf{G}(\boldsymbol{x}_{t,\eta})\dot{\boldsymbol{x}}_{t,\eta} \rangle$ in terms of $h_j(\boldsymbol{x})$, we get

$$\mathcal{L}_{\text{traj}}(\eta) = \mathbb{E}_{t,(\boldsymbol{x}_0,\boldsymbol{x}_1)\sim\pi_{0,1}} \langle \dot{\boldsymbol{x}}_{t,\eta}, \mathbf{G}(\boldsymbol{x}_{t,\eta})\dot{\boldsymbol{x}}_{t,\eta} \rangle \tag{34}$$

$$= \mathbb{E}_{t,(\boldsymbol{x}_0,\boldsymbol{x}_1)\sim\pi_{0,1}} \left[ \sum_{j=1}^d \frac{(\dot{\boldsymbol{x}}_{t,\eta})_j^2}{h_j(\boldsymbol{x}_{t,\eta}) + \varepsilon} \right] \tag{35}$$

When $h_j(\boldsymbol{x})$ is large, the loss is minimized, and when $h_j(\boldsymbol{x})$ is small, the loss is scaled up. While the LAND metric effectively defines the data manifold in low dimensions, in high-dimensional state spaces, setting a suitable variance $\sigma$ in $h_j^{\text{LAND}}(\boldsymbol{x}_t)$ to ensure that the path does not deviate far from the data manifold without overfitting is challenging. To overcome this limitation, the RBF metric clusters the dataset into $N_c$ clusters with centroids denoted as $\hat{\boldsymbol{x}}_n \in \mathbb{R}^d$ and trains a set of parameters $\{\omega_{\alpha,n}\}_{n=1}^{N_c}$ to enforce $h_j(\boldsymbol{x}_i) \approx 1$ for all points in the dataset such that points $\boldsymbol{x}$ within the data

manifold are also assigned $h_j(\boldsymbol{x}) \approx 1$. Specifically, $h_j^{\mathrm{RBF}}$ is defined as

$$h_j^{\mathrm{RBF}}(\boldsymbol{x}) = \sum_{n=1}^{N_c} \omega_{n,j}(\boldsymbol{x}) \exp\left(-\frac{\lambda_{n,j}}{2}\|\boldsymbol{x} - \hat{\boldsymbol{x}}_n\|_2^2\right) \tag{36}$$

$$\lambda_n = \frac{1}{2}\left(\frac{\kappa}{|C_n|}\sum_{\boldsymbol{x}\in C_n}\|\boldsymbol{x} - \hat{\boldsymbol{x}}_n\|_2^2\right)^{-2} \tag{37}$$

where $C_n$ denotes the $n$th cluster, $\kappa$ is a tunable hyperparameter, and $\lambda_{n,j}$ is the bandwidth of cluster $n$ the $j$th dimension. To train the parameters, we minimize the following loss

$$\mathcal{L}_{\mathrm{RBF}}(\{\omega_{\alpha,n}\}) = \sum_{\boldsymbol{x}_i\in\mathcal{D}}\left(1 - h_n^{\mathrm{RBF}}(\boldsymbol{x}_i)\right)^2 \tag{38}$$

In our experiments, we use the LAND metric for the LiDAR and mouse hematopoiesis datasets with dimensions $d = 2$ and $d = 3$ respectively, and the RBF metric for the perturbation modeling experiment with gene expression data of dimensions $d \in \{50, 100, 150\}$.

**Sampling on the Data Manifold** In Riemannian geometry, each Euclidean step $\Delta t \cdot u_t^\theta(\boldsymbol{x})$ following the tangent vector $u_t^\theta(\boldsymbol{x})$ along the manifold requires mapping back to the manifold via an exponential map $X_{t+\Delta t} = \exp_{\boldsymbol{x}}(\Delta t \cdot u_t^\theta(\boldsymbol{x}))$. In general, computing the exponential map $\exp_{\boldsymbol{x}}$ under the Riemannian metric $\mathbf{G}(\boldsymbol{x})$ requires simulation of the geodesic flow (i.e., approximating the final state at $t = 1$ under the initial conditions $\boldsymbol{x}_0 = \boldsymbol{x}$ and $\dot{\boldsymbol{x}}_0 = u_t^\theta(\boldsymbol{x})$).

While the manifold defined by the data-dependent metric $\mathbf{G}(\boldsymbol{x}, \mathcal{D})$ induces a Riemannian geometry in Euclidean space that follows an optimal cost structure, its underlying space is still Euclidean. $\mathbf{G}(\boldsymbol{x}, \mathcal{D})$ just assigns varying costs of moving in the Euclidean space $\mathbb{R}^d$. Therefore, we can avoid computing the exponential map and generate trajectories with simple Euclidean Euler integration following

$$\boldsymbol{x}_{t+\Delta t} = \boldsymbol{x}_t + \Delta t \cdot u_t^\theta(\boldsymbol{x}_t) \tag{39}$$

where $\Delta t = 1/N_{\mathrm{steps}}$ is the discretized step size.

## B COMPARISON TO EXISTING WORKS

In this section, we discuss the relationship between our proposed formulation for Branched Schrödinger Bridge Matching and related previous works. We establish reasons why BranchSBM is the theoretically optimal formulation to solve the problem of modeling stochastic dynamical systems with diverging trajectories over time by modeling branched Schrödinger bridges. We conclude by introducing an alternative perspective of the Branched GSB problem defined in Section 3.2 as the problem of modeling probabilistic trajectories of dynamic systems with nondeterministic states, which BranchSBM is well-suited to solve.

### B.1 MODELING BRANCHED SCHRÖDINGER BRIDGES

**Schrödinger Bridge Matching** Computational methods for solving the Schrödinger Bridge (SB) problem for predicting trajectories between initial and target distributions have been extensively studied in existing literature (De Bortoli et al., 2021; Chen et al., 2021b; Korotin et al., 2023; Bunne et al., 2022a; Chizat et al., 2018; Liu et al., 2023a; 2022; Shi et al., 2023; Kim et al., 2024; Wang et al., 2021; Tong et al., 2024b; Peluchetti, 2024; Bunne et al., 2022a; Somnath et al., 2023; Gushchin et al., 2024; Pavon et al., 2021; Garg et al., 2024; De Bortoli et al., 2024; Shen et al., 2024; Chen et al., 2023; Noble et al., 2023). Previous work has framed the SB problem as an entropy-regularized Optimal Control (EOT) problem (Cuturi, 2013; Léonard, 2014; Pavon et al., 2018) or a stochastic optimal control (SOC) problem (Chen et al., 2016; 2021a; Liu et al., 2023a), which we build on in this work. To ensure that the intermediate trajectories follow a path distribution that approximates that observed in the data, multimarginal Schrödinger bridge matching ensures reconstruction of multiple intermediate marginal distributions at specified time steps along the path (Shen et al., 2024; Chen et al., 2023; Theodoropoulos et al., 2025).

**Stochastic Optimal Control**    Several works (Chen et al., 2016; 2021a; Liu et al., 2023a) have reframed the SB problem as a Stochastic Optimal Control (SOC) problem, which takes the canonical form

$$\min_{u_t} \int_0^1 \mathbb{E}_{p_t} \left[ \frac{1}{2} \|u_t(X_t)\|_2^2 + V_t(X_t) \right] dt + \mathbb{E}_{p_1} \left[ \phi(X_1) \right] \tag{40}$$

$$\text{s.t. } dX_t = u_t(X_t)dt + \sigma dB_t, \quad X_0 \sim \pi_0 \tag{41}$$

where $\phi(X_1) : \mathcal{X} \to \mathbb{R}$ acts as a reconstruction loss that enforces that the distribution of $X_1 \sim p_1$ matches the true distribution $\pi_1$. Due to the intractability of $\pi_0, \pi_1$, GSBM (Liu et al., 2023a) uses spline optimization to learn an optimal Gaussian probability path $p_{t|0,1}^\star = \mathcal{N}(\mu_t, \gamma_t^2 \mathbf{I}_d)$ using only samples $\boldsymbol{x}_0 \sim \pi_0$ and $\boldsymbol{x}_1 \sim \pi_1$ from the initial and terminal distributions. Although GSBM does not require knowledge of the densities, it follows an iterative optimization scheme that alternates between matching the drift $u_t$ given a fixed marginal $p_t$, and updating the marginal given the drift. This strategy can get stuck in suboptimal solutions and is sensitive to the initialization of $u_t^\theta$. GSBM is also limited to learning unimodal Gaussian paths between one source and one target distribution with balanced mass, making it unsuitable for modeling tasks with multimodal terminal distributions and splitting of mass over multiple distinct paths.

**Regularized Unbalanced Optimal Transport**    Several previous works have studied the unbalanced optimal transport problem (Zhang et al., 2025c; Chen et al., 2021b; Lübeck et al., 2022; Pariset et al., 2023); however, these approaches address a fundamentally different setting from the unbalanced Generalized Schrödinger Bridge (GSB) problem considered in this work. Specifically, DeepRUOT (Zhang et al., 2025c) solves the Regularized Unbalanced Optimal Transport (RUOT) problem by parameterizing the canonical stochastic bridge SDE $dX_t = f_t(X_t)dt + \sigma_t dB_t$ with a probability flow ODE given by

$$dX_t = \underbrace{\left\{ f_t(X_t) + \frac{1}{2}\sigma_t^2 \nabla_{\boldsymbol{x}} \log p_t^\theta(X_t) \right\}}_{u_t^\theta(X_t)} dt \tag{42}$$

where $u_t^\theta(X_t)$ is the drift of the probability flow ODE that is learned along with the probability density $p_t^\theta(X_t)$ to derive the drift of the SDE $f_t(X_t)$. Unlike GSBM (Liu et al., 2023a), which enforces a hard constraint on the endpoints, DeepRUOT models a probability density flow by minimizing a reconstruction loss that encourages alignment with both intermediate and terminal distributions. While effective when intermediate distributions are observed, the method fails to learn meaningful trajectories in settings where these intermediate snapshots are unavailable, limiting its applicability in many real-world scenarios.

**Learning Diverging Trajectories with Single Target SBM**    To model diverging trajectories with single-branch SBM where the target distribution is multi-modal, we can follow a set of $N_{\text{samples}}$ samples from the initial distribution $\pi_0$ to determine the distribution of samples that end at each of the modes of the target distribution $\pi_1$. While this can estimate the splitting of mass across different trajectories, it does not explicitly learn the optimal distribution of mass in the latent space over time.

Furthermore, if the path toward a specific cluster in the target distribution has lower potential than that of the other clusters, mode collapse could occur, where all samples follow the same trajectory without reaching the other clusters. BranchSBM learns to generate the correct mass distribution over each of the target states by optimizing the growth term with respect to the matching loss $\mathcal{L}_{\text{match}}$ defined in Equation 15.

With the standard SBM formulation, it is also challenging to determine the time at which branching occurs and the population diverges toward different targets, as all samples follow stochastic trajectories. With BranchSBM, we model population-wide branching dynamics with growth networks that are trained to predict the origin of a branch from having zero mass ($w_{t,k} = 0$) and the rate at which it grows/shrinks over time $\partial_t w_{t,k} = g_{t,k}(X_{t,k})$. The mass of a branch at any given time step can be simulated with $w_{t,k}(X_{t,k}) = w_{0,k} + \int_0^t g_{s,k}^\phi(X_{s,k})ds$.

**Branching Dynamics**    While branched dynamics have been explored in the context of optimal transport (Lippmann et al., 2022) and Brownian motion (Baradat & Lavenant, 2021), no previous

work has explicitly formulated or solved the branched Schrödinger bridge problem that seeks to match an initial distribution to multiple terminal distributions via stochastic bridges. For instance, the concept of Branching Brownian Motion (BBM) introduced in Baradat & Lavenant (2021) models a population of particles that each independently follow stochastic trajectories according to standard Brownian motion with positive diffusivity $\nu > 0$. To model branching dynamics, each particle has an independent branching rate $\lambda > 0$ that determines the probability that the particle undergoes a *branching event*, defined as the particle dying and generating $k$ new particles that then evolve independently. The number of generated particles is sampled from a probability distribution over non-negative integers $k \sim p = (p_k)_{k \in \mathbb{N}}$, where $p_k$ is the probability of generating $k$ particles at the branching event and $\sum_{k \in \mathbb{N}} p_k = 1$. Given $p_k$, $q_k = \lambda p_k$ defines the rate of branching events that generate $k$ new particles and defines a new probability measure $q = (q_k)_{k \in \mathbb{N}}$ called the *branching mechanism*.

While BBM defines branching as the generation of additional particles from a single particle following an independent, temporal probability measure $q$, this model fails to model the *division* of mass across multiple trajectories, where total mass remains constant but the mass of each branch changes. In addition, BBM assumes that each branched particle undergoes independent Brownian motion, without explicitly defining a terminal state or branch-specific drift. For these reasons, the BBM model is unsuitable for modeling branching in the majority of real-world contexts, such as cell state transitions, where undifferentiated cells split probabilistically into distinct fates rather than proliferating in number. In such systems, branching reflects a redistribution of probability mass over developmental trajectories, governed by underlying regulatory patterns rather than purely stochastic reproduction.

Where existing frameworks fall short is in modeling **meaningful energy-aware, conditional stochastic trajectories** with **unbalanced and multi-modal dynamics**, which we address in this work. Specifically, we formulate the Unbalanced CondSOC problem followed by the Branched CondSOC problem that defines a *set* of optimal drifts and growth fields that define a set of branched trajectories following optimal energy-minimizing trajectories defined by the state cost $V_t(X_t)$. Instead of spline optimization, we leverage a parameterized network $\varphi^{\star}_{t,\eta}(\boldsymbol{x}_0, \boldsymbol{x}_1)$ that learns to predict an optimal interpolating path given a pair of endpoints.

To model dynamic growth of mass along branched trajectories, we initialize a normalized population weight of 1 at a *primary branch* ($k = 0$), that can split across $K$ branches and generate weights $\{w_{t,k}\}_{k=0}^{K}$ that evolve co-currently via learned growth rates $\{g_{t,k}^{\phi}\}_{k=0}^{K}$. This approach relies on learning the growth rate over an entire population of samples across all branches rather than learning independent growth rates of individual samples, which enforces stronger constraints during training to ensure that the model captures true population growth dynamics.

## B.2 MODELING PERTURBATION RESPONSES

In single-cell transcriptomics, perturbations such as gene knockouts, transcription factor induction, or drug treatments frequently induce cell state transitions that diverge into distinct fates, reflecting differentiation, resistance, and off-target effects (Shalem et al., 2014; Kramme et al., 2021; Dixit et al., 2016; Gavriilidis et al., 2024; Zhang et al., 2025a; Kobayashi et al., 2022; Smela et al., 2023; Pierson Smela et al., 2025; Yeo et al., 2021). Trajectory inference methods for modeling cell-state dynamics with single-cell RNA sequencing data, including flow matching (Zhang et al., 2025b; Rohbeck et al., 2025; Atanackovic et al., 2024; Tong et al., 2024b; Wang et al., 2025), Schrödinger Bridge Matching (Alatkar & Wang, 2025; Tong et al., 2024b), and optimal transport (Zhang et al., 2025c; Tong et al., 2020; Bunne et al., 2023; Driessen et al., 2025; Huguet et al., 2022; Klein et al., 2024; Yachimura et al., 2024; Schiebinger et al., 2019; Zhang et al., 2021; Bunne et al., 2022b), have been widely explored. While many algorithms have been developed to predict perturbation responses (Bunne et al., 2023; Megas et al., 2025; Rohbeck et al., 2025; Roohani et al., 2023; Ryu et al., 2024), they either predict only the terminal perturbed state without modeling the intermediate cell-state transitions or model single unimodal perturbation trajectories. Schrödinger Bridge Matching frameworks like [SF]²M (Tong et al., 2024b) and DeepGSB (Liu et al., 2022) have been shown to effectively model stochastic transitions in biological systems, offering better scalability and sampling efficiency than classical OT.

However, these models are still limited to modeling trajectories between a single pair of boundary distributions, limiting their ability to represent divergent trajectories arising from the same perturbed

initial state. This is a key limitation when modeling processes like fate bifurcation post-perturbation, where a cell population exposed to the same stimulus may split into multiple phenotypically distinct outcomes. BranchSBM extends the SBM framework to support multiple terminal marginals, enabling modeling of stochastic bifurcations in a mathematically principled way. By learning a mixture of conditional stochastic processes from a common source to multiple target distributions, BranchSBM can capture the heterogeneity and uncertainty of cell fate decisions under perturbation. Moreover, it retains the empirical tractability of previous SB-based models, requiring only samples from distributions, and ensures that intermediate trajectories lie on the manifold of feasible cell states.

### B.3 MODELING PROBABILISTIC TRAJECTORIES WITH BRANCHSBM

We conclude the discussion with an alternative interpretation of the Branched Schrödinger Bridge problem that deviates from the branching population dynamics problem. We instead consider Branched SB as a probabilistic trajectory matching problem, where each branch is one of multiple possible trajectories that a sample $X_t$ could follow. Since single-path SBM learns only a single deterministic drift field $u_t(X_t)$ that determines the direction and flow of the SDE $dX_t = u_t^\theta(X_t)dt + \sigma dB_t$, the probabilistic aspect of the trajectory remains restricted to Brownian motion via $\sigma dB_t$. This fails to capture probabilistic dynamics where probability densities begin concentrated at a single state but evolve into multi-modal probability densities $\{p_{t,k}(X_{t,k})\}_{k=0}^K$ over multiple different states, each of which evolve according to an SDE $dX_{t,k} = u_{t,k}(X_{t,k})dt + \sigma dB_t$ with an independent drift term $u_{t,k}(X_{t,k})$ and noise scaling $\sigma$.

Where other SBM frameworks fall short, BranchSBM is capable of modeling multiple probabilistic trajectories, where a system begins at a single deterministic state $X_0 = \boldsymbol{x}_0$ with probability $w_{1,0} = 1$ and evolves via multiple probabilistic trajectories that diverge in the state space governed by the SDEs $dX_{t,k} = u_{t,k}(X_{t,k})dt + \sigma dB_t$. At time $t$, the system exists in a non-deterministic superposition of states $\{X_{t,k}, w_{t,k}\}_{k=0}^K$, each with a probability $w_{t,k}$ such that $\sum_{k=0}^K w_{t,k}^\phi = 1$. In addition, BranchSBM can model the evolution of the probability weights $w_{t,k}^\phi$ by parameterizing the *probabilistic growth rates* $\{g_{t,k}^\phi\}_{k=0}^K$ that preserve conservation of probability mass $\sum_{k=0}^K w_{t,k}^\phi = 1$ at all times $t \in [0, 1]$ by minimizing the mass loss $\mathcal{L}_{\text{mass}}(\phi)$ (16). This problem is prevalent in many biological and physical systems, where a system does not exist in a single deterministic state, but rather a *superposition* over a distribution of states.

Table 6: **BranchSBM enables robust modeling of branched Schröodinger bridges compared to existing frameworks.** BranchSBM can model both branching and unbalanced trajectories, follows intermediate trajectories governed by a task-specific state cost, requires only endpoint samples for training, and samples trajectories from only a single sample from the starting distribution.

| Method | Models Branching | Models Unbalanced | Intermediate Dynamics | Requirements for Training | Requirements for Sampling |
|---|---|---|---|---|---|
| Generalized SBM Liu et al. (2023a) | No | No | Entropic OT with learned drift $u_t^\theta$ | Samples from $\pi_0, \pi_1$ | Endpoints $(\boldsymbol{x}_0, \boldsymbol{x}_1) \sim p_{0,1}^\theta$ |
| DeepRUOT (Zhang et al., 2025c) | Requires simulating multiple samples | Growth rate $g_t^\phi(X_t)$ | Regularized OT with learned drift $u_t^\theta$ and density $p_t^\theta$ | Samples from $\pi_0, \pi_1$ | Sample $\boldsymbol{x}_0 \sim \pi_0$ |
| **BranchSBM** (Ours) | Simulates divergent trajectories and terminal states from single sample $\boldsymbol{x}_0$ | Branched growth rates $\{g_{t,k}^\phi(X_t)\}_{k=0}^K$ | Branched drifts $\{u_{t,k}^\theta\}_{k=0}^K$ that minimize state-cost $V_t$ | Samples from $\pi_0$ and clusters $\{\pi_{1,k}\}_{k=0}^K$ | Sample $\boldsymbol{x}_0 \sim \pi_0$ |

## C  THEORETICAL PROOFS

### C.1  PROOF OF PROPOSITION 3.1

**Proposition 3.1** (Unbalanced Conditional Stochastic Optimal Control). *Suppose the marginal density can be decomposed as $p_t(X_t) = \int p_t(X_t|\boldsymbol{x}_0, \boldsymbol{x}_1)\pi_{0,1}(d\boldsymbol{x}_0, d\boldsymbol{x}_1)$, where $\pi_{0,1}$ is a fixed joint coupling of the data. Then, we can identify the optimal drift $u_t^\star$ and growth $g_t^\star$ that solves the Unbalanced GSB problem in (4) by minimizing the **Unbalanced Conditional Stochastic Optimal Control** objective given by*

$$\min_{u_t, g_t} \mathbb{E}_{(\boldsymbol{x}_0, \boldsymbol{x}_1)\sim\pi_{0,1}} \left[ \int_0^1 \mathbb{E}_{p_{t|0,1}} \left[ \left( \frac{1}{2}\|u_t(X_t|\boldsymbol{x}_0, \boldsymbol{x}_1)\|_2^2 + V_t(X_t) \right) w_t(X_t) \right] dt \right] \quad (5)$$

$$\text{s.t.} \ dX_t = u_t(X_t|\boldsymbol{x}_0, \boldsymbol{x}_1)dt + \sigma dB_t, \ X_0 = \boldsymbol{x}_0, \ X_1 = \boldsymbol{x}_1 \ w_0 = w_0^\star, \ w_1 = w_1^\star \quad (6)$$

*where $w_t = w_0 + \int_0^t g_s(X_s)ds$ is the time-dependent weight initialized at $w_0^\star$, $u_t$ is the drift, $g_t$ is the growth rate, and $\pi_{0,1}$ is the weighted coupling of paired endpoints $(\boldsymbol{x}_0, w_0^\star, \boldsymbol{x}_1, w_1^\star) \sim \pi_{0,1}$.*

*Proof.* We define the Unbalanced Generalized Schrödinger Bridge problem as the solution $(u_t, p_t, g_t)$ to the energy minimization problem such that the unbalanced Fokker-Planck equation is satisfied.

$$\min_{u_t, g_t} \int_0^1 \mathbb{E}_{p_t} \left[ \frac{1}{2}\|u_t(X_t)\|_2^2 + V_t(X_t) \right] \left( w_0 + \int_0^t g_s(X_s)ds \right) dt \quad (43)$$

$$\text{s.t.} \begin{cases} \frac{\partial}{\partial t} p_t(X_t) = -\nabla \cdot (u_t(X_t)p_t(X_t)) + \frac{\sigma^2}{2}\Delta p_t(X_t) + g_t(X_t)p_t(X_t) \\ p_0 = \pi_0, \ p_1 = \pi_1 \end{cases} \quad (44)$$

Under the assumption that the joint probability $\pi_{0,1}(\boldsymbol{x}_0, \boldsymbol{x}_1)$ is fixed over all times $t \in [0, 1]$ and the marginal probability can be factorized as $p_t(X_t) = \mathbb{E}_{\pi_{0,1}} \left[ p_{t|0,1}(X_t|\boldsymbol{x}_0, \boldsymbol{x}_1) \right]$, we can decompose the minimization objective into

$$\int_0^1 \mathbb{E}_{p_t} \left[ \frac{1}{2}\|u_t(X_t)\|_2^2 + V_t(X_t) \right] \left( w_0 + \int_0^t g_s(X_s)ds \right) dt \quad (45)$$

$$= \int_0^1 \mathbb{E}_{p_{0,1}} \mathbb{E}_{p_{t|0,1}} \left[ \frac{1}{2}\|u_t(X_t)\|_2^2 + V_t(X_t) \right] \left( w_0 + \int_0^t g_s(X_s)ds \right) dt \quad \text{(law of total expectation)}$$

$$= \mathbb{E}_{p_{0,1}} \int_0^1 \mathbb{E}_{p_{t|0,1}} \left[ \frac{1}{2}\|u_t(X_t)\|_2^2 + V_t(X_t) \right] \left( w_0 + \int_0^t g_s(X_s)ds \right) dt \quad \text{(Fubini's theorem)}$$

which can be solved by identifying the conditional drift $u_t$ that minimizes the expected objective value over all endpoint samples $(\boldsymbol{x}_0, \boldsymbol{x}_1) \sim \pi_{0,1}$. Under similar assumptions, we can decompose all terms in the Fokker-Planck equation. For the left-hand side, we have:

$$\frac{\partial}{\partial t} p_t(X_t) = \frac{\partial}{\partial t} \int p_{t|0,1}(X_t|\boldsymbol{x}_0, \boldsymbol{x}_1)\pi_{0,1}(d\boldsymbol{x}_0, d\boldsymbol{x}_1) = \mathbb{E}_{(\boldsymbol{x}_0, \boldsymbol{x}_1)\sim\pi_{0,1}} \left[ \frac{\partial}{\partial t} p_{t|0,1}(X_t|\boldsymbol{x}_0, \boldsymbol{x}_1) \right] \quad (46)$$

For the divergence term, we have:

$$\nabla \cdot (u_t(X_t)p_t(X_t)) = \nabla \cdot \left( u_t(X_t) \int p_{t|0,1}(X_t|\boldsymbol{x}_0, \boldsymbol{x}_1)\pi_{0,1}(d\boldsymbol{x}_0, d\boldsymbol{x}_1) \right)$$

$$= \nabla \cdot \left( \int u_t(X_t)p_{t|0,1}(X_t|\boldsymbol{x}_0, \boldsymbol{x}_1)\pi_{0,1}(d\boldsymbol{x}_0, d\boldsymbol{x}_1) \right) \quad (47)$$

By Fubini's Theorem and the linearity of the divergence operator, we can switch the order of integration to get

$$\nabla \cdot (u_t(X_t)p_t(X_t)) = \int \nabla \cdot \left( u_t(X_t)p_{t|0,1}(X_t|\boldsymbol{x}_0, \boldsymbol{x}_1) \right)\pi_{0,1}(d\boldsymbol{x}_0, d\boldsymbol{x}_1)$$

$$= \mathbb{E}_{(\boldsymbol{x}_0, \boldsymbol{x}_1)\sim\pi_{0,1}} \left[ \nabla \cdot \left( u_t(X_t)p_{t|0,1}(X_t|\boldsymbol{x}_0, \boldsymbol{x}_1) \right) \right] \quad (48)$$

For the Laplacian term, we have:

$$
\begin{aligned}
\frac{\sigma^2}{2}\Delta p_t(X_t) &= \frac{\sigma^2}{2}\nabla \cdot (\nabla p_t(X_t)) \\
&= \frac{\sigma^2}{2}\nabla \cdot \left(\nabla \int p_{t|0,1}(X_t|\boldsymbol{x}_0,\boldsymbol{x}_1)\pi_{0,1}(d\boldsymbol{x}_0,d\boldsymbol{x}_1)\right) \\
&= \frac{\sigma^2}{2}\int \left(\nabla \cdot \nabla p_{t|0,1}(X_t|\boldsymbol{x}_0,\boldsymbol{x}_1)\right)\pi_{0,1}(d\boldsymbol{x}_0,d\boldsymbol{x}_1) \\
&= \frac{\sigma^2}{2}\mathbb{E}_{(\boldsymbol{x}_0,\boldsymbol{x}_1)\sim\pi_{0,1}}\left[\Delta p_{t|0,1}(X_t|\boldsymbol{x}_0,\boldsymbol{x}_1)\right]
\end{aligned}
\tag{49}
$$

For the growth term, we have:

$$
\begin{aligned}
g_t(X_t)p_t(X_t) &= g_t(X_t)\int p_{t|0,1}(X_t|\boldsymbol{x}_0,\boldsymbol{x}_1)\pi_{0,1}(d\boldsymbol{x}_0,d\boldsymbol{x}_1) \\
&= \int g_t(X_t)p_{t|0,1}(X_t|\boldsymbol{x}_0,\boldsymbol{x}_1)\pi_{0,1}(d\boldsymbol{x}_0,d\boldsymbol{x}_1) \\
&= \mathbb{E}_{(\boldsymbol{x}_0,\boldsymbol{x}_1)\sim\pi_{0,1}}\left[g_t(X_t)p_{t|0,1}(X_t|\boldsymbol{x}_0,\boldsymbol{x}_1)\right]
\end{aligned}
\tag{50}
$$

Combining all the terms of the Fokker-Planck equation, we have shown that (44) can be rewritten as

$$
\frac{\partial}{\partial t}p_{t|0,1}(X_t|\boldsymbol{x}_0,\boldsymbol{x}_1) = -\nabla \cdot \left(u_t p_{t|0,1}(X_t|\boldsymbol{x}_0,\boldsymbol{x}_1)\right) + \Delta p_{t|0,1}(X_t|\boldsymbol{x}_0,\boldsymbol{x}_1) + g_t p_{t|0,1}(X_t|\boldsymbol{x}_0,\boldsymbol{x}_1)
\tag{51}
$$

Therefore, the Unbalanced GSB problem is equivalent to solving the Unbalanced CondSOC problem, and we conclude our proof of Proposition 3.1. □

## C.2 PROOF OF PROPOSITION 3.2

**Proposition 3.2** (Branched Conditional Stochastic Optimal Control). *For each branch, let $p_{t,k}(X_{t,k}) = \int p_{t,k}(X_{t,k}|\boldsymbol{x}_0,\boldsymbol{x}_{1,k})\pi_{0,1,k}(d\boldsymbol{x}_0,d\boldsymbol{x}_{1,k})$, where $\pi_{0,1,k}$ is the joint coupling distribution of samples $\boldsymbol{x}_0 \sim \pi_0$ from the initial distribution and $\boldsymbol{x}_{1,k} \sim \pi_{1,k}$ from the kth target distribution. Then, we can identify the set of optimal drift and growth functions $\{u_{t,k}^\star, g_{t,k}^\star\}_{k=0}^K$ that solve the Branched GSB problem in (7) by minimizing the **sum of Unbalanced CondSOC objectives** given by*

$$
\min_{\{u_{t,k},g_{t,k}\}_{k=0}^K}\mathbb{E}_{(\boldsymbol{x}_0,\boldsymbol{x}_{1,0})\sim\pi_{0,1,0}}\int_0^1\left\{\mathbb{E}_{p_{t|0,1,0}}\left[\frac{1}{2}\|u_{t,0}(X_{t,0})\|_2^2 + V_t(X_{t,0})\right]w_{t,0}\right\}dt
$$

$$
+ \sum_{k=1}^K\mathbb{E}_{(\boldsymbol{x}_0,\boldsymbol{x}_{1,k})\sim\pi_{0,1,k}}\int_0^1\left\{\mathbb{E}_{p_{t|0,1,k}}\left[\frac{1}{2}\|u_{t,k}(X_{t,k})\|_2^2 + V_t(X_{t,k})\right]w_{t,k}\right\}dt
\tag{8}
$$

$$
s.t. \quad dX_{t,k} = u_{t,k}(X_{t,k})dt + \sigma dB_t, \ X_0 = \boldsymbol{x}_0, \ X_{1,k} = \boldsymbol{x}_{1,k}, \ w_{0,k} = \delta_{k=0}, \ w_{1,k} = w_{1,k}^\star
\tag{9}
$$

*where $w_{t,0} = 1 + \int_0^t g_{s,0}(X_{s,0})ds$ is the weight of the primary paths initialized at 1 and $w_{t,k} = \int_0^t g_{s,k}(X_{s,k})ds$ are the weights of the K secondary branches initialized at 0.*

*Proof.* We extend the proof of Proposition 3.1 to the branching case by defining each branch $k$ as an independent Unbalanced Generalized Schrödinger Bridge problem in (4) given by

$$
\min_{u_{t,k},g_{t,k}}\int_0^1\mathbb{E}_{p_{t,k}}\left[\frac{1}{2}\|u_{t,k}(X_{t,k})\|_2^2 + V_t(X_{t,k})\right]\left(w_{0,k} + \int_0^t g_{s,k}(X_{s,k})ds\right)dt
\tag{52}
$$

$$
s.t. \begin{cases} \frac{\partial}{\partial t}p_{t,k}(X_{t,k}) = -\nabla \cdot (u_{t,k}(X_{t,k})p_{t,k}(X_{t,k})) + \frac{\sigma^2}{2}\Delta p_{t,k}(X_{t,k}) + g_{t,k}(X_{t,k})p_{t,k}(X_{t,k}) \\ p_0 = \pi_0, \ p_{1,k} = \pi_{1,k} \quad \forall k \in \{0,\dots,K\} \end{cases}
\tag{53}
$$

such that each branch independently solves the Fokker-Planck equation defined as $\frac{\partial}{\partial t}p_{t,k} = -\nabla \cdot (u_{t,k}p_{t,k}) + \frac{\sigma^2}{2}\Delta p_{t,k}$. Now, we show that the sum of unbalanced CondSOC problems still satisfies

the global Fokker-Planck equation

$$\frac{\partial}{\partial t} p_t = -\nabla \cdot (u_t p_t) + \frac{\sigma^2}{2} \Delta p_t \tag{54}$$

where we define $p_t$ as the weighted sum of the branched distributions given by

$$p_t(X_t) = \sum_{k=0}^{K} w_{t,k} p_{t,k}(X_{t,k}) \tag{55}$$

To obtain an expression for the global Fokker-Planck equation, we differentiate both sides and substitute the branched FPE as follows

$$\frac{\partial}{\partial t} p_t = \frac{\partial}{\partial t} \left[ \sum_{k=0}^{K} w_{t,k} p_{t,k} \right]$$

$$\frac{\partial}{\partial t} p_t = \sum_{k=0}^{K} \left[ w_{t,k} \left( \frac{\partial}{\partial t} p_{t,k} \right) + \left( \frac{\partial}{\partial t} w_{t,k} \right) p_{t,k} \right]$$

$$\frac{\partial}{\partial t} p_t = \sum_{k=0}^{K} \left[ w_{t,k} \left( -\nabla \cdot (u_{t,k} p_{t,k}) + \frac{\sigma^2}{2} \Delta p_{t,k} \right) + g_{t,k} p_{t,k} \right]$$

$$\text{(substitute branched FPE and } \partial_t w_{t,k} = g_{t,k})$$

$$\frac{\partial}{\partial t} p_t = \sum_{k=0}^{K} \left[ -w_{t,k} \nabla \cdot (u_{t,k} p_{t,k}) + w_{t,k} \frac{\sigma^2}{2} \Delta p_{t,k} + g_{t,k} p_{t,k} \right]$$

$$\frac{\partial}{\partial t} p_t = \sum_{k=0}^{K} (-w_{t,k} \nabla \cdot (u_{t,k} p_{t,k})) + \sum_{k=0}^{K} w_{t,k} \frac{\sigma^2}{2} \Delta p_{t,k} + \sum_{k=0}^{K} g_{t,k} p_{t,k} \tag{56}$$

By linearity of the Laplacian, the diffusion term can be rewritten as

$$\sum_{k=0}^{K} w_{t,k} \frac{\sigma^2}{2} \Delta p_{t,k} = \frac{\sigma^2}{2} \Delta \underbrace{\left( \sum_{k=0}^{K} w_{t,k} p_{t,k} \right)}_{p_t} = \frac{\sigma^2}{2} \Delta p_t \tag{57}$$

Since the global growth term $\sum_{k=0}^{K} g_{t,k} p_{t,k}$ is the sum of the growth dynamics across all branches and is separate from the drift and diffusion dynamics, it doesn't alter the direction or motion of particles along the branched fields. Thus, both the diffusion and growth terms satisfy the global FPE.

Now, we set the divergence term in (56) equal to the global divergence $\nabla \cdot (u_t p_t)$ and derive the expression for the total drift field $u_t(X_t)$ that satisfies the global FPE. By the linearity of the divergence operator, we get

$$-\nabla \cdot (u_t p_t) = \sum_{k=0}^{K} (-w_{t,k} \nabla \cdot (u_{t,k} p_{t,k}))$$

$$-\nabla \cdot (u_t p_t) = -\nabla \cdot \left( \sum_{k=0}^{K} w_{t,k} u_{t,k} p_{t,k} \right)$$

$$-\nabla \cdot (u_t p_t) = -\nabla \cdot \underbrace{\left( \frac{1}{p_t} \sum_{k=0}^{K} w_{t,k} u_{t,k} p_{t,k} \right)}_{u_t} p_t \tag{58}$$

Under the global FPE constraint, the drift $u_t$ is defined as the mass-weighted average of the drift fields for each branch, given by $u_t(X_t) = \frac{1}{p_t(X_t)} \sum_{k=0}^{K} w_{t,k}(X_{t,k}) u_{t,k}(X_{t,k}) p_{t,k}(X_{t,k})$. Intuitively, this means that in the global context, for any $X_t = x_t$, the drift of state $X_t$ along the dynamics of branch $k$ is scaled by the weight of $x_t$ at time $t$ along branch $k$ and the ratio of probability density of $x_t$ under branch $k$ over the total probability density of $x_t$ across all branches. Therefore, our definition of the weighted drift decoupled over individual branches satisfies the global FPE equation in (54), and this concludes the proof of Proposition 3.2. □

**Remark C.1** (Reduction to Single Path GSBM). *When $g_{t,0}(X_{t,0}) = 0$ and $g_{t,k}(X_{t,k}) = 0$ for all $t \in [0,1]$ and $k \in \{1, \ldots, K\}$, then the Branched CondSOC problem is the solution to the single path GSB problem given by*

$$\min_{u_t} \int_0^1 \mathbb{E}_{p_{t|0,1}} \left[ \frac{1}{2} \|u_t(X_t)\|_2^2 + V_t(X_t) \right] dt \ \ s.t. \ \ \begin{cases} dX_t = u_t(X_t)dt + \sigma dB_t \\ X_0 \sim \pi_0, \ \ X_1 \sim \pi_1 \end{cases} \tag{59}$$

*where the probability density $p_{t|0,1}(X_t|\boldsymbol{x}_0, \boldsymbol{x}_1)$ is conditioned explicitly on a pair of endpoints $(\boldsymbol{x}_0, \boldsymbol{x}_1) \sim \pi_{0,1}$ drawn from the joint distribution.*

## C.3 PROOF OF PROPOSITION 4.1

**Proposition 4.1** (Solving the GSB Problem with Stage 1 and 2 Training). *Stage 1 and Stage 2 training yield the optimal drift $u_t^\star(X_t)$ that generates the optimal marginal probability distribution $p_t^\star(X_t)$ that solves the GSB problem in (3).*

*Proof.* Let the marginal probability distribution $p_t^\star(X_t)$ and corresponding drift $u_t^\star(X_t)$ define the optimal solution to the GSB problem. It suffices to show that

1. Given $(\boldsymbol{x}_0, \boldsymbol{x}_1) \sim \pi_{0,1}$, Stage 1 training with the trajectory loss $\mathcal{L}_{\text{traj}}(\eta)$ (12) yields the interpolant $\boldsymbol{x}_{t,\eta}^\star$ and time-derivative $\dot{\boldsymbol{x}}_{t,\eta}^\star$ that define $p_t^\star(X_t)$.

2. Given $p_t^\star(X_t)$, Stage 2 training with the explicit flow matching loss $\mathcal{L}_{\text{flow}}(\theta)$ (13) yields the optimal drift $u_t^\star(X_t)$.

To prove Part 1, we establish the following Lemma.

**Lemma C.1.** *Given the Markovian reference process $\mathbb{Q}$ with drift $v_t^\star(X_t)$ from the unconstrained GSB problem in (3), define the reciprocal projection $\Pi^\star = proj_{\mathcal{R}(\mathbb{Q})}\{\mathbb{P} : \mathbb{P}_{0,1} = \pi_{0,1}\}$ of the endpoint coupling $\pi_{0,1}$ onto the reciprocal class $\mathcal{R}(\mathbb{Q})$. Stage 1 training then learns the velocity field $\dot{\boldsymbol{x}}_{t,\eta}^\star$ whose induced interpolant matches the intermediate time dynamics of $\Pi^\star$ from endpoint samples $(\boldsymbol{x}_0, \boldsymbol{x}_1) \sim \pi_{0,1}$.*

*Proof of Lemma.* Since $\mathbb{Q} \in C([0,1]; \mathbb{R}^d)$ is a Markov measure with well-defined bridges $\mathbb{Q}_{\cdot|0,1}(\cdot|X_0 = \boldsymbol{x}_0, X_1 = \boldsymbol{x}_1)$, we define the reciprocal projection $\Pi^\star$ with the endpoint coupling $\pi_{0,1}$ as:

$$\Pi^\star = \int \mathbb{Q}_{\cdot|0,1}(\cdot|X_0 = \boldsymbol{x}_0, X_1 = \boldsymbol{x}_1)\pi_{0,1}(d\boldsymbol{x}_0, d\boldsymbol{x}_1) \tag{60}$$

which is in the reciprocal class $\Pi^\star \in \mathcal{R}(\mathbb{Q})$, satisfies the endpoint marginals $\Pi_{0,1}^\star = \pi_{0,1}$, and is the unique minimizer of the KL divergence:

$$\Pi^\star = \underset{\mathbb{P} \in \mathcal{R}(\mathbb{Q}); \mathbb{P}_{0,1} = \pi_{0,1}}{\arg\min} \ \text{KL}(\mathbb{P}\|\mathbb{Q}) \tag{61}$$

Therefore, it suffices to prove that $\dot{\boldsymbol{x}}_{t,\eta}^\star$ approximates $v_t^\star(X_t)$ under endpoint constraints $X_0 = \boldsymbol{x}_0$ and $X_1 = \boldsymbol{x}_1$ for $(\boldsymbol{x}_0, \boldsymbol{x}_1) \sim \pi_{0,1}$.

First, we recall that the unconstrained Markovian drift $v_t^\star$ is the minimizer of the energy function given by:

$$v_t^\star = \underset{v_t}{\arg\min} \int_0^1 \mathbb{E}_{p_t} \left[ \frac{1}{2} \|v_t(X_t)\|_2^2 + V_t(X_t) \right] dt \tag{62}$$

where $p_t$ is the marginal law induced by $v_t$. Furthermore, the class of interpolants is given by parameters $\eta$ is defined as:

$$\boldsymbol{x}_{t,\eta} = (1-t)\boldsymbol{x}_0 + t\boldsymbol{x}_1 + t(1-t)\varphi_{t,\eta}(\boldsymbol{x}_0, \boldsymbol{x}_1) \tag{63}$$

$$\dot{\boldsymbol{x}}_{t,\eta} = \boldsymbol{x}_1 - \boldsymbol{x}_0 + t(1-t)\dot{\varphi}_{t,\eta}(\boldsymbol{x}_0, \boldsymbol{x}_1) + (1-2t)\varphi_{t,\eta}(\boldsymbol{x}_0, \boldsymbol{x}_1) \tag{64}$$

Stage 1 training yields the optimal interpolant $\dot{\boldsymbol{x}}_{t,\eta}^\star$ that minimizes the energy function across all time points $t \in [0,1]$, defined as

$$\dot{\boldsymbol{x}}_{t,\eta}^\star = \underset{\dot{\boldsymbol{x}}_{t,\eta}}{\arg\min} \int_0^1 \mathbb{E}_{p_t} \left[ \frac{1}{2} \|\dot{\boldsymbol{x}}_{t,\eta}(\boldsymbol{x}_0, \boldsymbol{x}_1)\|_2^2 + V_t(\boldsymbol{x}_{t,\eta}) \right] dt \tag{65}$$

Therefore, $\dot{\boldsymbol{x}}_{t,\eta}^\star$ matches the reference drift $v_t^\star$ defined in (62) along interpolated states while preserving $(\boldsymbol{x}_0, \boldsymbol{x}_1) \sim \pi_{0,1}$. This is an approximation of the bridge dynamics induced by the reciprocal projection $\Pi^\star = \text{proj}_{\mathcal{R}(\mathbb{Q})}\{\mathbb{P} : \mathbb{P}_{0,1} = \pi_{0,1}\}$, which concludes the proof of Lemma C.1. $\qquad\square$

By Lemma C.1, we know that $v_t^\star$ is the optimal drift energy function in the GSB problem without endpoint constraints and $\dot{\boldsymbol{x}}_{t,\eta}^\star$ is the approximation of the bridge dynamics induced by $\Pi^\star$ which follows $v_t^\star$ while preserving the coupling $\pi_{0,1}$. Therefore, we can define $u_{t|0,1}^\star(X_t|\boldsymbol{x}_0, \boldsymbol{x}_1) \equiv \dot{\boldsymbol{x}}_{t,\eta}^\star(\boldsymbol{x}_0, \boldsymbol{x}_1)$ as the conditional drift that generates the conditional probability distribution $p_{t|0,1}^\star(X_t|\boldsymbol{x}_0, \boldsymbol{x}_1)$ that satisfies the Fokker-Planck equation

$$\frac{\partial}{\partial t} p_{t|0,1} = -\nabla \cdot (u_{t|0,1} p_{t|0,1}) + \frac{1}{2}\sigma^2 \Delta p_{t|0,1} \tag{66}$$

Given that $p_{t|0,1}^\star(X_t|\boldsymbol{x}_0, \boldsymbol{x}_1)$ is the optimal bridge that solves the GSB problem for a pair of endpoints $(\boldsymbol{x}_0, \boldsymbol{x}_1) \sim \pi_{0,1}$, we can define the marginal probability distribution $p_t^\star$ as the mixture of bridges

$$p_t^\star = \mathbb{E}_{(\boldsymbol{x}_0, \boldsymbol{x}_1) \sim \pi_{0,1}} \left[ p_{t|0,1}^\star(X_t|\boldsymbol{x}_0, \boldsymbol{x}_1) \right] \tag{67}$$

which concludes the proof of Part 1.

For Part 2 of the proof, we aim to show that Stage 2 training yields the optimal Markovian drift $u_t^\star(X_t)$ that *generates* $p_t^\star(X_t)$. To do this, we write the Fokker-Planck equation for $p_t^\star(X_t)$ in terms of the conditional bridge $p_{t|0,1}$ and drift field $u_{t|0,1}$ to extract an expression for $u_t^\star(X_t)$ that satisfies it. Starting from the definition of $p_t^\star$, we have

$$p_t^\star = \int p_{t|0,1}^\star d\pi_{0,1}$$

$$\frac{\partial}{\partial t} p_t^\star = \int \left( \frac{\partial}{\partial t} p_{t|0,1}^\star \right) d\pi_{0,1}$$

$$= \int \left( -\nabla \cdot (u_{t|0,1} p_{t|0,1}^\star) + \frac{1}{2}\sigma^2 \Delta p_{t|0,1}^\star \right) d\pi_{0,1}$$

$$= \int \left( -\nabla \cdot (u_{t|0,1} p_{t|0,1}^\star) \right) d\pi_{0,1} + \int \left( \frac{1}{2}\sigma^2 \Delta p_{t|0,1}^\star \right) d\pi_{0,1}$$

$$= -\nabla \cdot \int (u_{t|0,1} p_{t|0,1}^\star) d\pi_{0,1} + \frac{1}{2}\sigma^2 \Delta \int p_{t|0,1}^\star d\pi_{0,1}$$

$$= -\nabla \cdot \underbrace{\int (u_{t|0,1} p_{t|0,1}^\star) d\pi_{0,1}}_{(u_t^\star p_t^\star)} + \frac{1}{2}\sigma^2 \Delta \underbrace{\int p_{t|0,1}^\star d\pi_{0,1}}_{p_t^\star} \tag{68}$$

For (68) to satisfy the Fokker-Planck equation, we set the first term equal to $(u_t^\star p_t^\star)$ and solve for $u_t^\star$ to get

$$u_t^\star p_t^\star = \int (u_{t|0,1}^\star p_{t|0,1}^\star) d\pi_{0,1} \tag{69}$$

$$u_t^\star = \frac{\mathbb{E}_{\pi_{0,1}} \left[ u_{t|0,1}^\star p_{t|0,1}^\star \right]}{p_t^\star} \tag{70}$$

Therefore, the optimal Markovian drift (or *Markovian projection*) is the *average* of the conditional drifts defined in part 1 as $u_{t|0,1}^\star \equiv \dot{\boldsymbol{x}}_{t,\eta}^\star$ over the joint distribution $\pi_{0,1}$. This means that the minimizer of the conditional flow matching loss $\mathcal{L}_{\text{flow}}(\theta)$ in (13) defined as the expected mean-squared error

between a Markovian drift field $u_t(X_t)$ and $\dot{\boldsymbol{x}}_{t,\eta}^\star$ over pairs $(\boldsymbol{x}_0, \boldsymbol{x}_1) \sim \pi_{0,1}$ in the dataset is the optimal drift $u_t^\star(X_t)$ that solves the GSB problem.

$$u_t^\star(X_t) = \arg\min_\theta \mathcal{L}_{\text{flow}}(\theta) = \arg\min_{u_t^\theta} \int_0^1 \mathbb{E}_{(\boldsymbol{x}_0,\boldsymbol{x}_1)\sim\pi_{0,1}}\|\dot{\boldsymbol{x}}_{t,\eta}^\star - u_t^\theta(X_t)\|_2^2 dt \tag{71}$$

which concludes the proof of Proposition 4.1. $\qquad\square$

Since the drift for each branch $u_{t,k}^\theta(X_{t,k})$ are trained independently in Stage 2, we can extend this result across all $K + 1$ branches and conclude that the sequential Stage 1 and Stage 2 training procedures yields the optimal set of drifts $\{u_{t,k}^\star\}_{k=0}^K$ that *generate* the optimal probability paths $\{p_{t,k}^\star\}_{k=0}^K$ that solves the GSB problem for each branch.

### C.4 PROOF OF PROPOSITION 4.2

**Lemma C.2.** *Suppose the optimal drift field $u_{t,k}^\star : \mathbb{R}^d \to \mathbb{R}^d$ and probability density $p_{t,k}^\star : \mathbb{R}^d \to \mathbb{R}$ that minimizes the GSB problem in (3) is well-defined over the state space $\mathcal{X} \subseteq \mathbb{R}^d$ for each branch. Then, the optimal weight $w_{t,k}^\star$ at any of the secondary branches $k \in \{1, \ldots, K\}$ is non-decreasing over the interval $t \in [0, 1]$. Equivalently, the optimal growth rates $g_{t,k}^\star(X_{t,k}) \geq 0$ for all $t \in [0, 1]$.*

*Proof.* We will prove this lemma by contradiction. Suppose there exists a branch $k$ that decreases in weight over the time interval $[t_1, t_2]$ for $0 \leq t_1 < t_2 \leq 1$, such that $w_{t_1,k} > w_{t_2,k}$. We know that the target weight at time $t = 1$ is non-negative $w_{1,k} \geq 0$ and the total weight across all branches is conserved (i.e. $\sum_{k=0}^K w_{t,k} = w_t^{\text{total}}$) and non-decreasing (i.e. $\partial_t w_t^{\text{total}} > 0$ for all $t \in [0, 1]$). In this proof, we let $\mathcal{E}_k(t_1, t_2) = \int_{t_1}^{t_2} \left[\frac{1}{2}\|u_{t,k}^\star(X_{t,k})\|_2^2 + V_t(X_{t,k})\right]$ denote the energy of following the dynamics of the $k$th branch over the time interval $[t_1, t_2]$.

Then, there can only be two possible reasons for the loss of mass along a branch $k$: **(1)** the mass is destroyed, or **(2)** the mass is transferred to a different branch $j \neq k$.

**Case 1.** The destruction of mass directly violates the assumption that the total mass across all branches is conserved and non-decreasing. So, we only need to consider the possibility of Case 2.

**Case 2.** Suppose the mass is transferred to a different branch $j \neq k$ over the interval $[t_1, t_2]$. By Proposition 4.1, Stage 1 and 2 training yields the optimal velocity fields $\{u_{t,k}^\star(X_{t,k})\}_{k=0}^K$ that generate the optimal interpolating probability density $\{p_{t,k}^\star(X_{t,k})\}_{k=0}^K$ that independently minimize the GSB problem in (3).

If mass is transferred from branch $k$ to branch $j$ over the interval $[t_1, t_2]$, it must be compensated for over time $[t_2, 1]$ to reach the target weight $w_{1,k} > 0$. Then, without loss of generality, we can consider two sub-cases: (2.1) the mass is compensated from the primary branch, and (2.2) the mass that diverges to the $j$th branch returns to the $k$th branch following a continuous trajectory.

*Case 2.1.* Since all mass along the $K$ secondary branches originates from the primary branch, this implies that there exists a positive weight $\tilde{w} > 0$ that first follows the dynamics of branch $k$ and is transferred to branch $j$, contributing to the final weight of the $j$th endpoint, and the total weight supplied from the primary branch to branch $k$ is $w_{1,k} + \tilde{w}$. Given that each branch has no capacity constraints, it follows that the dynamics along branch $k$ over $[0, t_1]$ and branch $j$ over $[t_1, 1]$ are optimal for all mass reaching endpoint $j$. Formally, we express this in terms of energy as

$$\mathcal{E}_k(0, t_1) + \mathcal{E}_j(t_1, 1) < \mathcal{E}_j(0, 1) \tag{72}$$

which contradicts the assumption that the dynamics of branch $j$ given by $(u_{t,j}^\star, p_{t,j}^\star)$ are optimal over the state space $\mathcal{X}$.

*Case 2.2.* In this case, there exists a positive mass that follows the dynamics of branch $k$ over the interval $[0, t_1]$, the dynamics of branch $j$ over $[t_1, t_2]$, and back to branch $k$ over $[t_2, 1]$. Similarly to Case 2.1, given that each branch has no capacity constraints, this implies that this concatenation of

dynamics is optimal for all mass reaching endpoint $k$. Expressing in terms of energy, we have

$$\mathcal{E}_k(0, t_1) + \mathcal{E}_j(t_1, t_2) + \mathcal{E}_k(t_2, 1) < \mathcal{E}_k(0, 1) \tag{73}$$

which contradicts the assumption that the dynamics of branch $k$ given by $(u_{t,k}^\star, p_{t,k}^\star)$ are optimal over the state space $\mathcal{X}$.

Given that both sub-cases lead to a contradiction of the optimality assumption, we conclude that mass along each branch cannot be transferred to another branch and is non-decreasing over $t \in [0, 1]$.

Note that we do not need to consider the case where the mass is compensated from another secondary branch $\ell \neq j$, as this would imply that mass is transferred from branch $\ell$ to branch $k$, which is not possible under the same argument and we conclude our proof of Proposition 4.2. $\qquad\square$

---

**Proposition 4.2** (Existence of Optimal Growth Functions). *Assume the state space $\mathcal{X} \subseteq \mathbb{R}^d$ is a bounded domain within $\mathbb{R}^d$. Let the optimal probability density of branch $k$ be a known non-negative function bounded in $[0, 1]$, denoted as $p_{t,k}^\star : \mathcal{X} \times [0, 1] \to [0, 1] \in L^\infty(\mathcal{X} \times [0, 1])$. By Lemma C.2, we can define the set of feasible growth functions in the set of square-integrable functions $L^2$ as*

$$\mathcal{G} := \{g = (g_{t,0}, \dots, g_{t,K}) \in L^2(\mathcal{X} \times [0, 1]; \mathbb{R}^{K+1}) \mid \forall k \geq 1, \ g_{t,k}(\boldsymbol{x}) \geq 0, \textstyle\sum_{k=0}^K g_{t,k} = 0\} \tag{18}$$

*Let the growth loss be the functional $\mathcal{L}(g) : L^2(\mathcal{X} \times [0, 1]; \mathbb{R}^{K+1}) \to \mathbb{R}$. Then, there exists an optimal function $g^\star = (g_{t,0}^\star, \dots, g_{t,K}^\star) \in L^2(\mathcal{X} \times [0, 1]; \mathbb{R}^{K+1})$ where $g^\star \in \mathcal{G}$ such that $\mathcal{L}(g^\star) = \inf_{g \in \mathcal{G}} \mathcal{L}(g)$ which can be obtained by minimizing $\mathcal{L}(g)$ over $\mathcal{G}$.*

---

*Proof.* This proof draws on several concepts from functional and convex analysis. For a comprehensive background on these concepts, see Benešová & Kružík (2016). We follow the *direct method in the calculus of variations* (Dacorogna, 1989) for proving there exists a minimizer for the functional $\mathcal{L}_{\text{growth}}(g)$ in (17) with the following steps:

1. Show that the set $\mathcal{G}$ of *feasible* growth functions is convex and weakly closed under the set of square integrable functions $L^2(\Omega; \mathbb{R}^{K+1})$ (Lemma C.3).

2. Show that the minimizing sequence $\{g^{(n)}\}$ has a weakly convergent subsequence (Lemma C.4).

3. Show that the functional $\mathcal{L}_{\text{growth}}(g)$ is weakly lower semi-continuous (Lemmas C.5, C.6, C.7, C.8).

We prove each with a sequence of Lemmas.

---

**Lemma C.3.** *The set of feasible growth functions $\mathcal{G} := \{g = (g_{t,0}, \dots, g_{t,K}) \in L^2(\mathcal{X} \times [0, 1]; \mathbb{R}^{K+1}) \mid \forall k \geq 1, \ g_{t,k}(\boldsymbol{x}) \geq 0 \text{ and } \sum_{k=0}^K g_{t,k} = 0\}$ is convex and weakly closed under $L^2(\mathcal{X} \times [0, 1]; \mathbb{R}^{K+1})$.*

---

*Proof.* We first prove convexity and then weak closure under $L^2(\mathcal{X} \times [0, 1]; \mathbb{R}^{K+1})$.

*Proof of Convexity.* To prove that the set of functions $\mathcal{G}$ is convex, we first define what it means for a set of functions to be convex.

---

**Definition C.1** (Convex Set). *A set of functions $\mathcal{G}$ is said to be convex if any convex combination of two functions in the set $g_1, g_2 \in \mathcal{G}$ is also in the set. Formally, $\mathcal{G}$ if*

$$\forall g, g' \in \mathcal{G}, \quad \forall \lambda \in [0, 1], \quad g^\lambda = \lambda g + (1 - \lambda)g' \in \mathcal{G} \tag{74}$$

---

Recall our definition of $\mathcal{G}$ as the set of $K + 1$ growth rate functions that sum to zero:

$$\mathcal{G} := \{g = (g_{t,0}, \dots, g_{t,K}) \in L^2(\mathcal{X} \times [0, 1]; \mathbb{R}^{K+1}) \mid \forall k \geq 1, \ g_{t,k}(\boldsymbol{x}) \geq 0 \text{ and } \textstyle\sum_{k=0}^K g_{t,k} = 0\} \tag{75}$$

If $g, g' \in \mathcal{G}$ and $\lambda \in [0, 1]$, then for $k \geq 1$, we have

$$\lambda g_{t,k} + (1 - \lambda)g_{t,k}' \geq \lambda \cdot 0 + (1 - \lambda) \cdot 0 = 0 \tag{76}$$

and adding $k = 0$, we have:

$$\lambda g + (1-\lambda)g' \in \mathcal{G} = (\lambda g_{t,0} + (1-\lambda)g'_{t,0}) + \sum_{k=1}^{K}(\lambda g_{t,k} + (1-\lambda)g'_{t,k})$$
$$= \lambda \underbrace{\left(g_{t,0} + \sum_{k=1}^{K} g_{t,k}\right)}_{=0} + (1-\lambda)\underbrace{\left(g'_{t,0} + \sum_{k=1}^{K} g'_{t,k}\right)}_{=0}$$
$$= 0 \tag{77}$$

which means $\lambda g + (1-\lambda)g' \in \mathcal{G}$ and $\mathcal{G}$ is convex. $\qquad\square$

*Proof of Closure.* First, we define what it means to be weakly closed under $L^2$.

> **Definition C.2** (Weak Closure). *A set of functions $\mathcal{G}$ is said to be closed in the weak topology of $L^2$ if the statement is true: if a sequence of functions $\{g^{(n)} : g^{(n)} \in \mathcal{G}\}$ indexed by $n$, converges to some function $g^{(\infty)} \in L^2$ as $n \to \infty$, then $g^{(\infty)} \in \mathcal{G}$.*

To show that $\mathcal{G}$ is weakly closed under $L^2(\Omega; \mathbb{R}^{K+1})$, where $\Omega := \mathcal{X} \times [0,1]$, we need to show that all sequences $\{g_{t,k}^{(n)} : g_{t,k}^{(n)} \in \mathcal{G}\}$ converge to $g \in \mathcal{G}$ as $n \to \infty$, such that $g \geq 0$.

We decompose $\mathcal{G} := C \cap S$, where $C$ is the non-negative convex cone on the secondary branch growth rates and $S$ is the linear mass conservation constraint, and show that $C$ and $S$ are weakly closed in $L^2(\Omega; \mathbb{R}^{K+1})$. First, we define $C$ as:

$$C := \{g \in L^2(\Omega; \mathbb{R}^{K+1}) : g_{t,k} \geq 0 \text{ a.e. on } \mathcal{X} \times [0,1] \ \forall k \in \{1,\ldots,K\}\} \tag{78}$$

To prove $C$ is weakly closed, we need to show that if $g^{(n)} \in C, g^{(n)} \rightharpoonup g \implies g \in C$. Suppose $g^{(n)}$ converges weakly to $g$, i.e., $g^{(n)} \rightharpoonup g$. Then, for any non-negative square integrable function $\psi \in L^2(\Omega)$, clearly the inner product with $g^{(n)} \geq 0$ is non-negative:

$$\langle g^{(n)}, \psi \rangle \geq 0 \implies \langle g, \psi \rangle \geq 0 \tag{79}$$

by weak convergence $\langle g^{(n)}, \psi \rangle \to \langle g, \psi \rangle$. Since this holds for all non-negative $\varphi \in L^2(\Omega)$, we have that $g_{t,k} \geq 0$ almost everywhere and $g \in C$, so we conclude that $C$ is weakly closed under $L^2(\Omega; \mathbb{R}^{K+1})$.

For the linear constraint set $S$ defined as:

$$S := \{g \in L^2(\Omega; \mathbb{R}^{K+1}) : g_0 + \sum_{k=1}^{K} g_k = 0 \text{ a.e. on } \mathcal{X} \times [0,1]\} \tag{80}$$

Let $T : L^2(\Omega; \mathbb{R}^{K+1}) \to L^2(\Omega)$ be the bounded linear operator defined as:

$$T(g) := g_0 + \sum_{k=1}^{K} g_k \tag{81}$$

Then, we have that $S = \text{kernel}(T) = \{g \in L^2(\Omega; \mathbb{R}^{K+1}) : T(g) = 0\}$ is the kernel of $T$. To prove weak closure of $S$ on $L^2(\Omega; \mathbb{R}^{K+1})$, we need to show for all $T(g^{(n)}) \to 0 \implies T(g) = 0$. Since $T$ is a bounded linear functional, we have:

$$g^{(n)} \rightharpoonup g \implies T(g^{(n)}) \rightharpoonup T(g) \in L^2(\Omega) \tag{82}$$

But we know $T(g^{(n)}) = 0$, which means that:

$$T(g^{(n)}) \to 0 \implies T(g) = 0 \tag{83}$$

which proves that $S$ is weakly closed under $L^2(\Omega; \mathbb{R}^{K+1})$. Since we have shown that both $C$ and $S$ are weakly closed under $L^2(\Omega; \mathbb{R}^{K+1})$, we can conclude that their intersection $\mathcal{G} = C \cap S$ is weakly closed. This concludes our proof of Lemma C.3. $\qquad\square$

> **Lemma C.4.** *Given a minimizing sequence $\{g^{(n)} \in \mathcal{G}\}$ under the functional $\mathcal{L}(g) : L^2 \to \mathbb{R}$ such that $\mathcal{L}(g^{(n)}) \to \inf_{g \in \mathcal{G}} \mathcal{L}(g)$, there exists a subsequence $\{g^{(n_i)} \in \mathcal{G}\}$ that converges weakly in $L^2$ to some limit $g^\star \in \mathcal{G}$.*

*Proof.* It suffices to show the following:

1. The functional $\{g^{(n)}\}$ is bounded in $L^2$ such that there exists a positive value $M$ where $\|g^{(n)}\|_{L^2} \leq M$ for all $n$.

2. Since $L^2(\Omega; \mathbb{R}^{K+1})$ is reflexive, every bounded sequence has a weakly convergent subsequence.

The proof of Part 1 follows from the growth penalty term in $\mathcal{L}(g)$ defined as the squared-norm $\|g\|_{L^2}^2$ of the growth functional. This term ensures that $\mathcal{L}(g)$ is *coercive*, such that

$$\|g^{(n)}\|_{L^2} \to \infty \implies \mathcal{L}(g^{(n)}) \to \infty \tag{84}$$

which ensures that the sequence does not diverge to infinity in norm without incurring a penalty from the loss functional. Given that $\mathcal{L}(g^{(n)}) \to \inf_{g \in \mathcal{G}} \mathcal{L}(g)$, where $\inf_{g \in \mathcal{G}} \mathcal{L}(g) < \infty$, it follows from coercivity that $\|g^{(n)}\|_{L^2}$ does not diverge to infinity and $\{g^{(n)}\}$ is bounded in $L^2$.

The proof of Part 2 follows from the well-established result that for $1 < p < \infty$, the space $L^p$ is reflexive (Beauzamy, 2011). Therefore, by reflexivity of $L^2$, we have that every minimizing sequence $\{g^{(n)}\}$ has a weakly convergent subsequence $\{g^{(n_i)}\}$, such that $g^{(n_i)} \to g^\star$ in $L^2$, concluding our proof of Lemma C.4. $\qquad \square$

Before proving that each component of the loss functional is weakly lower semi-continuous, we establish the definitions for weakly continuous and weakly lower semi-continuous functionals.

**Definition C.3** (Weakly Continuous Functionals). *A functional $\mathcal{L} : L^2 \to \mathbb{R}$ is said to be weakly continuous if it satisfies*

$$n \to \infty \implies g^{(n)} \to g \implies \mathcal{L}(g^{(n)}) \to \mathcal{L}(g) \tag{85}$$

*such that if a sequence $\{g^{(n)} : g^{(n)} \in \mathcal{G}\}$ converges $g^{(n)} \to g$ as $n \to \infty$, then the functional also converges $\mathcal{L}(g^{(n)}) \to \mathcal{L}(g)$ and $g^{(n)} \rightharpoonup g$.*

**Definition C.4** (Weakly Lower Semi-Continuous Functionals). *A functional $\mathcal{L} : L^2 \to \mathbb{R}$ is said to be weakly lower semi-continuous (w.l.s.c.) if it satisfies*

$$n \to \infty \implies g^{(n)} \to g \implies \liminf_{n \to \infty} \mathcal{L}(g^{(n)}) \geq \mathcal{L}(g) \tag{86}$$

*such that if a sequence $\{g^{(n)} : g^{(n)} \in \mathcal{G}\}$ converges $g^{(n)} \to g$ as $n \to \infty$, then the functional is lower bounded by $\mathcal{L}(g)$. By definition, weak continuity implies w.l.s.c.*

**Lemma C.5.** *The functional $\mathcal{L}_{match}(g) : L^2(\Omega; \mathbb{R}^{K+1}) \to \mathbb{R}$ defined as*

$$\mathcal{L}_{match}(g) = \sum_{k=0}^K \mathbb{E}_{p_{t,k}^\star} \left( w_{0,k} + \int_0^1 g_{t,k} dt - w_{1,k}^\star \right)^2$$

$$= \sum_{k=0}^K \left( w_{0,k} + \int_0^1 \mathbb{E}_{p_{t,k}^\star}[g_{t,k}] dt - w_{1,k}^\star \right)^2 \qquad \text{(linearity of expectation)}$$

$$= \sum_{k=0}^K \left( \int_0^1 \int_{\mathcal{X}} p_{t,k}^\star g_{t,k} d\boldsymbol{x} dt + c \right)^2 \tag{87}$$

*where $c = w_{0,k} - w_{1,k}^\star$ is a constant, is weakly continuous in $L^2(\Omega; \mathbb{R}^{K+1})$.*

*Proof of Lemma.* Define the map:

$$\phi(g_{t,k}) := \int_0^1 \int_{\mathcal{X}} p_{t,k}^\star g_{t,k} d\boldsymbol{x} dt \tag{88}$$

First, we show that $\phi(g_{t,k})$ is a bounded linear functional in $L^2(\Omega)$. By linearity of integration, we have that for two functions $g, g' \in \mathcal{G}$, $\phi(cg + c'g') = c\phi(g) + c'\phi(g')$, so $\phi(\cdot)$ is a linear

map. To establish that $\phi(\cdot)$ is bounded in $L^2(\Omega)$, we must show that $|\phi(g)| \leq C\|g\|_{L_2(\Omega)}$. By the Cauchy-Schwarz Inequality (Steele, 2004), we have

$$|\phi(g_{t,k})| = |\langle p^{\star}_{t,k}, g_{t,k}\rangle| = \left|\int_0^1 \int_{\mathcal{X}} p^{\star}_{t,k} g_{t,k} d\boldsymbol{x}\right| \leq \|p^{\star}_{t,k}\|_{L^2(\Omega)}\|g_{t,k}\|_{L^2(\Omega)} \tag{89}$$

which is valid given $p^{\star}_{t,k}, g_{t,k} \in L^2(\Omega)$. Since $\|p^{\star}_{t,k}\|_{L^2(\Omega)}$ is a fixed constant with respect to $g_{t,k}$, we have shown that $|\phi(g_{t,k})| \leq C\|g_{t,k}\|_{L_2(\Omega)}$ and $g_{t,k}$ is bounded. By the definition of weak topology on $L^2(\Omega)$, all bounded linear functionals are weakly continuous, such that $\lim_{n\to\infty} \phi(g^{(n)}) \to \phi(g)$. Given that $\phi(g)$ is weakly continuous, as $g^{(n)} \to g^{\star}$, we have $\phi(g^{(n)}) \to \phi(g^{\star})$. Since the square function $\psi(\cdot) = (\cdot)^2$ is convex, continuous, and bounded below by $\psi \geq 0$, the function does not contain discontinuities and we have weak convergence of $(\phi(g^{(n)}) - c)^2 \to (\phi(g^{\star}) - c)^2$. Therefore, we can conclude that each term in the sum is weakly continuous, and since the sum of weakly continuous functionals is also weakly continuous, we conclude our proof of Lemma C.5. $\square$

**Lemma C.6.** *The functional $\mathcal{L}_{energy}(g) : L^2(\Omega; \mathbb{R}^{K+1}) \to \mathbb{R}$ defined as*

$$\mathcal{L}_{energy}(g) = \int_0^1 \mathbb{E}_{p^{\star}_{t,k}}\left[\frac{1}{2}\|u_{t,k}(X_{t,k})\|_2^2 + V_t(X_{t,k})\right]\left(w_{0,k} + \int_0^t g_{s,k}(X_{s,k})ds\right)dt$$

$$= \int_0^1 \alpha(t)\left(w_{0,k} + \int_0^t \int_{\mathcal{X}} p^{\star}_{s,k} g_{s,k} d\boldsymbol{x}ds\right)dt \qquad \text{(linearity of expectation)}$$

*where $\alpha(t)$ is a constant not dependent on $g_{t,k}$, is weakly lower semi-continuous in $L^2(\Omega; \mathbb{R}^{K+1})$ (w.l.s.c.).*

*Proof of Lemma.* For each $s \in [0, 1]$, we define the functional:

$$\phi_s(g_{s,k}) = \int_{\mathcal{X}} p^{\star}_{s,k} g_{s,k} d\boldsymbol{x} \tag{90}$$

which is a bounded linear functional on $L^2(\mathcal{X})$ for all $s$. Applying Fubini's theorem to change the order of integration, we have:

$$\int_0^1 \alpha(t)\left(\int_0^t \phi_s(g)ds\right)dt = \int_0^1 \underbrace{\left(\int_s^1 \alpha(t)dt\right)}_{=:\beta(s)}\phi_s(g)ds \tag{91}$$

Defining $\beta(s) := \int_s^1 \alpha(t)dt$, we have:

$$\int_0^1 \beta(s)\int_{\mathcal{X}} p^{\star}_{s,k} g_{s,k} d\boldsymbol{x}ds = \langle \beta p^{\star}_k, g_k\rangle_{L^2(\Omega)} \tag{92}$$

which is a bounded linear functional in $L^2$, and thus is weakly continuous in $L^2$, and we have shown that $\mathcal{L}_{energy}(g)$ is weakly continuous. Since weak continuity implies w.l.s.c., and the sum of w.l.s.c. functionals is also w.l.s.c., we conclude our proof of Lemma C.6. $\square$

**Lemma C.7.** *The mass loss functional $\mathcal{L}_{mass}(g) : L^2(\Omega; \mathbb{R}^{K+1}) \to \mathbb{R}$ defined as*

$$\mathcal{L}_{mass}(g) = \int_0^1 \mathbb{E}_{\{p_{t,k}\}_{k=0}^K}\left[\sum_{k=0}^K w^{\phi}_{t,k} - w^{total}_t\right]^2 dt$$

$$= \int_0^1 \left[\sum_{k=0}^K \left(w_{0,k} + \int_0^t \mathbb{E}_{p_{t,k}}[g_{t,k}]\right) - w^{total}_t\right]^2 dt \qquad \text{(linearity of expectation)}$$

$$= \int_0^1 \left[1 + \sum_{k=0}^K \int_0^t \mathbb{E}_{p_{t,k}}[g_{t,k}] - w^{total}_t\right]^2 dt \tag{93}$$

*where $w^{total}_t$ is a constant not dependent on $g_{t,k}$, is weakly lower semi-continuous in $L^2(\Omega; \mathbb{R}^{K+1})$ (w.l.s.c.). Note that we do not need to account for the negative penalty loss as we assume the sum of growth functions is zero.*

*Proof of Lemma.* For this proof, we consider the bounded linear operator $\mathcal{A} : L^2(\Omega; \mathbb{R}^{K+1}) \rightarrow L^2(0, 1)$ defined as:

$$(\mathcal{A}g)(t) := \sum_{k=0}^{K} \int_0^t \int_{\mathcal{X}} p_{s,k}(\boldsymbol{x}) g_{t,k}(\boldsymbol{x}) d\boldsymbol{x} ds \tag{94}$$

Then, we have:

$$\mathcal{L}_{\text{mass}}(g) = \int_0^1 \left[1 + (\mathcal{A}g)(t) - w_t^{\text{total}}\right]^2 dt = \left\|1 + (\mathcal{A}g)(t) - w_t^{\text{total}}\right\|_{L^2([0,1])}^2 \tag{95}$$

Since the bounded linear operator $\mathcal{A}$ is weak-to-weak continuous and the squared norm $\|\cdot\|_{L^2}^2$ is w.l.s.c. on $L^2([0, 1])$, if $g^{(n)} \rightharpoonup g$ in $L^2(\Omega; \mathbb{R}^{K+1})$, then $\mathcal{A}g^{(n)} \rightharpoonup \mathcal{A}g$ in $L^2([0, 1])$ and we have:

$$\liminf_{n \to \infty} \mathcal{L}_{\text{mass}}(g^{(n)}) = \liminf_{n \to \infty} \left\|1 + \mathcal{A}g^{(n)} - w_t^{\text{total}}\right\|_{L^2([0,1])}^2 \geq \left\|1 + \mathcal{A}g - w_t^{\text{total}}\right\|_{L^2([0,1])}^2 = \mathcal{L}_{\text{mass}}(g) \tag{96}$$

which means $\mathcal{L}_{\text{mass}}(g)$ is w.l.s.c. and we conclude our proof of Lemma C.7. $\square$

**Lemma C.8.** *The combined Stage 3 training loss $\mathcal{L}_{growth}(g) = \lambda_{energy}\mathcal{L}_{energy}(g) + \lambda_{match}\mathcal{L}_{match}(g) + \lambda_{mass}\mathcal{L}_{mass}(g) + \lambda_{growth}\|g\|_{L^2}^2$ is weak lower semi-continuous in $L^2(\Omega; \mathbb{R}^{K+1})$ (w.l.s.c.).*

*Proof of Lemma.* Given the weighted sum of w.l.s.c. functionals $\sum_{i=1}^{M} \lambda_i \mathcal{L}_i(g)$ that each satisfy $\lim_{n\to\infty} \mathcal{L}_i(g^{(n)}) \geq \mathcal{L}_i(g)$ for constants $\{\lambda_i\}_{i=1}^{M}$ for a converging sequence $g^{(n)} \rightarrow g$ as $n \rightarrow \infty$, it follows easily that

$$\liminf_{n \to \infty} \sum_{i=1}^{M} \lambda_i \mathcal{L}_i(g^{(n)}) \geq \sum_{i=1}^{M} \lambda_i \mathcal{L}_i(g) \tag{97}$$

where the weightes $\lambda_i \geq 0$. By definition, the $L^2$ norm is weakly lower semi-continuous. Since $\mathcal{L}_{\text{energy}}$, $\mathcal{L}_{\text{match}}$, and $\mathcal{L}_{\text{mass}}$ are w.l.s.c. by Lemmas C.5, C.6, and C.7, we conclude that $\mathcal{L}_{\text{growth}}(g)$ is w.l.s.c., which concludes the proof of Lemma C.8. $\square$

In total, we have shown:

1. The set $\mathcal{G}$ of *feasible* growth functions is convex and closed under the weak topology of the set of square integrable functions $L^2$ (Lemma C.3).

2. The minimizing sequence $\{g^{(n)}\}$ has a weakly convergent subsequence (Lemma C.4).

3. The functional $\mathcal{L}(g)$ is weakly lower semi-continuous (Lemmas C.5, C.6, C.7, C.8).

Thus, by the *direct method in the calculus of variations*, we have shown

$$\exists g^\star = (g_{t,0}^\star, \ldots, g_{t,1}^\star) \in \mathcal{G} \text{ s.t. } \mathcal{L}(g^\star) = \inf_{g \in \mathcal{G}} \mathcal{L}(g) \tag{98}$$

which concludes the proof of Proposition 4.2. $\square$

# D  ADDITIONAL EXPERIMENTS AND DISCUSSIONS

## D.1  COMPARISON TO SINGLE-BRANCH SCHRÖDINGER BRIDGE MATCHING

**Setup** For each experiment, we compared the performance of BranchSBM against single-branch SBM. Instead of clustering the data at $t = 1$ into distinct endpoint distributions, we left the unclustered data as a single target distribution $p_1$ and let the model learn the optimal Schrödinger Bridge from the initial distribution $\pi_0$. For the single-branch task, we assume mass conservation and set the weights for all samples from $\pi_0$ and $\pi_1$ to 1.0, while keeping the model architecture, state-cost $V_t$, and hyperparameters equivalent to BranchSBM. Since single-branch SBM does not require modeling the growth of separate branches, we train only Stages 1 and 2 to optimize the drift field $u_t^\theta$ of the single branch. For evaluation, the trajectories of validation samples from the initial distribution $\boldsymbol{x}_0 \sim \pi_0$ were simulated over $t \in [0, 1]$ and compared with the ground truth distribution at $t = 1$. For BranchSBM, we take the overall distribution generated across all branches and compare it to the overall ground truth distribution.

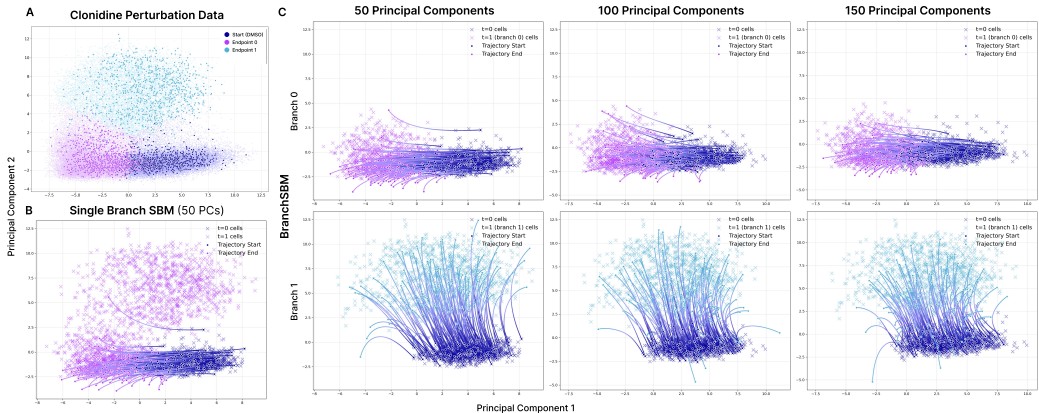

Figure 6: **Results for Clonidine Perturbation Modeling.** **(A)** Gene expression data of DMSO control (set to $t = 0$) and cell states (set to $t = 1$) after Clonidine perturbation with two distinct endpoints (pink and purple). **(B)** The simulated trajectories for single-branch SBM on the top 50 PCs with both clusters. All samples take the low-energy path without reaching the second cluster. **(C)** The simulated endpoints of the top 50, 100, and 150 PCs at $t = 1$ on the validation data for each branch.

**Modeling Mouse Hematopoiesis Differentiation** In Figure 7, we provide a side-by-side comparison of the reconstructed distributions at time $t_1$ and $t_2$ using BranchSBM (top) with two branches and single-branch SBM (bottom), as well as the learned trajectories over the time interval $t \in [t_1, t_2]$. While single-branch SBM produces samples that loosely capture the two target cell fates, the resulting distributions display high variance and fail to align with the true differentiation trajectories. In contrast, BranchSBM generates intermediate distributions that are sharply concentrated along the correct developmental paths, more faithfully reflecting the underlying branching structure of the data.

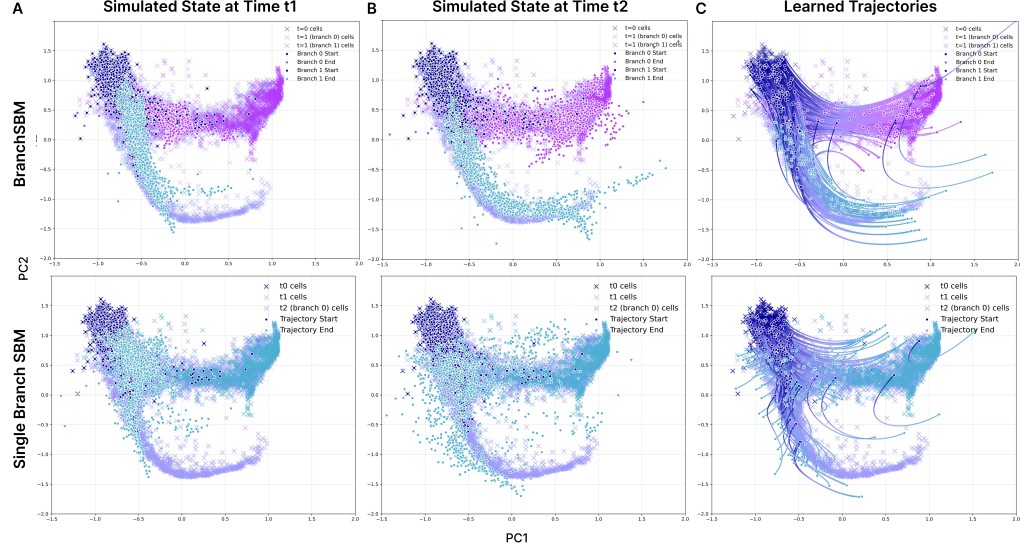

Figure 7: **Comparison of BranchSBM to Single-Branch SBM for Cell-Fate Differentiation.** Mouse hematopoiesis scRNA-seq data is provided for three time points $t_0, t_1, t_2$. **(A, B)** Distribution of simulated cell states at time **(A)** $t_1$ and **(B)** $t_2$ across both branches for BranchSBM (top) and single-branch SBM (bottom). **(C)** Learned trajectories of BranchSBM and single-branch SBM over the $t \in [t_1, t_2]$ on validation samples.

**Modeling Clonidine and Trametinib Perturbation** For both Clonidine and Trametinib, we performed the single-branch experiment on the top 50 principal components identified by PCA. After training, we simulated all the validation samples from the initial distribution $\pi_0$ by integrating the single velocity field $u_t^\theta$ to $t = 1$. We evaluated the performance of the single-branch parameterization

by computing the RBF-MMD (104) of all PCs and $\mathcal{W}_1$ (102) and $\mathcal{W}_2$ (103) distances of the top-2 PCs of the simulated samples at time $t = 1$ with the ground truth data points.

In Table 4, we show that BranchSBM with two branches trained on gene expression vectors across all dimensions $d \in \{50, 100, 150\}$ outperforms single-branch SBM on dimension $d = 50$ in reconstructing the distribution of cells perturbed with Clonidine. In Table 5, we further show improved performance of BranchSBM with three branches compared to single-branch SBM.

In Figure 8A and 8B, we see that single-branch SBM only reconstructs cluster 0, while failing to generate samples from cluster 1 for Clonidine perturbation or clusters 1 and 2 for Trametinib perturbation. The endpoints for both perturbation experiments are clustered largely on variance in the first two or three principal components (PCs), where PC1 captures the divergence from the control state to the perturbed clusters and PC2 and higher captures the divergence between clusters in the perturbed population, where cluster 0 is closest to the control state along PC2 and cluster 1 and 2 are farther from the control. From Figure 8A and B, we can conclude that single-branch SBM is not expressive enough to capture the complexities of higher-dimensional PCs and follows the most obvious trajectory from the control to cluster 0, resulting in an inaccurate representation of the perturbed cell population.

In contrast, we demonstrate that BranchSBM is capable of stimulating trajectories to both clusters in the population perturbed with Clonidine (Figure 8B) and three clusters in the population perturbed with Trametinib (Figure 8D), generating branched distributions that accurately capture the location and spread of the perturbed cell distribution in the dataset.

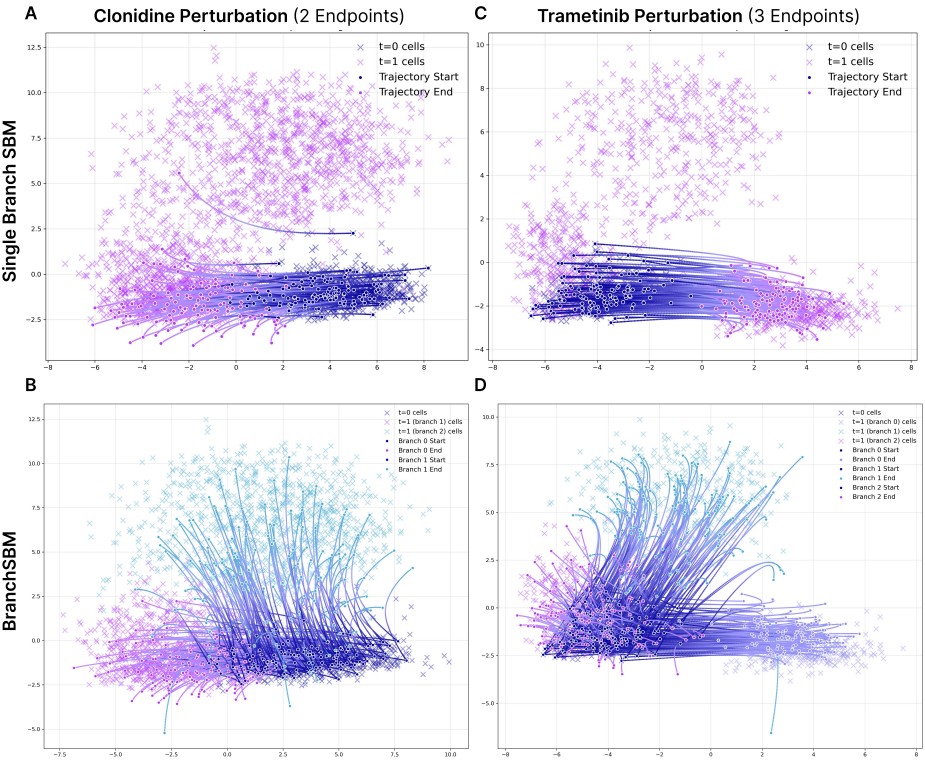

Figure 8: **Comparison of BranchSBM to Single-Branch SBM for Perturbation Modelling. (A, B)** Clonidine perturbation trajectories with two target clusters generated by **(A)** single-branch SBM and **(B)** BranchSBM from the validation data. **(C, D)** Trametinib perturbation trajectories with three target clusters generated by **(C)** single-branch SBM and **(D)** BranchSBM. In both experiments, single-branch SBM only generated states in cluster 0 and not cluster 1 or 2, whereas BranchSBM reconstructed all perturbed clusters via branched trajectories.

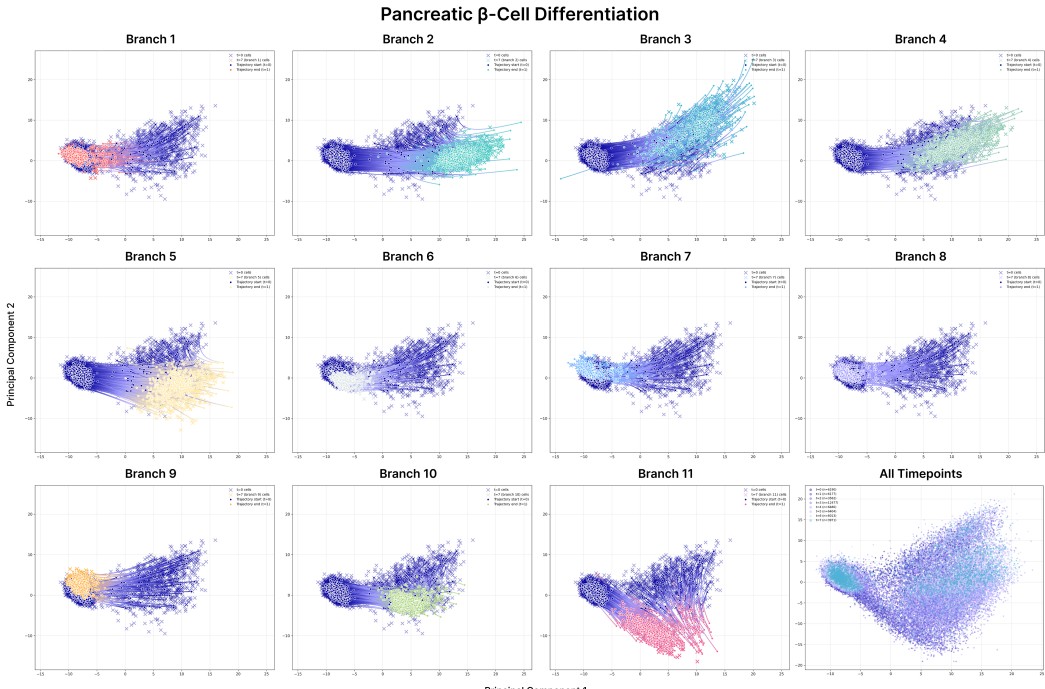

Figure 9: **Results for Pancreatic $\beta$-Cell Differentiation Modeling with BranchSBM.** We evaluate BranchSBM on a pancreatic $\beta$-cell differentiation dataset containing $51,274$ cells collected across eight time points. Cells are projected into a 30-dimensional PCA space, and Leiden clustering is used to define 11 terminal cell populations at the final time point. The trajectories of each branch $k \in \{0, \dots, 10\}$ are simulated from validation points at $t_0$ and are plotted with the top-2 PCs. The bottom-right plot shows the cells from each intermediate time point.

## D.2 EFFECT OF FINAL JOINT TRAINING ON LOSSES

In Table 7, we show the final losses after convergence, summed across each branch and averaged over the batch size, for Stage 3 training of only the growth networks and Stage 4 joint training of the flow and growth networks discussed in Section 4. All losses are calculated exactly as shown in Section 4. Crucially, we find that the final joint training stage refines the parameters of both the flow and growth networks simultaneously to minimize the energy loss $\mathcal{L}_{\text{energy}}(\theta, \phi)$ (14) while ensuring the growth parameters maintain minimal losses across $\mathcal{L}_{\text{match}}(\phi)$ (15) and $\mathcal{L}_{\text{mass}}(\phi)$ (16) for all experiments. This indicates that jointly optimizing both the drift and growth dynamics leads to further refinement towards modeling the optimal branching trajectories in the data.

Table 7: **Validation Losses for Stage 3 and 4 Training Across Experiments.** Losses are summed across both branches and averaged over batch size. The final Stage 4 joint training stage that refines both the flow and growth networks simultaneously minimize the energy loss $\mathcal{L}_{\text{energy}}(\theta, \phi)$ (14) from Stage 3 while ensuring the growth parameters maintain minimal losses across $\mathcal{L}_{\text{match}}(\phi)$ (15) and $\mathcal{L}_{\text{mass}}(\phi)$ (16)

| Experiment | Stage 3 | | | Stage 4 | | |
|---|---|---|---|---|---|---|
| | $\mathcal{L}_{\text{energy}}(\theta, \phi)$ | $\mathcal{L}_{\text{mass}}(\theta, \phi)$ | $\mathcal{L}_{\text{match}}(\theta, \phi)$ | $\mathcal{L}_{\text{energy}}(\theta, \phi)$ | $\mathcal{L}_{\text{mass}}(\theta, \phi)$ | $\mathcal{L}_{\text{match}}(\theta, \phi)$ |
| LiDAR | 1.276 | $3.0 \times 10^{-5}$ | 0.007 | 0.768 | $2.0 \times 10^{-5}$ | 0.102 |
| Mouse Hematopoiesis | 2.209 | $1.2 \times 10^{-4}$ | 0.054 | 1.918 | $5.0 \times 10^{-5}$ | 0.076 |
| Chlonidine Perturbation | 36.469 | 0.030 | 0.109 | 25.798 | 0.053 | 0.153 |
| Trametinib Perturbation | 35.834 | 0.023 | 0.078 | 32.843 | 0.017 | 0.056 |

# E  EXPERIMENT DETAILS

## E.1  MULTI-STAGE TRAINING

To ensure stable training while incorporating all loss functions, we introduce a multi-stage training approach (Algorithm 1).

**Stage 1**  First, we train a neural interpolant $\varphi_{t,\eta}(\boldsymbol{x}_0, \boldsymbol{x}_{t,k}) : \mathbb{R}^d \times \mathbb{R}^d \times [0,1] \to \mathbb{R}^d$ that takes the endpoints of a branched coupling and defines the optimal interpolating state $X_t$ at time $t$ by minimizing the energy function $\mathcal{L}_{\text{traj}}(\eta)$ in (12). This is used to calculate the optimal conditional velocity $\dot{\boldsymbol{x}}_{t,\eta,k}$ that preserves the endpoints $X_0 = \boldsymbol{x}_0$ and $X_{1,k} = \boldsymbol{x}_{1,k}$ for the flow matching objective in Stage 2.

**Stage 2**  Next, we train a set of flow networks $\{u_{t,k}^{\theta}\}_{k=0}^{K}$ that *generate* the optimal interpolating trajectories for each branch with the conditional flow matching objective $\mathcal{L}_{\text{flow}}$ in (13).

**Stage 3**  We freeze the parameters of the flow networks and only train the growth networks $\{g_{t,k}^{\phi}\}_{k=0}^{K}$ by minimizing $\mathcal{L}_{\text{growth}}$ in (17).

**Stage 4**  Finally, we unfreeze the parameters of both the flow and growth networks and jointly train $\{u_{t,k}^{\theta}, g_{t,k}^{\phi}\}_{k=0}^{K}$ by minimizing the growth loss $\mathcal{L}_{\text{growth}}$ in (17) from Stage 3 in addition to the distribution reconstruction loss $\mathcal{L}_{\text{recons}}$ in (19).

## E.2  GENERAL TRAINING DETAILS

**Leiden Clustering**  To identify branch endpoints in the dataset, we apply an automated clustering pipeline to the final time point ($t = 7$), where the differentiated cell-states are most clearly defined. First, we construct a 20-nearest-neighbor (kNN) graph on the final-timepoint embeddings and run Leiden community detection using the `ModularityVertexPartition` objective. Leiden clustering is particularly suited and the state-of-the-art method for clustering single-cell data, given that it is robust to noise, can handle heterogeneous cluster sizes and shapes, and guarantees well-connected clusters, all while being fully unsupervised and automatic. This gives us stable and biologically meaningful terminal clusters compared to alternative methods such as K-means clustering, which assumes spherical clusters and struggles with sparse high-dimensional manifolds. After initial clustering, we merge any cluster with fewer than `min_cells` cells into the nearest large cluster based on Euclidean centroid distance.

**Model Architecture**  We parameterized the branched trajectory $\varphi_{t,\eta}(\boldsymbol{x}_0, \boldsymbol{x}_1)$ with a 3-layer MLP with Scaled Exponential Linear Unit (SELU) activations. The endpoint pair $(\boldsymbol{x}_0, \boldsymbol{x}_1)$ and the time step $t$ are concatenated into a single $(2d + 1)$-dimensional vector and used as input to the model. Similarly, we parameterize each flow network $u_{t,k}^{\theta}(\boldsymbol{x}_t)$ and growth network $g_{t,k}^{\theta}(\boldsymbol{x}_t)$ with the same 3-layer MLP and SELU activations but takes the interpolating state $\boldsymbol{x}_t$ and time $t$ concatenated into a $(d + 1)$-dimensional vector.

To ensure that the growth rates of all secondary branches are non-negative (i.e. for all $k \in \{1, \ldots, K\}$, $g_{t,k}(X_{t,k}) \geq 0$), we apply an additional softplus activation to the output of the 3-layer MLP in the growth networks, defined as $\text{softplus}(\cdot) = \log(1 + \exp(\cdot))$, which is smooth function that transforms negative values to be positive near 0. For the growth network of the primary branch ($k = 0$), we allow for both positive and negative growth, as all mass starts at the primary branch and flows into the secondary branches, but the primary branch itself can grow as well, depending on whether mass is conserved.

**State Cost $V_t$**  Depending on the dimensionality of the data type, we set the state cost $V_t(X_t) :$ $\mathbb{R}^d \to [0, +\infty]$ to be either the LAND or RBF metric discussed in Appendix A.2. For the experiments on LiDAR ($d = 3$) and Mouse Hematopoiesis scRNA-seq ($d = 2$) data, we used the LAND metric (33) with hyperparameters $\sigma = 0.125$ and $\varepsilon = 0.001$.

To avoid the task of setting a suitable variance $\sigma$ for the high-dimensional gene expression space $d \in \{50, 100, 150\}$, we use the RBF metric (37) that learns parameters to ensure the regions within the data manifold have low cost and regions far from the data manifold have high cost. Using the training scheme in Kapuśniak et al. (2024), we identified $N_c$ cluster centers with $k$-means clustering, and trained the parameters $\{\omega_{j,n}\}_{n=1}^{N_c}$ by minimizing $\mathcal{L}_{\text{RBF}}$ (38) on the training data. We found that setting the number of cluster centers $N_c$ too low resulted in non-decreasing $\mathcal{L}_{\text{RBF}}$ and that increasing $N_c$ for higher dimensions enabled more effective training. Furthermore, we found that modeling higher-dimensional PCs required setting a larger $\kappa$, which determines the *spread* of the RBF kernel around each cluster center. The specific values for $N_c$ and $\kappa$ depending on the dimension of principal components are provided in Table 11.

**Optimal Transport Coupling**  Since our experiments consist of unpaired initial and target distributions and we seek to minimize the energy of the interpolating bridge, we define pairings $(\boldsymbol{x}_0, \boldsymbol{x}_{1,k})$ using the optimal transport plan $\pi_{0,1,k}^\star$ that minimizes the distance between the initial distribution $\pi_0$ and each target distribution $\pi_{1,k}$ in probability space. Specifically, we define $\pi_{0,1,k}^\star$ as the 2-Wasserstein transport plan (Villani et al., 2008) between $\pi_0$ and $\pi_{1,k}$ defined as

$$\pi_{0,1,k}^\star = \arg\min_{\pi_{0,1,k} \in \Pi} \int_{\pi_0 \otimes \pi_{1,k}} \|\boldsymbol{x}_0 - \boldsymbol{x}_{1,k}\|_2^2 d\pi(\boldsymbol{x}_0, \boldsymbol{x}_{1,k}) \tag{99}$$

where $\pi_0 \otimes \pi_{1,k}$ is the set of all possible couplings between the endpoint distributions. For each of the branches, the dataset was paired such that $(\boldsymbol{x}_0, \boldsymbol{x}_{1,k}) \sim \pi_{0,1,k}^\star$.

**Training**  We train each stage for a maximum of 100 epochs. For Stage 1, we used the Adam optimizer (Kingma & Ba, 2014) with a learning rate of $1.0 \times 10^{-4}$ to train $\varphi_{t,\eta}(\boldsymbol{x}_0, \boldsymbol{x}_1)$. For Stage 2, 3, and 4, we used the AdamW optimizer (Loshchilov & Hutter, 2017) with weight decay $1.0 \times 10^{-5}$ and learning rate $1.0 \times 10^{-3}$ to train each flow network $u_{t,k}^\theta$ and growth network $g_{t,k}^\phi$. All experiments were performed on one NVIDIA A100 GPU. We trained on the LiDAR and mouse hematopoiesis data with a batch size of 128 and the Clonidine and Trametinib perturbation data with a batch size of 128, each divided with a 0.9/0.1 train/validation split. All hyperparameters across experiments are provided in Table 11.

**Computational Overhead**  Although we train $K + 1$ flow and growth networks, the overall training time remains comparable to that of single-branch SBM, since the networks for each branch are trained only on the subset of data corresponding to its respective target distribution. While the method does incur higher space complexity, we find that simple MLP architectures suffice for strong performance, suggesting that scalability is not a major concern. BranchSBM also significantly reduces inference time, as predicting branching population dynamics requires simulating only a single sample from the initial distribution, unlike other models that require simulating large batches of samples.

### E.3  LiDAR Experiment Details

**LiDAR Data**  We used the same LiDAR manifold from Liu et al. (2023a); Kapuśniak et al. (2024). The data is a collection of three-dimensional point clouds within 10 unit cubes $[-5, 5]^3 \subset \mathbb{R}^3$ that span the surface of Mount Rainier. Given any point $\boldsymbol{x} \in \mathbb{R}^3$ in the three-dimensional space, we project it onto the LiDAR manifold by identifying the $k$-nearest neighbors $\{\boldsymbol{x}_1, \ldots, \boldsymbol{x}_k\}$ and fitting a 2D tangent plane to the set of neighbors

$$\arg\min_{a,b,c} \frac{1}{k} \sum_{i=1}^{k} \exp\left(\frac{-\|\boldsymbol{x} - \boldsymbol{x}_i\|}{\tau}\right) (a\boldsymbol{x}_i^{(x)} + b\boldsymbol{x}_i^{(y)} + c - \boldsymbol{x}_i^{(z)})^2 \tag{100}$$

where $\tau = 0.001$ following Liu et al. (2023a). Then, we solve for the tangent plane $ax + by + c = z$ using the Moore-Penrose pseudoinverse from $k = 20$ neighbors. From the tangent plane, we can project any point $\boldsymbol{x}$ to the LiDAR manifold with the function $\pi(\boldsymbol{x})$ defined as

$$\pi(\boldsymbol{x}) = \boldsymbol{x} - \left(\frac{\boldsymbol{x}^\top \mathbf{v} + c}{\|\mathbf{v}\|_2^2}\right) \mathbf{v}, \quad \text{where } \mathbf{v} = \begin{bmatrix} a \\ b \\ -1 \end{bmatrix} \tag{101}$$

**Synthetic Distributions**    To reformulate the experiment in Liu et al. (2023a) as a branching problem, we define a single initial distribution and two divergent target distributions. Specifically, we define a single initial distribution $\pi_0 = \mathcal{N}(\mu_0, \sigma_0)$ as a mixture of four Gaussians and two target distributions $\pi_{1,0}, \pi_{1,1}$ on either side of the mountain, both as mixtures of three Gaussians. The exact parameters of each Gaussian are given in Table 8.

We sample a total of 5000 points i.i.d. from each of the Gaussian mixtures $\{\boldsymbol{x}_0^i\}_{i=1}^{5000} \sim \pi_0$, $\{\boldsymbol{x}_{1,0}^i\}_{i=1}^{5000} \sim \pi_{1,0}$, and $\{\boldsymbol{x}_{1,1}^i\}_{i=1}^{5000} \sim \pi_{1,1}$. The data points are projected to the LiDAR manifold with the projection function $\pi(\boldsymbol{x})$ in (101).

Table 8: **Synthetic Gaussian mixture distribution parameters for LiDAR experiment.** 5000 datapoints are drawn i.i.d. from each of the Gaussian mixtures and paired randomly $(\boldsymbol{x}_0, \boldsymbol{x}_{1,0}, \boldsymbol{x}_{1,1})$ to define the training dataset. A visualization of the training data on the LiDAR manifold is provided in Figure 3.

|  | Distribution | $\mu$ | $\sigma$ |
|---|---|---|---|
| $\pi_0$ | $\mathcal{N}(\mu_0, \sigma_0)$ | $(-4.5, -4.0, 0.5), (-4.2, -3.5, 0.5), (-4.0, -3.0, 0.5), (-3.75, -2.5, 0.5)$ | 0.02 |
| $\pi_{1,0}$ | $\mathcal{N}(\mu_{1,0}, \sigma_{1,0})$ | $(-2.5, -0.25, 0.5), (-2.25, 0.675, 0.5), (-2, 1.5, 0.5)$ | 0.03 |
| $\pi_{1,1}$ | $\mathcal{N}(\mu_{1,1}, \sigma_{1,1})$ | $(2, -2, 0.5), (2.6, -1.25, 0.5), (3.2, -0.5, 0.5)$ | 0.03 |

**Evaluation Metrics**    To determine how closely the simulated trajectories match the ground truth trajectories, we compute the Wasserstein-1 ($\mathcal{W}_1$) and Wasserstein-2 ($\mathcal{W}_2$) distances defined as

$$\mathcal{W}_1 = \left( \min_{\pi \in \Pi(p,q)} \int \|\boldsymbol{x} - \mathbf{y}\|_2 d\pi(\boldsymbol{x}, \mathbf{y}) \right) \tag{102}$$

$$\mathcal{W}_2 = \left( \min_{\pi \in \Pi(p,q)} \int \|\boldsymbol{x} - \mathbf{y}\|_2^2 d\pi(\boldsymbol{x}, \mathbf{y}) \right)^{1/2} \tag{103}$$

where $p$ denotes the ground truth distribution and $q$ denotes the predicted distribution. After training the velocity and growth networks on samples from the initial Gaussian mixture $\pi_0$ and target Gaussian mixtures $\pi_{1,0}$ and $\pi_{1,1}$, we evaluate $\mathcal{W}_1$ and $\mathcal{W}_2$ of the reconstructed distribution simulated from the validation points in the initial distribution $\pi_0$ against the true distribution at $t = 1$.

### E.4    DIFFERENTIATING SINGLE-CELL EXPERIMENT DETAILS

**Mouse Hematopoiesis scRNA-seq Data**    We used the mouse hematopoiesis dataset from Zhang et al. (2025c) consisting of three timesteps $t_0, t_1, t_2$, with a total of 1429 cells at $t_0$, 3781 cells from $t_1$, and 5788 cells from $t_2$. The data points at $t_0$ form a homogeneous cluster, while the data points at $t_2$ are clearly divided into two distinct cell fates. We performed $k$-means clustering with $k = 2$ clusters to create branching on the cells at $t_2$, splitting the cells into two clusters: endpoint 0 with 2902 cells and endpoint 1 with 2886 cells. Since the two endpoints are near equal in ratio, we set the final weights of both endpoints as 0.5 (i.e. $w_{1,0} = w_{1,1} = 0.5$). To match the size of the $t_0$ samples, we randomly sampled 1429 samples from each of these two clusters and used them as the endpoints for branches 0 and 1, respectively. Training and validation follow a $0.9/0.1$ split ratio.

**Pancreatic $\beta$-Cell Differentiation Data**    The pancreatic $\beta$-cell dataset from Veres et al. (2019) contains $51,274$ cells over eight time points that evolve from human pluripotent stem cells to pancreatic $\beta$-like cells. Following (Zhang et al., 2025d), the gene expression representations are projected to a 30-dimensional PCA space. After Leiden clustering with $k = 20$ and `min_cells = 100`, we obtain 11 clusters for $K = 11$ branches.

**Evaluation Metrics**    To determine how closely the reconstructed distributions along the trajectory match the ground truth distributions, we compute the 1-Wasserstein ($\mathcal{W}_1$) (102) and 2-Wasserstein ($\mathcal{W}_2$) (103) distances similar to the LiDAR experiment. After training the velocity and growth networks on samples from the initial cell distribution $\pi_{t_1}$ and differentiated target cell distributions $\pi_{t_2,0}$ and $\pi_{t_2,1}$, we evaluate $\mathcal{W}_1$ and $\mathcal{W}_2$ between the reconstructed branched distributions at the intermediate time $t_1$ ($p_{t_1,0}$ and $p_{t_1,1}$) and the target distributions at the final time $t_2$ ($p_{t_2,0}$ and $p_{t_2,1}$) simulated from the validation points in the initial distribution $\pi_0$ and the true distributions $\pi_{t_1}$ and $\pi_{t_2}$.

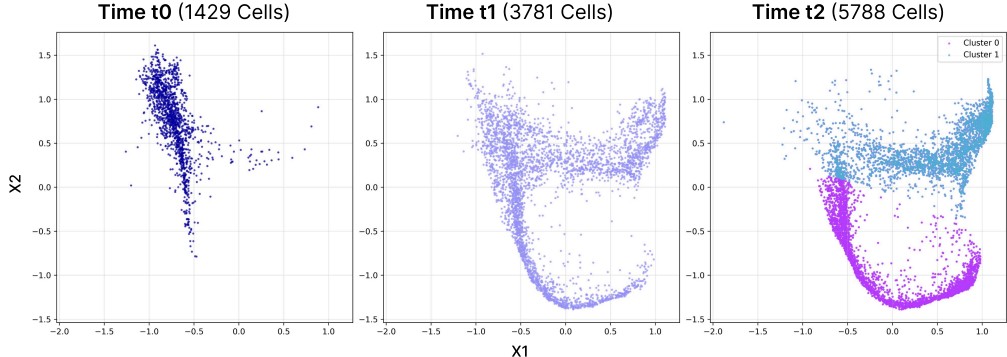

Figure 10: **Mouse Hematopoiesis Single-Cell RNA Sequencing Data Plotted by Time Point.** Real scRNA-seq data is projected to 2D force-directed SPRING plots (Sha et al., 2023; Weinreb et al., 2020; Zhang et al., 2025c). There is a clear divergence of cell fate between times $t_0$ (left), $t_1$ (middle), and $t_2$ (right) from the initial homogeneous progenitor cells into two distinct cell fates (shown in pink and purple in the $t_2$ plot). Cells at time $t_2$ are clustered into endpoint 0 (pink; 2902 cells) and endpoint 1 (turquoise; 2886 cells).

### E.5 CELL-STATE PERTURBATION MODELING EXPERIMENT DETAILS

**Tahoe Single-Cell Perturbation Data** The Tahoe-100M dataset consists of 50 cell lines and over 1000 different drug-dose conditions (Zhang et al., 2025a). For this experiment, we extract the data for a single cell line (A-549) under two drug perturbation conditions selected based on cell abundance and response diversity.

Clonidine at 5 $\mu$L was selected first due to having the largest number of cells at this dosage, while Trametinib was chosen as the second drug based on its second-highest cell count under the same condition. For both drugs, we selected the top 2000 highly variable genes (HVGs) based on normalized expression and projected the data into a 50-dimensional PCA space, which captured approximately 38% of the total variance in both cases.

We applied $K$-nearest neighbor ($K = 50$) and conducted Leiden clustering separately for drugged versus DMSO control conditions. The most abundant DMSO cluster was selected as the initial state ($t = 0$). For Clonidine, we identified two clusters that were most distinct from the DMSO control along PC1 and PC2, respectively. These were selected as two distinct endpoints for branches 1 and 2 at $t = 1$. We applied centroid-based sampling to obtain balanced training sets of 1033 cells per cluster (Figure 11).

For Trametinib, we extended the branching up to three endpoints. From its Leiden clustering results, we identified three clusters that were the most divergent from the DMSO control clusters along PC1, PC2, and PC3, respectively. All selected clusters contained at least 100 cells and were subsampled to 381 cells each for branch training. The remaining cells were clustered with $K$-means into three (Clonidine) or four (Trametinib) groups to construct the metrics dataset. The training and validation dataset split followed a 0.9/0.1 ratio. The final visualization utilized the first two principal components.

Table 9: **Training cluster cell counts for perturbation experiments.**

|  | Clonidine | | Trametinib | | |
| --- | --- | --- | --- | --- | --- |
|  | Cluster 0 | Cluster 1 | Cluster 0 | Cluster 1 | Cluster 2 |
| Original Cell Count | 1675 | 1033 | 1622 | 686 | 381 |
| Initial Weight $w_{0,k}$ | 1.0 | 0 | 1.0 | 0 | 0 |
| Target Weight $w_{1,k}$ | 0.619 | 0.381 | 0.603 | 0.255 | 0.142 |

**Evaluation Metrics** To quantify the alignment of the reconstructed and ground-truth distributions for the cell-state perturbation experiment on principal component (PCs) dimensions

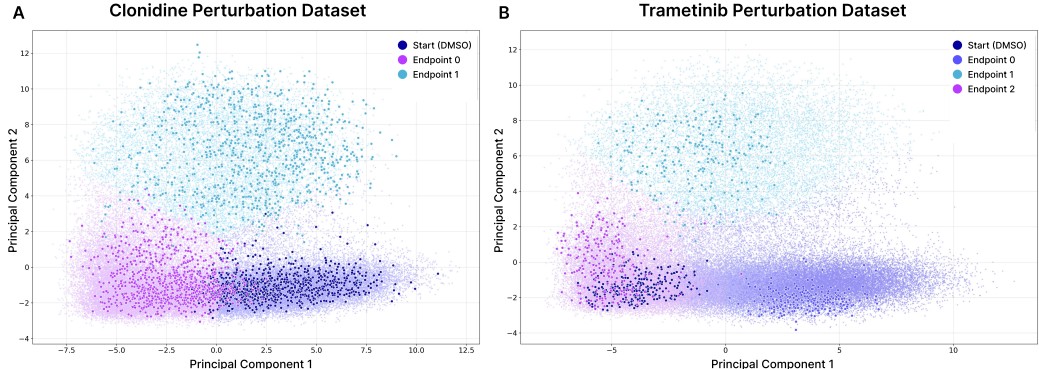

Figure 11: **Clustered Cell-State Perturbation Data from the Tahoe-100M Dataset.** PCA was conducted on all cells for the control DMSO-treated and two drug-treated populations, and clustered. Plots show divergence along the top 2 PCs. **(A)** Clonidine-treated cells ($5\mu M$) are plotted in pink (endpoint 0), and turquoise (endpoint 1), and the distribution of control cells is plotted in navy. **(B)** Trametinib-treated cells ($5\mu M$) are plotted in purple (endpoint 0), turquoise (endpoint 1), and pink (endpoint 2), and the distribution of control cells is plotted in navy.

$d \in \{50, 100, 150\}$, we calculate the Maximum Mean Discrepancy with the RBF kernel (RBF-MMD) on all predicted PCs and the 1-Wasserstein ($\mathcal{W}_1$) (102) and 2-Wasserstein ($\mathcal{W}_2$) (103) distances on the top-2 PCs.

This choice is made because the Wasserstein distance between empirical measures (i.e., discrete point clouds) is known to converge to the true Wasserstein distance between the underlying distributions at a rate that scales as $\varepsilon = \mathcal{O}(n^{-1/d})$ for some constant $C$, where $n$ is the number of samples and $d$ is the dimensionality of the state space. Therefore, an exponential number of samples is required to achieve the same estimation accuracy in high dimensions.

Computing the Wasserstein distance on the top-2 principal components provides a statistically stable and computationally tractable approximation that preserves the major axes of variation in the data. In addition, we evaluate the RBF-MMD metric on *all simulated PCs*, ensuring that higher-dimensional reconstruction accuracy is still quantitatively assessed.

Given the predicted distribution $p$ and true distribution $q$ and $n$ samples from each distribution $\{\boldsymbol{x}_i \sim p\}_{i=1}^n$ and $\{\mathbf{y}_i \sim q\}_{i=1}^n$, the RBF-MMD between $p$ and $q$ is calculated as

$$\text{MMD}(p, q) = \frac{1}{n^2} \sum_{i=1}^n \sum_{j=1}^n k_{\text{mix}}(\boldsymbol{x}_i, \boldsymbol{x}_j) + \frac{1}{n^2} \sum_{i=1}^n \sum_{j=1}^n k_{\text{mix}}(\mathbf{y}_i, \mathbf{y}_j) - \frac{2}{n^2} \sum_{i=1}^n \sum_{j=1}^n k_{\text{mix}}(\boldsymbol{x}_i, \mathbf{y}_j) \quad (104)$$

where $k_{\text{mix}}(\cdot, \cdot)$ is a mixture of RBF kernel functions defined as

$$k_{\text{mix}}(\boldsymbol{x}, y) = \frac{1}{|\Sigma|} \sum_{\sigma \in \Sigma} \exp\left(-\frac{\|\boldsymbol{x} - y\|_2^2}{2\sigma^2}\right) \quad (105)$$

where $\Sigma = \{0.01, 0.1, 1, 10, 100\}$ is the set of values that determine how much the distances between pairs of points are scaled when computing the overall discrepancy. The equations for 1-Wasserstein ($\mathcal{W}_1$) and 2-Wasserstein ($\mathcal{W}_2$) distances are provided in (102) and (103) respectively.

### E.6 COMPUTATIONAL COSTS

For each experiment, we report the GPU hours on a single NVIDIA A100 GPU, demonstrating that only a minor increase in compute time is required for increasing the number of branches.

Table 10: Total training time for each experiment on a NVIDIA A100 GPU for different dimensions and number of branches $K$.

| Experiment (Branches) | Total Training Time (min) |
|---|---|
| Mouse Hematopoiesis ($K = 1$) | 13m 4s |
| Mouse Hematopoiesis ($K = 2$) | 14m 3s |
| Clonidine 50D ($K = 1$) | 17m 22s |
| Clonidine 50D ($K = 2$) | 22m 50s |
| Clonidine 100D ($K = 2$) | 23m 3s |
| Clonidine 150D ($K = 2$) | 18m 31s |
| Trametinib ($K = 1$) | 4m 52s |
| Trametinib ($K = 3$) | 6m 44s |
| Pancreatic $\beta$-Cell ($K = 11$) | 75m 46s |

### E.7 HYPERPARAMETER SELECTION AND DISCUSSION

In this section, we present the hyperparameters used in each experiment. While the model architecture remained largely the same across experiments, we increased the hidden dimension to $1024$ for dimensions $d \in \{50, 100, 150\}$. For low-dimensional data, we found that increasing model complexity underperforms in comparison to lower hidden dimensions, and we established that a hidden dimension of $64$ achieves relatively optimal performance for $d \in \{2, 3\}$. While beyond the scope of this study, we believe that further exploration of diverse model architectures and hyperparameter tuning could improve the performance of BranchSBM. Exploration of diverse task-dependent state costs for novel applications is another exciting extension of our work.

Table 11: **Hyperparameter settings for different datasets.** The Clonidine perturbation experiment is split into three columns for each of the three dimensions of principal components (PCs) used $d \in \{50, 100, 150\}$.

| Parameter | Dataset | | | | | | |
|---|---|---|---|---|---|---|---|
| | LiDAR | Mouse Hematopoiesis scRNA | Clonidine 50PCs | 100PCs | 150PCs | Trametinib | Pancreatic $\beta$-Cell |
| branches | 2 | 2 | | 2 | | 3 | 11 |
| data dimension | 3 | 2 | 50 | 100 | 150 | 50 | 30 |
| batch size | 128 | 128 | | 32 | | 32 | 256 |
| $\lambda_{\text{energy}}$ | 1.0 | 1.0 | | 1.0 | | 1.0 | 1.0 |
| $\lambda_{\text{mass}}$ | 100 | 100 | | 100 | | 100 | 100 |
| $\lambda_{\text{match}}$ | $1.0 \times 10^3$ | $1.0 \times 10^3$ | | $1.0 \times 10^3$ | | $1.0 \times 10^3$ | $1.0 \times 10^3$ |
| $\lambda_{\text{recons}}$ | 1.0 | 1.0 | | 1.0 | | 1.0 | 1.0 |
| $\lambda_{\text{growth}}$ | 0.01 | 0.01 | | 0.01 | | 0.01 | 0.01 |
| $V_t$ | LAND | LAND | | RBF | | RBF | RBF |
| RBF $N_c$ | - | - | 150 | 300 | 300 | 150 | 300 |
| RBF $\kappa$ | - | - | 1.5 | 2.0 | 3.0 | 1.5 | 3.0 |
| hidden dimension | 64 | 64 | | 1024 | | 1024 | 1024 |
| lr $\varphi_{t,\eta}$ | $1.0 \times 10^{-4}$ | $1.0 \times 10^{-4}$ | | $1.0 \times 10^{-4}$ | | $1.0 \times 10^{-4}$ | $1.0 \times 10^{-4}$ |
| lr $u_t^\theta$ | $1.0 \times 10^{-3}$ | $1.0 \times 10^{-3}$ | | $1.0 \times 10^{-3}$ | | $1.0 \times 10^{-3}$ | $1.0 \times 10^{-3}$ |
| lr $g_t^\phi$ | $1.0 \times 10^{-3}$ | $1.0 \times 10^{-3}$ | | $1.0 \times 10^{-3}$ | | $1.0 \times 10^{-3}$ | $1.0 \times 10^{-3}$ |

## F  Training Algorithm

Here, we provide the pseudocode for BranchSBM's multi-stage training algorithm for stable optimization of the velocity and growth networks over the $K$ branched trajectories.

---
**Algorithm 1** Multi-Stage Training of **BranchSBM**

---
1: **Stage 1: Learning the Branched Neural Interpolants**
2: **while** Training **do**
3:     $\forall k, (\boldsymbol{x}_0, \boldsymbol{x}_{1,k}) \sim \pi^\star_{0,1,k}, t \sim \mathcal{U}(0,1)$
4:     **for** $k = 0$ to $K$ **do**
5:         $\boldsymbol{x}_{t,\eta,k} \leftarrow (1-t)\boldsymbol{x}_0 + t\boldsymbol{x}_{1,k} + t(1-t)\varphi_{t,\eta}(\boldsymbol{x}_0, \boldsymbol{x}_{1,k})$
6:         $\dot{\boldsymbol{x}}_{t,\eta,k} \leftarrow \boldsymbol{x}_1 - \boldsymbol{x}_0 + t(1-t)\dot{\varphi}_{t,\eta}(\boldsymbol{x}_0, \boldsymbol{x}_{1,k}) + (1-2t)\varphi_{t,\eta}(\boldsymbol{x}_0, \boldsymbol{x}_{1,k})$
7:         Compute $V_t(\boldsymbol{x}_{t,\eta,k})$ given the task-specific definition
8:         $\mathcal{L}_{\text{traj}}(\eta) \leftarrow \int_0^1 \left[ \frac{1}{2} \|\dot{\boldsymbol{x}}_{t,\eta,k}\|_2^2 + V_t(\boldsymbol{x}_{t,\eta,k}) \right] dt$
9:         Update $\varphi_{t,\eta}$ using gradient $\nabla_\eta \mathcal{L}_{\text{traj}}(\eta)$
10:     **end for**
11: **end while**
12: **Stage 2: Initial Training of Velocity Networks**
13: **while** Training **do**
14:     Initialize $K+1$ flow networks $\{u^\theta_{t,k}(X_{t,k})\}_{k=0}^K$
15:     **for** $k = 0$ to $K$ **do**
16:         Calculate $\boldsymbol{x}_{t,\eta,k}$ and $\dot{\boldsymbol{x}}_{t,\eta,k}$ with the trained network $\varphi^\star_{t,\eta}(\boldsymbol{x}_0, \boldsymbol{x}_{1,k})$
17:         $\mathcal{L}_{\text{flow}}(\theta) \leftarrow \|\dot{\boldsymbol{x}}_{t,\eta,k} - u^\theta_{t,k}(\boldsymbol{x}_{t,\eta,k})\|_2^2$
18:         Update $u^\theta_{t,k}$ using gradient $\nabla_\theta \mathcal{L}_{\text{flow}}(\theta)$
19:     **end for**
20: **end while**
21: **Stage 3: Initial Training of Growth Networks**
22: **while** Training **do**
23:     Freeze parameters of flow networks and initialize $K+1$ growth networks $\{g^\phi_{t,k}(X_{t,k})\}_{k=0}^K$
24:     **for** $t = 0$ to $1$ **do**
25:         **for** $k = 0$ to $K$ **do**
26:             $\boldsymbol{x}_{t,k} \leftarrow \int_0^t u^\theta_{s,k}(\boldsymbol{x}_{s,k}) ds$
27:             **if** $k = 0$ **then**
28:                 $w^\phi_{t,k} \leftarrow 1 + \int_0^t g^\phi_{t,k}(\boldsymbol{x}_{s,k}) ds$
29:             **else**
30:                 $w^\phi_{t,k} \leftarrow \int_0^t g^\phi_{t,k}(\boldsymbol{x}_{s,k}) ds$
31:             **end if**
32:             $\mathcal{L}_{\text{energy}}(\phi) \leftarrow \mathcal{L}_{\text{energy}}(\phi) + \int_t^{t+\Delta t} \left[ \frac{1}{2} \|u^\theta_{t,k}\|_2^2 + V_t(X_{t,k}) \right] w^\phi_{t,k}$
33:         **end for**
34:         $\mathcal{L}_{\text{mass}} \leftarrow \left( \sum_{k=0}^K w^\phi_{t,k} - w^{\text{total}} \right)^2$
35:     **end for**
36:     $\mathcal{L}_{\text{match}} \leftarrow \sum_{k=0}^K \left( w^\phi_{1,k}(\boldsymbol{x}_{1,k}) - w^\star_{1,k} \right)^2$
37:     $\mathcal{L}_{\text{recons}}(\theta) \leftarrow \sum_{k=0}^K \sum_{\boldsymbol{x}_{1,k} \in \mathcal{N}_n(\boldsymbol{x}_{1,k})} \max(0, \|\tilde{\boldsymbol{x}}_{1,k} - \boldsymbol{x}_{1,k}\|_2 - \epsilon)$
38:     $\mathcal{L}_{\text{growth}}(\phi) \leftarrow \lambda_{\text{energy}} \mathcal{L}_{\text{energy}}(\theta, \phi) + \lambda_{\text{match}} \mathcal{L}_{\text{match}}(\phi) + \lambda_{\text{mass}} \mathcal{L}_{\text{mass}}(\phi) + \lambda_{\text{growth}} \sum_{k=0}^K \|g^\phi_{t,k}\|_2^2$
39:     Update $g^\phi_{t,k}$ using gradient $\nabla_\phi \mathcal{L}_{\text{growth}}(\phi)$
40: **end while**
41: **Stage 4: Final Joint Training**
42: **while** Training **do**
43:     Unfreeze parameters of flow networks $\{u^\theta_{t,k}(X_{t,k})\}_{k=0}^K$
44:     Repeat steps of Stage 3 and calculate $\mathcal{L}_{\text{joint}}(\theta, \phi) \leftarrow \mathcal{L}_{\text{growth}}(\theta, \phi) + \mathcal{L}_{\text{recons}}(\theta)$
45:     Jointly update $u^\theta_{t,k}$ and $g^\phi_{t,k}$ for all branches using gradients $\nabla_\theta \mathcal{L}_{\text{joint}}(\theta, \phi)$ and $\nabla_\phi \mathcal{L}_{\text{joint}}(\theta, \phi)$
46: **end while**

---

## G    USE OF LARGE LANGUAGE MODELS (LLMS)

We acknowledge the use of large language models (LLMs) to assist in polishing and editing parts of this manuscript. LLMs were used to refine phrasing, improve clarity, and ensure consistency of style across sections. All technical content, experiments, analyses, and conclusions were developed by the authors, with LLM support limited to language refinement and editorial improvements.

