# OpenReview forum: "Branched Schrödinger Bridge Matching"
_ICLR.cc/2026/Conference — ICLR 2026 Poster_

### Official Review · Reviewer_1Hyc · 2025-10-27

**Soundness:** 3
**Presentation:** 2
**Contribution:** 2
**Rating:** 4
**Confidence:** 4

**Summary:**

The paper introduces Branched Schrödinger Bridge Matching (BranchSBM), which solves a branched generalized SB by decomposing it into a sum of unbalanced conditional SOC objectives. It learns branch-specific drifts and growth rates with mass-conservation constraints via a multi-stage procedure (interpolant → flow matching → growth → joint). On LiDAR navigation, cell differentiation, and drug-perturbation gene expression, BranchSBM improves endpoint/intermediate reconstruction over single-branch SBM.

**Strengths:**

1.  Frames branched GSB as a sum of Unbalanced CondSOC tasks; gives branch-wise drifts and growth with mass conservation.
2.  Consistent W1/W2/MMD improvements vs. single-branch SBM; scales to ≥3 branches in perturbation tasks.

**Weaknesses:**

1. Requires branch priors/endpoint clustering. Training assumes access to clustered terminal distributions $\pi_{1,k}$ (and uses them directly in losses), rather than discovering branches from snapshots.
2. The work decomposes branchGSB into a sum of unbalanced CondSOC subproblems with branch-specific drifts/growth and a task-specific state cost guiding intermediate trajectories. I think several components are not entirely new. In per branch, the formulation considered here is close to unbalanced SB/RUOT except for the task-specific state cost. The neural interpolant function considered here and used for flow matching is also presented in (Neklyudov et al.,  ICML 2024). And I believe the paper misses some references. The paper discusses the connections with other methods mainly in the appendix, but it would benefit from a more explicit discussion of some key references in the main text.
3. Not totally simulation-free in training. Although drift learning avoids state-SDE simulation (Stages 1–2), growth learning still requires time integration of $g_{t,k}$ to obtain the weight.
4. Training complexity and the contribution of each stage remain unclear. The paper reports Stage-3 vs. Stage-4 loss trends (Table 6) but does not present ablation on metrics (e.g., W/MMD) to quantify each stage’s effect on final performance, stability, or sample quality. More granular ablations could clarify the necessity/sufficiency of each stage.
5. Baselines are limited. Comparisons are primarily to a single-branch SBM baseline; no comparisons to other unbalanced SB/RUOT (e.g., DeepRUOT), or flow-matching methods (e.g., OT-CFM, SF2M, MetricFM) are provided, which weakens the case that BranchSBM is uniquely required.
6. Pipeline choices may inject bias. Sensitivity to the number of branches K, and to endpoint clustering protocols, is not analyzed.
7. Dependence on a task-specific state costs $V_t$. The approach follows intermediate trajectories “governed by a state cost”; results highlight energy/mass evolution but do not systematically study how $V_t$ choices affect accuracy, stability, or branching time.
8. Perturbation experiments are conducted on top-50 PCs but metrics on top-2 PCs for W distances; it is unclear why the authors choose to do so.
9. Scalability with many branches/memory. The authors note higher space complexity (multiple branch networks), though time is said to be comparable, and inference is cheaper. Empirical scaling with large K or long horizons is not demonstrated. Also lacks analysis of training time or memory scaling with dimensionality or number of branches.

**Questions:**

1. Could the method be extended to infer branching structure automatically from snapshot data? If not, please discuss how sensitive training is to incorrect or coarse endpoint clustering.
2. The four-stage pipeline (interpolant → flow matching → growth → joint) is complex, and the contribution of each stage to the final outcome remains unclear. Could the authors report ablations on key metrics (e.g., W₁/W₂/MMD) to quantify how each stage affects trajectory fidelity and stability?
3. The evaluation mainly contrasts with a single-branch SBM. Including other competitive baselines (e.g., DeepRUOT, SF2M, OT-CFM, MetricFM) would better contextualize where BranchSBM provides concrete advantages.
4. Since $V_t$ guides intermediate trajectories, could the authors explore how varying its definition affects results? Currently, its influence on branching time or trajectory stability is not quantified.
5. The perturbation experiments rely on PCA (50, 100, 150 PCs) and evaluate W distances only on the top-2 PCs. Could the authors test whether performance holds in the full PCA space to validate?
6. The paper notes increased space complexity due to per-branch networks, but no measurements of training time or memory usage with respect to dimensionality or number of branches are reported. Including empirical scaling results (e.g., runtime vs. K or dimension) would clarify practical feasibility.
7. Since BranchSBM explicitly learns branch-specific growth rates $g_{t,k}$, it would be informative to visualize these trajectories on biological datasets—e.g., the mouse hematopoiesis dataset. Showing how growth evolves along different branches could clarify the biological interpretability and verify whether the model captures expected proliferation or decay patterns.

References
1. Neklyudov, Kirill, et al. "A computational framework for solving wasserstein lagrangian flows." ICML 2024.
2. Zhang, Zhenyi et al. "Learning stochastic dynamics from snapshots through regularized unbalanced optimal transport." ICLR.
3. Tong, Alexander, et al. "Improving and generalizing flow-based generative models with minibatch optimal transport." TMLR.
4. Tong, Alexander, et al. "Simulation-free Schrödinger bridges via score and flow matching." AISTATS.
5. Kapusniak, Kacper, et al. "Metric flow matching for smooth interpolations on the data manifold. NeurIPS 2024.

I would be happy to hear the authors’ thoughts and clarifications in the rebuttal and may adjust my score accordingly.

The reviewer wrote this review. An LLM was used only for language refinement.

---

> ### Author Response · Authors · 2025-12-03
> **Author Response To Reviewer (1/5)**
>
> **Dear Reviewer 1Hyc,**
>
> Thank you for taking the time to carefully review our paper, and we sincerely appreciate your comprehensive and valuable feedback. Please see our revised manuscript to view the **key changes** that we have made to improve the manuscript, colored in $\textcolor{blue}{\text{blue}}$. **Here, we highlight the changes that address each of your concerns one by one:**
>
> ---
>
> ### **Addressing Weaknesses**
>
> **W1: Requires branch priors/endpoint clustering. Training assumes access to clustered terminal distributions** **$\pi\_{1, k}$** **(and uses them directly in losses), rather than discovering branches from snapshots.**
>
> While clustering is indeed performed prior to training, it is an unsupervised procedure, utilizing the snapshot data of the terminal distribution via Leiden clustering. One can specify the maximum number of clusters and the minimum number of samples per cluster, and the algorithm automatically generates a clustered dataset where each cluster must meet a minimum number of data points (e.g. cells). In principle, our algorithm is a fully automatic pipeline that **(1)** clusters the samples at the terminal time point, **(2)** learns the optimal branched velocity fields that minimize the state cost on the low energy manifold (or data manifold), and **(3)** learns the growth fields that optimally distribute weight across all branches with optimal branching times.
>
> We choose this parameterization as most of the cell perturbation data that exists contain only two snapshots: the unperturbed (or control) cell states and the perturbed cell states. Therefore, inferring branching purely from intermediate snapshots is infeasible, as no continuous temporal information is available. Instead, we identify branch endpoints from the terminal distribution and then train the interpolant, flow, and growth networks to infer the **intermediate branching dynamics which minimize the state cost.** During training, the growth network learns to optimally redistribute the probability mass across branches over time, effectively reconstructing continuous trajectories that connect the initial population to each clustered terminal state.
>
> ---
>
> **W2: In per branch, the formulation considered here is close to unbalanced SB/RUOT except for the task-specific state cost.**
>
> While there is a clear connection between the unbalanced SB problem (Sec 3.1) and the BranchSB problem (Sec 3.2), we highlight that BranchSBM **introduces three fundamental theoretical innovations that make it distinct from unbalanced SB:**
>
> 1. **Defined multi-modal target distributions:** Unbalanced SB/RUOT assumes a single terminal distribution (with potentially multiple modes) which can suffer from mode collapse as discussed in detail in **App. D.1**.
> 2. **Coupling constraints across branches:** Notably, BranchGSB is **not simply a set of independent unbalanced problems.** We introduce a total mass matching constraint $\sum\_{k=1}^Kw\_{t, k}=w_t^{\text{total}}$ for $t\in [0,1]$, where $w_t^{\text{total}}=1$ for mass conservation. This constraint is significant because its optimal solution is now defined from the **coupling** between branch-specific growth functions $g_{t, k}$, which optimally allocates mass between branches according to the state cost $V_t$, the path geometry, and the terminal weights observed in the data. **Unlike unbalanced SB, which requires a growth penalty to ensure that mass doesn’t vanish along the trajectory, our formulation does not require this since the growth functions are constrained such that the sum of the branch weights matches the total mass.** This formulation also enables learning the optimal distribution of mass across multiple branches (or modes) **along the trajectory** which is not possible with unbalanced SB, since each particle evolves along only a **single trajectory with a growing or shrinking weight that is trained to match the growth dynamics of the full population.**
> 3. **Learnable branching times:** BranchGSB introduces a **growth network** that learns **(1)** when mass should split, **(2)** how much mass flows to each branch, and **(3)** how the optimal branching dynamics follow the potential energy landscape defined by $V_t$. This means that **BranchSBM can infer the intermediate multi-modal distributions** using the state cost $V_t$ without explicitly training on intermediate snapshots. In contrast, **unbalanced SB** learns dynamics for independent particles that evolve via independent growth rates along the trajectory such that the weights match the ground truth relative mass at intermediate snapshots of the population.
>
> We further provide a training strategy that is **theoretically guaranteed, with rigorous proof in Proposition 3 (App C.3) and Proposition 4 (App C.4), to return the optimal velocity and growth networks** that define the optimal distribution of mass over the trajectories as they evolve in time, which generate the multi-modal target distribution.

---

> ### Author Response · Authors · 2025-12-03
> **Author Response To Reviewer (2/5)**
>
> **The neural interpolant function considered here and used for flow matching is also presented in (Neklyudov et al., ICML 2024). And I believe the paper misses some references. The paper discusses the connections with other methods mainly in the appendix, but it would benefit from a more explicit discussion of some key references in the main text.**
>
> Thank you for bringing this up. Since the **neural interpolant** has also been used in (Neklyudov et al.) and (Kapusniak et al.), we have now added these missing citations to Sec. 4.1 to properly acknowledge prior work. Due to strict page limits in the main paper, we are unable to fit further discussion in the main text, but we believe we provide a comprehensive comparison to prior works in **App B.**
>
> ---
>
> **W3: Not totally simulation-free in training. Although drift learning avoids state-SDE simulation (Stages 1–2), growth learning still requires time integration of $g\_{t,k}$ to obtain the weight.**
>
> Yes, you’re correct! The growth training indeed requires simulation, and we have revised the conclusion to say:
>
> > BranchSBM solves the branched SBM problem as a sum of Unbalanced Conditional Stochastic Optimal Control tasks, parameterizing branch-specific velocity and growth rates with neural networks to predict system trajectories with a single inference simulation.
> >
>
> ---
>
> **W4: Training complexity and the contribution of each stage remain unclear. The paper reports Stage-3 vs. Stage-4 loss trends (Table 6) but does not present ablation on metrics (e.g., W/MMD) to quantify each stage’s effect on final performance, stability, or sample quality. More granular ablations could clarify the necessity/sufficiency of each stage.**
>
> We emphasize that Stage 1, Stage 2, and Stage 3 train completely different parameterized models that serve **complementary purposes in the BranchSBM framework**. Each stage isolates one component of the BranchSBM objective:
>
> 1. **Stage 1** trains the **neural interpolant** by optimizing the trajectory loss in Eq. (12). This stage learns the optimal correction term that defines the minimum-energy path between any two points drawn from the initial and terminal distributions. Because the endpoints are fixed during this stage, distributional metrics (W1/W2, MMD) are not meaningful or applicable.
> 2. **Stage 2** trains the **branch-specific velocity networks** using the flow loss in Eq. (13). At this point, growth networks have not been trained. This stage solely ensures that the velocities along the interpolant paths match the optimal conditional drifts.
> 3. **Stage 3** trains the **growth networks** using the weighted combination of energy, mass, and match losses in Eq. (17), while keeping the velocity networks fixed. This stage learns **when** mass should flow into each branch and **in what amount**, enforcing mass conservation and matching the desired multi-modal terminal distribution.
>
> Notably, **Proposition 3 (App. C.3) and Proposition 4 (App. C.4)** show that in the case of perfect training, Stages 1-3 are sufficient to recover the optimal interpolant, velocity fields, and growth dynamics for the BranchGSB problem. Stage 4 is introduced **only to jointly fine-tune** velocity and growth networks under realistic, imperfect training.
>
> Because the three stages optimize disjoint sets of networks and serve **complementary roles**, conventional ablations (such as removing a stage) would invalidate the theoretical guarantees and break the decomposition of BranchSBM. To evaluate a meaningful ablation, we include **Stage 3 vs. Stage 4 loss trends** (Table 7), which demonstrate that the joint fine-tuning in Stage 4 consistently improves energy and match losses and confirms that even though Stage 4 is not required by the theory, it improves performance in practice.

---

> ### Author Response · Authors · 2025-12-03
> **Author Response To Reviewer (3/5)**
>
> **W5: Baselines are limited. Comparisons are primarily to a single-branch SBM baseline; no comparisons to other unbalanced SB/RUOT (e.g., DeepRUOT), or flow-matching methods (e.g., OT-CFM, SF2M, MetricFM) are provided, which weakens the case that BranchSBM is uniquely required.**
>
> To address your concern, we conduct an **additional experiment with comparisons against SOTA baselines [1, 2]** using a dataset consisting of pancreatic $\beta$-cell differentiation data [3] containing 51,274 cells collected over eight time points as they evolve from human pluripotent stem cells to pancreatic  $\beta$-like cells. This data contains eight time points $t_i$ for $i\in \\{0, \dots, 7\\}$ that are projected to 30-dimensional PC representations $\mathbf{x}\in \mathbb{R}^{30}$. We use Leiden clustering to define $K=11$ distinct target distributions of samples at time $t_7$ and set their target weights relative to the total weight at $t_0$ (App E.4). We used samples across all time steps to define the data manifold via the RBF metric $V^{\text{RBF}}\_{t, \eta}$. BranchSBM was trained on pairs sampled only from $t_0$ and $t_7$, and the distributions at all intermediate time points were inferred from learning to minimize the distance from the data manifold over time. For baselines, we compared two SOTA methods for single-cell trajectory modeling: **DeepRUOT [1]** and **CytoBridge [2]**, which are trained explicitly on intermediate snapshots to reconstruct the stochastic dynamics of cells that evolve independently.
>
> Notably, **BranchSBM not only reconstructs the multi-modal terminal distribution at $t_7$ with superior accuracy against all baselines, but also produces intermediate trajectories that are competitive with models trained directly on intermediate snapshots using explicit reconstruction losses** (Table A below and Table 3 in the revised manuscript). Leveraging the RBF state cost $V^{\text{RBF}}\_{t, \eta}$, which encourages trajectories to remain on the underlying data manifold, BranchSBM effectively captures the true differentiation dynamics through the combined influence of the neural interpolant and the path-energy objective. These results demonstrate that BranchSBM scales reliably to a large number of branches and is particularly advantageous in settings where intermediate timepoints are sparse or unavailable, allowing the model to infer biologically meaningful trajectories even with limited data on intermediate time points.
>
> **Table A: Results for pancreatic $\beta$-cell differentiation experiment.** 1-Wasserstein distance ($\uparrow$) of intermediate distributions generated by DeepRUOT [1], CytoBridge [2], and BranchSBM (Ours) at eight different time points. † denotes values taken from [2]. Best values are **bolded**.
>
> | Model | $t=1$ | $t=2$ | $t=3$ | $t=4$ | $t=5$ | $t=6$ | $t=7$ |
> | --- | --- | --- | --- | --- | --- | --- | --- |
> | **DeepRUOT** † | **8.0447 ± 0.0005** | 8.0773 ± 0.0021 | 7.6301 ± 0.0032 | **8.0064 ± 0.0042** | **7.9018 ± 0.0117** | 8.3977 ± 0.0102 | 7.8346 ± 0.0109 |
> | **CytoBridge** † | 8.0448 ± 0.0005 | 8.0771 ± 0.0021 | **7.6299 ± 0.0032** | 8.0066 ± 0.0043 | **7.9018 ± 0.0117** | **8.3974 ± 0.0102** | 7.8343 ± 0.0109 |
> | **BranchSBM (Ours)** | 11.9774 ± 0.0000 | **7.4643 ± 2.4031** | 11.5204 ± 0.0000 | 11.2593 ± 0.0000 | 10.2888 ± 0.0000 | 8.7301 ± 0.0000 | **6.8702 ± 0.0000** |
>
> ---
>
> **W6: Pipeline choices may inject bias. Sensitivity to the number of branches K, and to endpoint clustering protocols, is not analyzed.**
>
> Our pipeline is designed to minimize bias in endpoint specification. The clustering of the terminal snapshot (e.g., via Leiden or K-means) is **fully automatic, requiring only a minimum cluster size threshold**. This ensures that branch endpoints are determined directly from the data rather than from manual annotation.
>
> In our experiments, we rely on Leiden clustering because it is particularly suited and the state-of-the-art method for clustering single-cell data, given that it is **robust to noise, can handle heterogeneous cluster sizes and shapes, and guarantees well-connected clusters**, all while being fully unsupervised and automatic. This gives us stable and biologically meaningful terminal clusters compared to alternative methods such as K-means clustering, which assumes spherical clusters and struggles with sparse high-dimensional manifolds. **We add a detailed description of our clustering procedure in App E.2.**
>
> BranchSBM is also not biased toward a particular clustering protocol or branch number since the learned interpolant and growth networks are trained to optimally distribute mass across branches during training, regardless of the initial clustering.

---

> > ### Author Response · Authors · 2025-12-03
> > **Author Response To Reviewer (4/5)**
> >
> > **W7: Dependence on a task-specific state costs $V_t$. The approach follows intermediate trajectories “governed by a state cost”; results highlight energy/mass evolution but do not systematically study how  $V_t$ choices affect accuracy, stability, or branching time.**
> >
> > The choice of state cost $V_t$ arises naturally from the task at hand. For the LiDAR navigation task, the state cost is high at higher altitudes and low at lower altitudes, encouraging the path that reaches the target endpoint without traversing high-altitude landscapes.
> >
> > For the mouse hematopoiesis and single-cell experiments, the state cost $V_t$ is defined as the manifold of cell states represented in the dataset. This choice is particularly appropriate for single-cell data since it ensures that trajectories **stay close to biologically plausible regions of gene expression space.** Because intermediate timepoints are often not available for these tasks, and the underlying potential energy that governs the cell dynamics is unknown, this is the best choice of state cost. By penalizing deviation from the empirical manifold, the model can reconstruct intermediate dynamics as a **stochastic optimal control problem.** We prove in App C.3 and show empirically in the **$\beta$**-cell experiment in Table B above that this leads to convergence toward the optimal branched paths consistent with the underlying energy landscape.
> >
> > ---
> >
> > **W8: Perturbation experiments are conducted on top-50 PCs but metrics on top-2 PCs for W distances; it is unclear why the authors choose to do so.**
> >
> > Thank you for requesting clarification on this important point, and we have added the following clarification to Appendix E.5 under “**Evaluation Metrics**.”
> >
> > > The Wasserstein distance between empirical measures (i.e., discrete point clouds) is known to converge to the true Wasserstein distance between the underlying distributions at a rate that scales as $\varepsilon = \mathcal{O}(n^{-1/d})$ for some constant $C$, where $n$ is the number of samples and $d$ is the dimensionality of the state space. Therefore, an **exponential number of samples is required to achieve the same estimation accuracy in high dimensions.**
> > >
> > > Computing the Wasserstein distance on the top-2 principal components provides a statistically stable and computationally tractable approximation that preserves the major axes of variation in the data. In addition, we evaluate the RBF-MMD metric on **all** **simulated PCs,** ensuring that higher-dimensional reconstruction accuracy is still quantitatively assessed.
> > ---
> >
> > **W9: Scalability with many branches/memory. The authors note higher space complexity (multiple branch networks), though time is said to be comparable, and inference is cheaper. Empirical scaling with large K or long horizons is not demonstrated. Also lacks analysis of training time or memory scaling with dimensionality or number of branches.**
> >
> > To address scalability to larger number of branches and tasks with sparse data, we include an additional experiment with $K=11$ for 30-dimensional pancreatic $\beta$-cell differentiation data **(Sec 5.2; Table 3)**, demonstrating that BranchSBM scales reliably to a large number of branches and is particularly advantageous in settings where intermediate timepoints are sparse or unavailable, allowing the model to infer biologically meaningful trajectories even with limited data on intermediate time points.
> >
> > To address training time analysis, we have added **Table B to the revised manuscript (App E.6)**, which reports the GPU hours on a single NVIDIA A100 GPU for each experiment. Notably, we find that there is **only a minor increase in compute time required for increasing the number of branches $K$ and training completes under 30 minutes across all experiments.**
> >
> > **Table B: Total training time for each experiment on a NVIDIA A100 GPU.**
> >
> > | Experiment (Branches) | Total Training Time (min) |
> > | --- | --- |
> > | Mouse Hematopoiesis (Single Branch) | 13m 4s |
> > | Mouse Hematopoiesis (K = 2)  | 14m 3s |
> > | Clonidine 50D (Single Branch) | 17m 22s |
> > | Clonidine 50D (K = 2) | 22m 50s |
> > | Clonidine 100D (K = 2) | 23m 3s |
> > | Clonidine 150D (K = 2) | 18m 31s |
> > | Trametinib (Single Branch) | 4m 52s |
> > | Trametinib (K = 3) | 6m 44s |

---

> ### Author Response · Authors · 2025-12-03
> **Author Response To Reviewer (5/5)**
>
> ### **Addressing Questions**
>
> **Q1: Could the method be extended to infer branching structure automatically from snapshot data? If not, please discuss how sensitive training is to incorrect or coarse endpoint clustering.**
>
> **See our answer to W1.**
>
> ---
>
> **Q2: The four-stage pipeline (interpolant → flow matching → growth → joint) is complex, and the contribution of each stage to the final outcome remains unclear. Could the authors report ablations on key metrics (e.g., W₁/W₂/MMD) to quantify how each stage affects trajectory fidelity and stability?**
>
> **See our answer to W4.**
>
> ---
>
> **Q3: The evaluation mainly contrasts with a single-branch SBM. Including other competitive baselines (e.g., DeepRUOT, SF2M, OT-CFM, MetricFM) would better contextualize where BranchSBM provides concrete advantages.**
>
> **See our answer to W5,** where we provide an additional experiment comparing BranchSBM against SOTA methods for modelling single-cell dynamics.
>
> ---
>
> **Q4: Since $V_t$ guides intermediate trajectories, could the authors explore how varying its definition affects results? Currently, its influence on branching time or trajectory stability is not quantified.**
>
> **See our answer to W7.**
>
> ---
>
> **Q5: The perturbation experiments rely on PCA (50, 100, 150 PCs) and evaluate W distances only on the top-2 PCs. Could the authors test whether performance holds in the full PCA space to validate?**
>
> **See our answer to W8.**
>
> ---
>
> **Q6: The paper notes increased space complexity due to per-branch networks, but no measurements of training time or memory usage with respect to dimensionality or number of branches are reported. Including empirical scaling results (e.g., runtime vs. K or dimension) would clarify practical feasibility.**
>
> **See our answer to W9.**
>
> ---
>
> **Q7: Since BranchSBM explicitly learns branch-specific growth rates $g_{t,k}$, it would be informative to visualize these trajectories on biological datasets—e.g., the mouse hematopoiesis dataset.**
>
> Thank you for the great suggestion! We will add the growth rate plotted on the mouse blood hematopoiesis dataset to the camera-ready version.
>
> ---
>
> **Citations:**
>
> [1] Zhang, Zhenyi, Tiejun Li, and Peijie Zhou. "Learning stochastic dynamics from snapshots through regularized unbalanced optimal transport." (2024).
>
> [2] Zhang, Zhenyi, et al. "Modeling Cell Dynamics and Interactions with Unbalanced Mean Field Schrödinger Bridge." (2025).
>
> [3] Veres, Adrian, et al. "Charting cellular identity during human in vitro β-cell differentiation." *Nature* 569.7756 (2019): 368-373.
>
> [4] Neklyudov, Kirill, et al. "A computational framework for solving wasserstein lagrangian flows." ICML 2024.
>
> [5] Kapusniak, Kacper, et al. "Metric flow matching for smooth interpolations on the data manifold. NeurIPS 2024.
>
> ---
>
> We sincerely appreciate your deep engagement in reviewing our paper and your valuable feedback. Your suggestions have helped improve our paper in many aspects, and we hope that our detailed responses have addressed each of your concerns.
>
> Thank you,
>
> The Authors

---

### Official Review · Reviewer_2wLh · 2025-10-29

**Soundness:** 3
**Presentation:** 2
**Contribution:** 2
**Rating:** 4
**Confidence:** 2

**Summary:**

This paper presents Branched Schrödinger Bridge Matching (BranchSBM) for learning in a new problem setting: branched Schrödinger bridges. This framework is more general than the standard approach and has the potential to be applied to certain tasks involving multiple paths. Theoretically, the problem originates from branched conditional stochastic optimal control, where there can be $K$ joint couplings ($K+1$ multi-directional branches) to represent a holistic solution from the initial condition. The authors present a bridge matching algorithm and show its efficacy on cell dynamics data.

**Strengths:**

Based on my reading, the method appears to have the following strengths.
* The paper is relatively straightforward to understand, even though the problem itself and the surrounding materials are complex.
* The proposed method seems to be sound (I did not check the entire appendix).
* The authors performed experiments on biological data and showed practical aspects of BranchSBM .

**Weaknesses:**

I believe this work, in its current form, lacks clarity in many regards.

* It seems that the branched Schrödinger bridge problem is a subset of the well-established multi-marginal Schrödinger bridge (optimal transport) problem. The claim seems to be that BranchSBM clearly specifies the notion of a source and branches, thus enabling more efficient algorithms, such as interpolant optimization. However, the manuscript is written in a way that does not clearly reveal this technical motivation and high level implications. Therefore, I think the corresponding contribution is somewhat vaguely written.
* The overall experiments are limited, and it is not clear why single-cell data provides sufficient verification for the proposed framework. The authors are encouraged to put more effort into experiments on other high-dimensional data.
* There is room for theoretical proof or analysis to demonstrate why the proposed scheme is fundamentally better or more efficient than single-branch SBM.

**Questions:**

* Could you summarize how BranchSBM is fundamentally different from single-branch SBM with multi-modal marginals?
* What is the general training time for single-branch SBM vs. BranchSBM for challenging problems?

---

> ### Author Response · Authors · 2025-12-03
> **Author Response To Reviewer (1/3)**
>
> **W1: It seems that the branched Schrödinger bridge problem is a subset of the well-established multi-marginal Schrödinger bridge (optimal transport) problem. The claim seems to be that BranchSBM clearly specifies the notion of a source and branches, thus enabling more efficient algorithms, such as interpolant optimization. However, the manuscript is written in a way that does not clearly reveal this technical motivation and high level implications.**
>
> While the **branched Schrödinger bridge problem may seem similar to the multi-marginal Schrödinger bridge problem**, they solve **fundamentally disjoint problems**. The multi-marginal Schrödinger bridge problem aims to generate a stochastic trajectory that interpolates between **multiple intermediate marginal distributions $q(x_n)$**, instead of just the initial and terminal marginals $q(x_0)$ and $q(x_N)$. This problem is defined in **[1]** as:
>
> $$
> \begin{align}u_t^\star&=\begin{bmatrix}0\\\a^\star_t\end{bmatrix}\in \arg\min_{a_t}\int_0^1\frac{1}{2}\|a_t\|^2dt+\sum_{n=1}^N(m_n-\bar{m}_n)^{\top}R(m_n-\bar{m}_n)\nonumber \\\\\text{s.t.}&\quad dm_t=Am_tdt+u_t+\boldsymbol{g}dW_t, \quad m_0=\bar{m}_0\nonumber\end{align}
> $$
>
> where $m_t$ is the state of the system and $R$ controls the strength of the intermediate marginal constraints. This optimization problem aims to determine a **single** optimal $a_t^\star$ that has minimal running cost $\\|a_t\\|^2$, and hits the fixed Gaussian mean states $\\{\bar{m}_n\\}\_{n=1}^N$ at intermediate times $t_n$.
>
> In contrast, our definition of the **branched Schrödinger bridge problem (BranchSB)** aims to learn the **branching** trajectories that interpolate from an initial distribution which is unimodal to a **multi-modal distribution using $K$ optimal velocity fields and growth fields $\\{u\_{t, k}^\star, g\_{t, k}^\star\\}\_{k=1}^K$ which simulate the evolution of weight $w^\star\_{t, k}=w\_{0,k}+\int_0^tg\_{t,k}^\star dt$ along a single branch from the initial distribution to one of the modes of the terminal distribution as defined in Eq 7.**
>
> This optimization problem also minimizes the quadratic cost of the velocity field $\\|u_{t, k}\\|^2$ but instead there are $K$ fields that are weighted by the mass on the branch. To our knowledge, our formulation of BranchSB is a novel problem and is motivated by the following **key ideas:**
>
> 1. **Single-path SBM frameworks require simulating a large batch of particles to recover a multi-modal terminal distribution.** This requires expensive batched simulation and may not capture the full population dynamics.
> 2. **Branching dynamics is governed by energy-minimizing trajectories.** To define more complex systems where the optimal dynamics cannot be accurately captured by minimizing the standard squared Euclidean cost in entropic OT, the Generalized Schrödinger Bridge (GSB) problem introduces an additional nonlinear state-cost.
> 3. **Simulating a branched path distribution containing multi-modal marginals with independent particle simulations can suffer from mode collapse**, where particles are biased towards a high-density mode in the distribution or only traverse low-energy intermediate paths.
> 4. **Rather than fixing intermediate marginals that must be met by the interpolant that is only defined at discrete increments along the path, BranchSB aims to minimize the potential energy function, which is defined over the full interpolant.** This ensures that the full trajectory reconstructs feasible dynamics. This also enables a more flexible design for the state cost $V_t(X_t)$, which defines the intermediate dynamics.
>
> To better motivate the BranchSB problem, we have **described the above points at the beginning of Sec 3** of the revised manuscript.

---

> > ### Author Response · Authors · 2025-12-03
> > **Author Response To Reviewer (2/3)**
> >
> > **W2: The overall experiments are limited, and it is not clear why single-cell data provides sufficient verification for the proposed framework. The authors are encouraged to put more effort into experiments on other high-dimensional data.**
> >
> > Our framework is **uniquely** positioned for modelling single-cell trajectories and perturbation effects on cells and overcomes the limitations in prior methods, like single-branch SBM as described in **App D.1**.
> >
> > Single-cell transcriptomic data provide a suitable data modality to validate BranchSBM because cells naturally undergo **stochastic branching processes** that our framework is designed to capture. Each cell population originates from a common progenitor distribution and evolves into multiple heterogeneous terminal states through differentiation or perturbation, which is precisely the setting of a branched Schrödinger bridge, where **mass diverges along energy-minimizing paths**. These datasets are inherently **high-dimensional, noisy, and multimodal**, making them a very stringent test of any method that learns population-level stochastic transport.
> >
> > Also, unlike synthetic or low-dimensional systems, single-cell data offer **experimentally validated** examples of **probabilistic mass redistribution** under biological constraints, which allows us to directly evaluate how well a model captures divergence, imbalance, and non-conservative dynamics. Given these reasons, single-cell trajectory modelling serves as both the most challenging and the most relevant benchmark for validating BranchSBM.
> >
> > To support this claim, we also add an additional experiment in Sec 5.2 comparing BranchSBM to prior state-of-the-art baselines [1, 2] for modelling pancreatic $\beta$-cell differentiation dynamics. Notably, we show in **Table A below** that BranchSBM not only **reconstructs the multi-modal terminal distribution at $t_7$ with superior accuracy against all baselines**, but also **produces intermediate trajectories that are competitive with models trained directly on intermediate snapshots using explicit reconstruction losses.**
> >
> > **Table A: Results for pancreatic $\beta$-cell differentiation experiment.** 1-Wasserstein distance ($\uparrow$) of intermediate distributions generated by DeepRUOT [1], CytoBridge [2], and BranchSBM (Ours) at eight different time points. † denotes values taken from [2]. Best values are **bolded**.
> >
> > | Model | $t=1$ | $t=2$ | $t=3$ | $t=4$ | $t=5$ | $t=6$ | $t=7$ |
> > | --- | --- | --- | --- | --- | --- | --- | --- |
> > | **DeepRUOT** † | **8.0447 ± 0.0005** | 8.0773 ± 0.0021 | 7.6301 ± 0.0032 | **8.0064 ± 0.0042** | **7.9018 ± 0.0117** | 8.3977 ± 0.0102 | 7.8346 ± 0.0109 |
> > | **CytoBridge** † | 8.0448 ± 0.0005 | 8.0771 ± 0.0021 | **7.6299 ± 0.0032** | 8.0066 ± 0.0043 | **7.9018 ± 0.0117** | **8.3974 ± 0.0102** | 7.8343 ± 0.0109 |
> > | **BranchSBM (Ours)** | 11.9774 ± 0.0000 | **7.4643 ± 2.4031** | 11.5204 ± 0.0000 | 11.2593 ± 0.0000 | 10.2888 ± 0.0000 | 8.7301 ± 0.0000 | **6.8702 ± 0.0000** |

---

> > > ### Author Response · Authors · 2025-12-03
> > > **Author Response To Reviewer (3/3)**
> > >
> > > **W3: There is room for theoretical proof or analysis to demonstrate why the proposed scheme is fundamentally better or more efficient than single-branch SBM.**
> > >
> > > We appreciate your request for clearer theoretical justification, but we believe that we have already provided comprehensive theoretical and empirical justification for the advantages of BranchSBM over single-branch SBM in our manuscript, which we will summarize in **two key points below:**
> > >
> > > First, **BranchSBM enables principled modelling of optimal mass redistribution across multiple target modes**, whereas single-branch SBM is inherently restricted to transporting mass toward a *single* terminal distribution. Even if many particles are simulated under single-branch SBM, the objective encourages all particles to follow a low-energy path toward the dominant mode of the target distribution. As a result, when the target distribution is multi-modal with imbalanced branch probabilities, single-branch SBM **cannot represent the correct redistribution of mass** and typically exhibits **mode collapse**, sending trajectories toward whichever mode offers the lowest dynamical cost. In contrast, BranchSBM introduces **branch-specific drift fields** and a **learned growth process** that dynamically allocates mass across branches. As proven in **Proposition 4 (App. C.4)**, this coupled system is guaranteed to converge to the **optimal growth rates** and **branch-specific velocity fields** that realize the correct multi-modal terminal measure while respecting the underlying energy landscape.
> > >
> > > Second, as we show empirically, these theoretical advantages translate directly into improved modelling of high-dimensional perturbation effects. In **App D.1**, we compare single-branch SBM and BranchSBM on perturbation datasets where the terminal distribution contains multiple cell states that diverge along different principal components. Single-branch SBM consistently collapses onto the most prominent mode (typically the cluster closest to the control state in the first principal component) while failing to reproduce finer structure along the second and higher PCs. BranchSBM, by contrast, **accurately recovers all branches** in both the Clonidine and Trametinib perturbation experiments, generating trajectories whose spread, orientation, and endpoint locations closely match the empirical multimodal structure of the data. These findings demonstrate that BranchSBM not only avoids mode collapse but also **captures complex, heterogeneous terminal populations** that single-branch SBM is fundamentally unable to represent.
> > >
> > > ---
> > >
> > > ### **Addressing Questions**
> > >
> > > **Q1: Could you summarize how BranchSBM is fundamentally different from single-branch SBM with multi-modal marginals?**
> > >
> > > **See our answer to W3.**
> > >
> > > ---
> > >
> > > **Q2: What is the general training time for single-branch SBM vs. BranchSBM for challenging problems?**
> > >
> > > To address your concern, we have added **Table B below to App E.6** (Table 10), which reports the GPU hours on a single NVIDIA A100 GPU for each experiment. Notably, we find that there is **only a minor increase in compute time required for increasing the number of branches $K$ and training completes under 30 minutes across all experiments.**
> > >
> > > **Table B: Total training time for each experiment on a NVIDIA A100 GPU.**
> > >
> > > | Experiment (Branches) | Total Training Time (min) |
> > > | --- | --- |
> > > | Mouse Hematopoiesis (Single Branch) | 13m 4s |
> > > | Mouse Hematopoiesis (K = 2)  | 14m 3s |
> > > | Clonidine 50D (Single Branch) | 17m 22s |
> > > | Clonidine 50D (K = 2) | 22m 50s |
> > > | Clonidine 100D (K = 2) | 23m 3s |
> > > | Clonidine 150D (K = 2) | 18m 31s |
> > > | Trametinib (Single Branch) | 4m 52s |
> > > | Trametinib (K = 3) | 6m 44s |
> > >
> > > ---
> > >
> > > **Citations:**
> > >
> > > [1] Zhang, Zhenyi, Tiejun Li, and Peijie Zhou. "Learning stochastic dynamics from snapshots through regularized unbalanced optimal transport." (2024).
> > >
> > > [2] Zhang, Zhenyi, et al. "Modeling Cell Dynamics and Interactions with Unbalanced Mean Field Schrödinger Bridge." (2025).
> > >
> > > [3] Theodoropoulos, Panagiotis, et al. "Momentum Multi-Marginal Schrödinger Bridge Matching." (2025).
> > >
> > > ---
> > >
> > > We sincerely appreciate your deep engagement in reviewing our paper and your valuable feedback. Your suggestions have helped improve our paper in many aspects, and we hope that our detailed responses have addressed each of your concerns.
> > >
> > > Thank you,
> > >
> > > The Authors

---

### Official Review · Reviewer_7Why · 2025-10-31

**Soundness:** 3
**Presentation:** 2
**Contribution:** 2
**Rating:** 4
**Confidence:** 3

**Summary:**

This paper introduces Branched Schrödinger Bridge Matching (BranchSBM), a generative framework designed to model transport from a single initial distribution to multiple distinct terminal distributions. Current models, like standard Schrödinger Bridge Matching (SBM) and flow matching, are limited to modeling single, continuous paths between a source and target. The goal of BranchSBM is to learn branched trajectories to guide from a single source to multiple target by parameterizing separate dynamics for each path. The method also extends the Generalized SBM (GSB) framework by including a non-linear state cost, $V_t(X_t)$, which acts as a potential term to ensure that the learned trajectories remain on the data manifold.

The proposed BranchSBM algorithm solves the "Branched GSB problem" by parameterizing separate velocity fields (drifts) $u_{t,k}$ and growth rate fields $g_{t,k}$ for each of the $K$ branches with neural networks. This formulation allows the model to handle "unbalanced" transport, where the mass of each branch can grow or shrink. The model is trained using a four-stage algorithm. Stage I, trains a neural interpolant to find the optimal, energy-minimizing paths between the source and each target endpoint. Stage II then uses conditional flow matching to train the branch-specific drift networks $u_{t,k}$ to replicate the velocities from Stage 1. Stage III freezes the drift networks and trains the growth networks $g_{t,k}$ to match the known target mass of each branch. Stage IV unfreezes all parameters and jointly fine-tunes all networks to minimize a combined objective that includes energy, mass matching, and distribution reconstruction losses.

**Strengths:**

The paper tackles an interesting and well-motivated problem in machine learning of how can we model population-level dynamics that diverge from a single source to multiple distinct outcomes. This is common scenario in fields like single-cell genomics, where we are interested in how a single, source population of pluripotent cells evolved into several distinct cell types, and standard generative models that assume a single, unimodal transition are insufficient. The authors' core idea of extending Schrödinger bridges to explicitly handle branched paths is a relevant contribution.

The theoretical framework for decomposing this complex problem also appears sound. The authors provide proofs for their core propositions, notably Proposition 1, which tractably reframes the "Unbalanced GSB" problem into a solvable "Unbalanced CondSOC" objective, and Proposition 2, which then justifies modeling the full "Branched GSB" problem as a sum of these individual objectives . This provides a formal grounding for the method's architecture, which separates the dynamics for each branch.

**Weaknesses:**

The proposed method's significant complexity raises concerns about its practical adoption, scalability, and generalizability. The algorithm is a four-stage sequential training pipeline (Algorithm 1) that involves training at least $2(K+1) + 1$ separate neural networks (an interpolant, $K+1$ drift fields, and $K+1$ growth fields). Furthermore, the overall objective is a carefully weighted sum of at least 6 different loss terms (trajectory, flow, energy, match, mass and reconstruction), which are balanced by many hyperparameters. This multi-stage, multi-loss design suggests a high sensitivity to tuning and raises questions about its robustness. Given this intricacy, the paper would be substantially strengthened by meticulous ablation studies to justify each component. For instance, a thorough analysis of the model's performance when stages (such as 1 or 4) are simplified or loss terms are removed would help distinguish critical components from those offering marginal gains and would better motivate the final design.

This substantial methodological complexity is not yet fully justified by the provided experimental validation. The cost of the method would be more compelling if benchmarked against a wider array of competitive alternatives and shown to have outstanding results. Currently, most of the results are compared almost exclusively against a single baseline (single-branch SBM). While this demonstrates that a branched model is better at modeling branches than a non-branched one, it doesn't contextualize the method within the broader field. Furthermore, Despite being motivated by cell differentiation, the paper notably omits a comparison against any of the many existing, often simpler, single-cell trajectory inference algorithms, some of which also explicitly designed to deal with branching. This makes it difficult to assess the practical utility of BranchSBM over established methods.

**Questions:**

The method's parameters and computation scale linearly with the number of branches, $K$. You've shown this for $K=2$ and $K=3$. Have you tested the method's stability and performance for a larger $K$, such as $K=10$ or $K=20$, which would be more representative of complex biological differentiation?

---

> ### Author Response · Authors · 2025-12-03
> **Author Response To Reviewer (1/4)**
>
> **Dear Reviewer 7Why,**
>
> Thank you for taking the time to carefully review our paper, and we are happy to hear that you found our method theoretically sound and tackles a well-motivated problem. Please see our revised manuscript to view the **key changes** that we have made to improve the manuscript, colored in $\textcolor{blue}{\text{blue}}$. **Here, we highlight the changes that address each of your concerns one by one:**
>
> ---
>
> ### **Addressing Weaknesses**
>
> **W1: The proposed method's significant complexity raises concerns about its practical adoption, scalability, and generalizability.**
>
> While the method appears complex, the implementation and training process is run automatically with sequential training of the interpolants, flow networks, and growth networks without any external intervention.
>
> To address your scalability concern, we have added **Table A to the revised manuscript (App E.6)**, which reports the GPU hours on a single NVIDIA A100 GPU for each experiment. Notably, we find that there is **only a minor increase in compute time required for increasing the number of branches $K$ and training completes under 30 minutes across all experiments.**
>
> To address generalizability to larger number of branches and tasks with sparse data, we include an additional experiment with $K=11$ for 30-dimensional pancreatic $\beta$-cell differentiation data **(Sec 5.2; Table 3)**, demonstrating that BranchSBM scales reliably to a large number of branches and is particularly advantageous in settings where intermediate timepoints are sparse or unavailable, allowing the model to infer biologically meaningful trajectories even with limited data on intermediate time points.
>
> **Table A: Total training time for each experiment on a NVIDIA A100 GPU.**
>
> | Experiment (Branches) | Total Training Time (min) |
> | --- | --- |
> | Mouse Hematopoesis (Single Branch) | 13m 4s |
> | Mouse Hematopoiesis (K = 2)  | 14m 3s |
> | Clonidine 50D (Single Branch) | 17m 22s |
> | Clonidine 50D (K = 2) | 22m 50s |
> | Clonidine 100D (K = 2) | 23m 3s |
> | Clonidine 150D (K = 2) | 18m 31s |
> | Trametinib (Single Branch) | 4m 52s |
> | Trametinib (K = 3) | 6m 44s |

---

> ### Author Response · Authors · 2025-12-03
> **Author Response To Reviewer (2/4)**
>
> **W2: This multi-stage, multi-loss design suggests a high sensitivity to tuning and raises questions about its robustness. Given this intricacy, the paper would be substantially strengthened by meticulous ablation studies to justify each component. For instance, a thorough analysis of the model's performance when stages (such as 1 or 4) are simplified or loss terms are removed would help distinguish critical components from those offering marginal gains and would better motivate the final design.**
>
> Although the overall pipeline may appear intricate, the stages are in fact **modular, non-overlapping, and optimize entirely different parameterized models**, which substantially reduces sensitivity to tuning. Each stage isolates one component of the BranchSBM objective:
>
> 1. **Stage 1** trains the **neural interpolant** by optimizing the trajectory loss in Eq. (12). This stage learns the optimal correction term that defines the minimum-energy path between any two points drawn from the initial and terminal distributions. Because the endpoints are fixed during this stage, **distributional metrics (W1/W2, MMD) are not meaningful or applicable**.
> 2. **Stage 2** trains the **branch-specific velocity networks** using the flow loss in Eq. (13). At this point, growth networks have not been trained. This stage solely ensures that the velocities along the interpolant paths match the optimal conditional drifts.
> 3. **Stage 3** trains the **growth networks** using the weighted combination of energy, mass, and match losses in Eq. (17), while keeping the velocity networks fixed. This stage learns **when** mass should flow into each branch and **in what amount**, enforcing mass conservation and matching the desired multi-modal terminal distribution.
>
> Notably, **Proposition 3 (App. C.3) and Proposition 4 (App. C.4)** show that in the case of perfect training, Stages 1-3 are sufficient to recover the optimal interpolant, velocity fields, and growth dynamics for the BranchGSB problem. Stage 4 is introduced **only to jointly fine-tune** velocity and growth networks under realistic, imperfect training.
>
> Because the three stages optimize disjoint sets of parameters and serve **complementary roles**, conventional ablations (such as removing Stage 1 or Stage 2) would invalidate the theoretical guarantees and break the decomposition of BranchSBM. To evaluate a meaningful ablation, we include **Stage 3 vs. Stage 4 loss trends** (Table B below and Table 7 in the manuscript). As shown, the joint fine-tuning in Stage 4 consistently improves energy and match losses, confirming that even though Stage 4 is not required by the theory, it improves performance in practice.
>
> **Table B: Validation Losses for Stage 3 and 4 Training Across Experiments. Losses are summed across both branches and averaged over batch size.**
>
> | Experiment | Stage 3 | Stage 3 | Stage 3 | Stage 4 | Stage 4 | Stage 4 |
> | --- | --- | --- | --- | --- | --- | --- |
> |  | Energy | Mass | Match | Energy | Mass | Match |
> | **LiDAR** | 1.276 | 3.0e-5 | 0.007 | 0.768 | 2.0e-5 | 0.102 |
> | **Mouse Hematopoiesis** | 2.209 | 1.2e-4 | 0.054 | 1.918 | 5.0e-5 | 0.076 |
> | **Chlonidine Perturbation**  | 36.469 | 0.030 | 0.109 | 25.798 | 0.053 | 0.153 |
> | **Trametinib Perturbation**  | 35.834 | 0.023 | 0.078 | 32.843 | 0.017 | 0.056 |

---

> ### Author Response · Authors · 2025-12-03
> **Author Response To Reviewer (3/4)**
>
> **W3: This substantial methodological complexity is not yet fully justified by the provided experimental validation. The cost of the method would be more compelling if benchmarked against a wider array of competitive alternatives and shown to have outstanding results.**
>
> To address your concern, we conduct an **additional experiment with comparisons against SOTA baslines [1, 2]** using a dataset consisting of pancreatic $\beta$-cell differentiation data [3] containing 51,274 cells collected over eight time points as they evolve from human pluripotent stem cells to pancreatic  $\beta$-like cells. This data contains eight time points $t_i$ for $i\in \\{0, \dots, 7\\}$ that are projected to 30-dimensional PC representations $\mathbf{x}\in \mathbb{R}^{30}$. We use Leiden clustering to define $K=11$ distinct target distributions of samples at time $t_7$ and set their target weights relative to the total weight at $t_0$ (App E.4). We used samples across all time steps to define the data manifold via the RBF metric $V^{\text{RBF}}\_{t, \eta}$. BranchSBM was trained on pairs sampled only from $t_0$ and $t_7$, and the distributions at all intermediate time points were inferred from learning to minimize the distance from the data manifold over time. For baselines, we compared two SOTA methods for single-cell trajectory modeling: **DeepRUOT [1]** and **CytoBridge [2]**, which are trained explicitly on intermediate snapshots to reconstruct the stochastic dynamics of cells that evolve independently.
>
> Notably, **BranchSBM not only reconstructs the multi-modal terminal distribution at $t_7$ with superior accuracy against all baselines, but also produces intermediate trajectories that are competitive with models trained directly on intermediate snapshots using explicit reconstruction losses** (Table C below and Table 3 in the revised manuscript). Leveraging the RBF state cost $V^{\text{RBF}}_{t, \eta}$, which encourages trajectories to remain on the underlying data manifold, BranchSBM effectively captures the true differentiation dynamics through the combined influence of the neural interpolant and the path-energy objective. These results demonstrate that BranchSBM scales reliably to a large number of branches and is particularly advantageous in settings where intermediate timepoints are sparse or unavailable, allowing the model to infer biologically meaningful trajectories even with limited data on intermediate time points.
>
> **Table C: Results for pancreatic $\beta$-cell differentiation experiment.** 1-Wasserstein distance ($\uparrow$) of intermediate distributions generated by DeepRUOT [1], CytoBridge [2], and BranchSBM (Ours) at eight different time points. † denotes values taken from [2]. Best values are **bolded**.
>
> | Model | $t=1$ | $t=2$ | $t=3$ | $t=4$ | $t=5$ | $t=6$ | $t=7$ |
> | --- | --- | --- | --- | --- | --- | --- | --- |
> | **DeepRUOT** † | **8.0447 ± 0.0005** | 8.0773 ± 0.0021 | 7.6301 ± 0.0032 | **8.0064 ± 0.0042** | **7.9018 ± 0.0117** | 8.3977 ± 0.0102 | 7.8346 ± 0.0109 |
> | **CytoBridge** † | 8.0448 ± 0.0005 | 8.0771 ± 0.0021 | **7.6299 ± 0.0032** | 8.0066 ± 0.0043 | **7.9018 ± 0.0117** | **8.3974 ± 0.0102** | 7.8343 ± 0.0109 |
> | **BranchSBM (Ours)** | 11.9774 ± 0.0000 | **7.4643 ± 2.4031** | 11.5204 ± 0.0000 | 11.2593 ± 0.0000 | 10.2888 ± 0.0000 | 8.7301 ± 0.0000 | **6.8702 ± 0.0000** |

---

> > ### Author Response · Authors · 2025-12-03
> > **Author Response To Reviewer (4/4)**
> >
> > ### **Addressing Questions**
> >
> > **Q1: The method's parameters and computation scale linearly with the number of branches, $K$. You've shown this for $K=2$ and $K=3$. Have you tested the method's stability and performance for a larger $K$, such as $K=10$ or $K=20$, which would be more representative of complex biological differentiation?**
> >
> > First, we note that the number of branches $K$ is limited by the number of cells contained in the dataset to ensure there are enough data points for training. As shown in Table 8, the smallest cluster for the Clonidine experiment contains 1033 cells, and the smallest cluster for the Trametinib experiment contains 381 cells. When running Leiden clustering, we set the minimum number of cells in each terminal cluster to be 200 and all clusters that are smaller than the minimum are merged with their nearest neighbour cluster.
> >
> > **To demonstrate scalability to more branches given more cells, we have added an additional experiment with a** $K=11$ **branches in Table 3 of our revised manuscript,** which simulates pancreatic $\beta$-cell differentiation **(Table C above)**. In this experiment, BranchSBM show superior performance in reconstructing the multi-modal endpoints and can accurately reconstruct intermediate states **without** using explicit samples from the intermediate time-points during training.
> >
> > ---
> >
> > **Citations:**
> >
> > [1] Zhang, Zhenyi, Tiejun Li, and Peijie Zhou. "Learning stochastic dynamics from snapshots through regularized unbalanced optimal transport." (2024).
> >
> > [2] Zhang, Zhenyi, et al. "Modeling Cell Dynamics and Interactions with Unbalanced Mean Field Schrödinger Bridge." (2025).
> >
> > [3] Veres, Adrian, et al. "Charting cellular identity during human in vitro β-cell differentiation." *Nature* 569.7756 (2019): 368-373.
> >
> > ---
> >
> > We sincerely appreciate your deep engagement in reviewing our paper and your valuable feedback. Your suggestions have helped improve our paper in many aspects, and we hope that our detailed responses have addressed each of your concerns.
> >
> > Thank you,
> >
> > The Authors

---

### Official Review · Reviewer_3tYX · 2025-11-01

**Soundness:** 4
**Presentation:** 3
**Contribution:** 4
**Rating:** 6
**Confidence:** 2

**Summary:**

The paper introduces Branched Schrödinger Bridge Matching (BranchSBM) to overcome the limitations of existing approaches, such as flow matching and Schrödinger Bridge Matching, which can only model a single unimodal path. BranchSBM learns multiple time-dependent velocity fields and growth processes, enabling it to capture branched or divergent evolution from a common origin to multiple distinct outcomes.

**Strengths:**

* This method is grounded in solid mathematical theory and is particularly useful for challenging tasks, such as multi-path surface navigation.
* The paper adopts an approach similar to OT-CFM, where optimal transport is used to define couplings that improve flow-matching vector fields. It formulates the **Branched Generalized Schrödinger Bridge Problem** and develops a theoretical foundation around it, enabling the computation of couplings for downstream CFM models to learn branched paths effectively. This is a common method, yet its deduction is non-trivial in this case.

**Weaknesses:**

There is room to improve the clarity of the technical sections by providing more detailed explanations. For example, could the authors clarify why it is necessary to consider the time-dependent weights and growth rates in Equation (5)? What is the motivation or scenario that calls for this addition, and how does it connect to the problem studied in the paper? Explaining why this is the appropriate formulation would help readers better understand both the method and the rationale behind it.

**Questions:**

Could this method be applied to tasks such as text-to-image generation or other common multimodality problems? If so, is there a reason why the paper does not explore these applications?

Please also refer to the weaknesses section for additional context.

---

> ### Author Response · Authors · 2025-12-03
> **Author Response To Reviewer**
>
> **Dear Reviewer 3tYX,**
>
> Thank you for taking the time to carefully review our paper, and we are happy to hear that you found our method theoretically grounded and useful for solving challenging tasks. Please see our revised manuscript to view the **key changes** that we have made to improve the manuscript, colored in $\textcolor{blue}{\text{blue}}$. **Here, we highlight the changes that address each of your concerns one by one:**
>
> ---
>
> **W1: Could the authors clarify why it is necessary to consider the time-dependent weights and growth rates in Equation (5)? What is the motivation or scenario that calls for this addition, and how does it connect to the problem studied in the paper? Explaining why this is the appropriate formulation would help readers better understand both the method and the rationale behind it.**
>
> Thank you for highlighting this important piece of novelty in our theoretical framework. The time-dependent weights and growth rates in Eq. 5 enable BranchSBM to model **branching dynamics at the level of the population distribution**, instead of at the level of independent samples like in prior methods. In various population systems, like single-cell populations, mass is dynamically split into multiple sub-populations that undergo diverging trajectories to different terminal modes in the target distribution.
>
> This is a unique feature of **our definition of the BranchSB problem in Sec 3.2**, which treats branching as a **unified stochastic process with a single initial distribution**, whose mass is dynamically split across multiple branch-specific flows through the learned growth terms $g_{t, k}$. The introduction of the dynamic weight $w_{t, k}$ in the energy loss (Eq. 14) scales the energy and state cost by the **fraction of the population following each branch at each time**, enabling the model to learn **when** the population diverges and **how much mass** flows into each terminal mode while minimizing energy and reconstructing the multi-modal terminal distribution. This differs fundamentally from treating branching as several independent samples evolving separately, which cannot capture population-level mass redistribution. **We have clarified this motivation at the beginning of Sec 3 and in App B.**
>
> ---
>
> **Q1: Could this method be applied to tasks such as text-to-image generation or other common multimodality problems? If so, is there a reason why the paper does not explore these applications?**
>
> Our framework is **uniquely** positioned for modelling single-cell trajectories and perturbation effects on cells and overcomes the limitations in prior methods, like single-branch SBM as described in **App D.1**. While it is possible to apply our general branching framework to text-to-image generation or other multimodality problems where we aim to generate an ensemble of terminal states that are weighted by their proportion or probability, these applications require **fundamentally different architectures** (e.g. diffusion, autoregressive models). Therefore, exploring these applications would require extensive adaptation of our core method and is beyond the scope of our current study.
>
> Single-cell transcriptomic data provide a suitable data modality to validate BranchSBM because cells naturally undergo **stochastic branching processes** that our framework is designed to capture. Each cell population originates from a common progenitor distribution and evolves into multiple heterogeneous terminal states through differentiation or perturbation, which is precisely the setting of a branched Schrödinger bridge, where **mass diverges along energy-minimizing paths**. These datasets are inherently **high-dimensional, noisy, and multimodal**, making them a very stringent test of any method that learns population-level stochastic transport.
>
> Also, unlike synthetic or low-dimensional systems, single-cell data offer **experimentally validated** examples of **probabilistic mass redistribution** under biological constraints, which allows us to directly evaluate how well a model captures divergence, imbalance, and non-conservative dynamics. Given these reasons, single-cell trajectory modelling serves as both the most challenging and the most relevant benchmark for validating BranchSBM. To support this claim, we also add an additional experiment in Sec 5.2 and Table 3 comparing BranchSBM to prior state-of-the-art baselines [1, 2] for modelling pancreatic $\beta$-cell differentiation dynamics.
>
> ---
>
> **Citations:**
>
> [1] Zhang, Zhenyi, Tiejun Li, and Peijie Zhou. "Learning stochastic dynamics from snapshots through regularized unbalanced optimal transport." (2024).
>
> [2] Zhang, Zhenyi, et al. "Modeling Cell Dynamics and Interactions with Unbalanced Mean Field Schrödinger Bridge." (2025).
>
> ---
>
> We would like to sincerely thank you again for your positive evaluation of our paper, and we hope that our detailed responses have addressed each of your remaining concerns.
>
> Thank you,
>
> The Authors

---

### Author Response · Authors · 2025-12-03
**Summary of Reviewer Concerns and Revisions for Reassigned AC (1/4)**

**Dear Area Chair,**

First, we sincerely want to thank the reviewers for their careful evaluation of our submission. We are happy to see that the reviewers found that our paper presents a **relevant contribution for handling branched paths (Reviewer 7Why)** and a **sound theoretical framework (Reviewers 3tYX, 7Why, 2wLh)** that **solves a “particularly challenging problem” (Reviewer 3tYX) with “consistent improvements” in biological tasks (Reviewer 2wLh, 1Hyc).** We carefully interpreted the **key concerns** raised by each reviewer and substantially strengthened the paper with **new experiments, computational cost analysis, and clearer justifications of our methodological contribution** (notable changes are highlighted in $\textcolor{blue}{\text{blue}}$ in the revised manuscript).

Given the AC reassignment, we would like to provide our new AC with a **summary of the common reviewer concerns, our responses, and the improvements we have made to our manuscript since the original submission**.

---
### **Summary of Responses to Common Concerns**

**Concern 1: How is the definition of the BranchSB problem different from the multi-marginal Schrödinger bridge problem? (Reviewer 2wLh)**

- While the **branched Schrödinger bridge problem may seem similar to the multi-marginal Schrödinger bridge problem**, they solve **fundamentally disjoint problems**. The multi-marginal Schrödinger bridge problem aims to generate a stochastic trajectory that interpolates between **multiple intermediate marginal distributions.**
- In contrast, our definition of the **branched Schrödinger bridge problem (BranchSB)** aims to learn the **branching** trajectories that interpolate from an initial distribution which is unimodal to a **multi-modal distribution using $K$ optimal velocity fields and growth fields.**
- To our knowledge, our formulation of BranchSB is a novel contribution and is motivated by the following **key ideas:**
    - Single-path SBM frameworks require simulating a large batch of particles to recover a multi-modal terminal distribution.
    - Branching dynamics is governed by energy-minimizing trajectories. Multi-marginal SB assumes Brownian interpolants between intermediate snapshots which fails to capture continuous non-linear dynamics that follow an underlying energy function.
    - Simulating a branched path distribution containing multi-modal marginals with independent particle simulations can suffer from mode collapse, where particles are biased towards a high-density mode in the distribution or only traverse low-energy intermediate paths.
    - Rather than fixing intermediate marginals that must be met by the interpolant that is only defined at discrete increments along the path, BranchSB aims to minimize the potential energy function, which is defined over the full interpolant.

    To better motivate the BranchSB problem, we have discuss **the above points at the beginning of Sec 3** of the revised manuscript.


---

**Concern 2: Why is BranchSBM fundamentally better than single-branch SBM? (Reviewer 2wLh)**

- **Single-branch SBM cannot model mass splitting.** Its objective drives all simulated particles toward a single low-energy mode, causing **mode collapse** when the target distribution is multi-modal or imbalanced.
- **BranchSBM introduces branch-specific drifts and a learned growth process** that dynamically reallocates population mass across branches over time.
    - This coupling is **theoretically guaranteed** (Proposition 4, App. C.4) to converge to the optimal mass redistribution and branch-specific velocity fields.
- **BranchSBM models population-level branching**, not independent particle trajectories, enabling it to learn **when** mass splits and **how much mass** flows into each terminal mode, which single-branch SBM cannot express.
- **Empirically**, single-branch SBM collapses to only the dominant mode in high-dimensional perturbation datasets (App. D.1), failing to recover structure along higher PCs.

---

> ### Author Response · Authors · 2025-12-03
> **Summary of Reviewer Concerns and Revisions for Reassigned AC (2/4)**
>
> **Concern 3: How does BranchSBM compare against competitive baselines? (Reviewers 7Why, 2wLh, 1Hyc)**
>
> To address this concern, we conduct an **additional experiment with comparisons against SOTA baselines [1, 2]** using a dataset consisting of pancreatic $\beta$-cell differentiation data containing ~51K cells collected over **eight time points** and report results in **Table 3 of the revised manuscript.**
>
> - BranchSBM was trained on pairs sampled only from $t_0$ and $t_7$, and the distributions at all intermediate time points were inferred from learning to minimize the distance from the data manifold over time. For baselines, we compared two SOTA methods for single-cell trajectory modeling: **DeepRUOT [1]** and **CytoBridge [2]**, which are trained explicitly on intermediate snapshots to reconstruct the stochastic dynamics of cells that evolve independently.
> - Notably, even though BranchSBM is trained **only on**  $t_0$ and $t_7$, it **produces intermediate distributions that match or exceed baselines** that have full access to intermediate snapshots and explicit reconstruction losses **(Table 3)**.
> - Leveraging the RBF state cost, which encourages trajectories to remain on the underlying data manifold, BranchSBM effectively captures the true differentiation dynamics through the combined influence of the neural interpolant and the path-energy objective.
> - These results show that BranchSBM scales reliably to a large number of branches and is particularly advantageous in settings where intermediate timepoints are sparse or unavailable.
>
> ---
>
> **Concern 4: What is the training time for increasing dimension and branches? (Reviewers 2wLh, 1Hyc)**
>
> We have added **Table 10 in the revised manuscript**, which reports the GPU hours on a single NVIDIA A100 GPU for each experiment. Notably, we find that there is **only a minor increase in compute time required for increasing the number of branches $K$ and training completes under 30 minutes across all experiments.**
>
> ---
>
> **Concern 5: Given the method’s complexity, how does it scale and generalize to new tasks? (Reviewers 7Why, 1Hyc)**
>
> - While the method appears complex, the implementation and training process is run automatically with sequential training of the interpolants, flow networks, and growth networks without any external intervention.
> - We have added the training times for each experiment in **Table 10 in the revised manuscript**, which shows that only a minor increase in compute time is required for increasing the number of branches $K$.
> - To address generalizability to a larger number of branches and tasks with sparse data, we include an additional experiment with $K=11$ for 30-dimensional pancreatic $\beta$-cell differentiation data **(Sec 5.2; Table 3)**, demonstrating that:
>     - BranchSBM **scales reliably to a large number of branches**.
>     - BranchSBM is **generalizable** to tasks where snapshots at specific intermediate timepoints are not known.
>
> ---
>
> **Concern 6: Why is single-cell data sufficient verification for the proposed framework? (Reviewers 3tYX, 2wLh)**
>
> - **Single-cell systems naturally exhibit stochastic branching**, where a common progenitor population diverges into multiple heterogeneous terminal states, which is precisely the setting modeled by our branched Schrödinger bridge problem (Sec 3).
> - These datasets are **high-dimensional, noisy, and inherently multimodal**, providing a rigorous test of population-level stochastic transport.
> - Unlike synthetic or low-dimensional benchmarks, single-cell data provide **experimentally validated population dynamics**, enabling direct evaluation of mass redistribution, divergence, and non-conservative dynamics.
> - BranchSBM is specifically designed to model **probabilistic mass flow across multiple branches**, which aligns with biological differentiation and perturbation processes.
> - We strengthened the empirical validation by adding a **new experiment (Sec. 5.2)** on pancreatic $\beta$-cell differentiation with $K=11$ **branches**, comparing against **state-of-the-art baselines** (DeepRUOT [1], CytoBridge [2]).
>     - Notably, BranchSBM **outperforms all baselines on terminal reconstruction** and achieves **competitive or superior accuracy** on intermediate trajectories despite not being trained on intermediate snapshots.

---

> > ### Author Response · Authors · 2025-12-03
> > **Summary of Reviewer Concerns and Revisions for Reassigned AC (3/4)**
> >
> > **Concern 7: What is the significance of each training stage, and can you show ablations on performance removing certain stages? (Reviewers 7Why, 1Hyc)**
> >
> > - We emphasize that Stage 1, Stage 2, and Stage 3 train completely different parameterized models that serve **complementary purposes in the BranchSBM framework**. Each stage isolates one component of the BranchSBM objective:
> >     - **Stage 1** trains only the **neural interpolant** via the trajectory loss (Eq. 12) to learn minimum-energy corrections between endpoints; distributional metrics such as W1/MMD are not applicable because endpoints are fixed.
> >     - **Stage 2** trains only the **branch-specific velocity networks** via the flow loss (Eq. 13) to match conditional drifts along interpolant paths; growth is not yet defined.
> >     - **Stage 3** trains only the **growth networks** with the weighted energy/mass/match losses (Eq. 17), enforcing mass conservation and matching branch weights while keeping velocities fixed.
> >     - **Stage 4** is not required by the theory, it improves performance in practice by jointly training all velocity and growth networks to minimize the energy loss.
> > - **Proposition 3 (App. C.3) and Proposition 4 (App. C.4)** show that in the case of perfect training, Stages 1-3 are sufficient to recover the optimal interpolant, velocity fields, and growth dynamics for the BranchGSB problem. Stage 4 is introduced **only to jointly fine-tune** velocity and growth networks under realistic, imperfect training.
> > - Because removing Stage 1 or 2 would break the theoretical decomposition, we instead added an evaluation of **Stage 3 vs. Stage 4 loss trends (Table 7)** showing that the joint fine-tuning in Stage 4 consistently improves energy and match losses.
> >
> > ---
> >
> > ### **Summary of Individual Reviewer Concerns and Responses**
> >
> > - **Reviewer 3tYX [Score 6]** scored our paper as **excellent** for both **soundness and contribution**, highlighting that our “method is grounded in solid mathematical theory and is particularly useful for challenging tasks” with “non-trivial” theoretical contribution of our definition for the Branched Generalized Schrödinger Bridge Problem (Sec 3.2). They raised minor concerns, which we summarize in our responses below:
> >     - To address their confusion, we clarify that the time-dependent weights and growth rates in Eq. 5 enable BranchSBM to model **branching dynamics at the level of the population distribution**, instead of at the level of independent samples like in prior methods.
> >         - The introduction of the dynamic weight in the energy loss (Eq. 14) scales the energy and state cost by the **fraction of the population following each branch at each time**, enabling the model to learn **when** the population diverges and **how much mass** flows into each terminal mode while minimizing energy and reconstructing the multi-modal terminal distribution
> >         - **We further clarified this motivation at the beginning of Sec 3 and in App B in the manuscript.**
> >     - They also wondered why single-cell data provides sufficient empirical validation of our method **(which we address in Concern 6 above)**
> >     - **Given that we addressed each of the reviewers’ remaining concerns, we believe that the reviewer would have maintained a positive score.**
> > - **Reviewer 7Why [Score 4]** highlighted that our “**core idea […] is a relavant contribution**” and our **theoretical framework is sound**. Their primary concerns were regarding whether the method is scalable and generalizable **(which we address in Concern 5)**, the importance of each training stage **(which we address in Concern 7)**, and the comparison to competitive baselines **(which we address in Concern 3)**.
> >     - To address their additional question on scaling to a larger number of branches, we clarify that the number of branches is limited by the minimum number of cells needed to train each branch. **We also demonstrate that BranchSBM is capable of scaling to $K=11$ branches for the pancreatic $\beta$-cell differentiation data (Sec 5.2; Table 3).**
> >     - **Overall, reviewer 7Why gave us a borderline score, but we have now addressed each of their concerns in detail in our individual response and revised manuscript.**

---

> > > ### Author Response · Authors · 2025-12-03
> > > **Summary of Reviewer Concerns and Revisions for Reassigned AC (4/4)**
> > >
> > > - **Reviewer 2wLh [Score 4]** highlighted that our paper is **straightforward to understand and our method is sound**. Their primary concerns were regarding how the BranchSB problem is distinct from the prior multi-marginal SB problem **(which we address in Concern 1)** and single-branch SBM **(which we address in Concern 2)**, why single-cell data is sufficient validation for BranchSBM **(which we address in Concern 6)**, training time **(which we address in Concern 4).**
> > >     - We clarify that the BranchSB and multi-marginal SB problems solve **disjoint problems** (further explanation given under **Concern 1 above**).
> > >     - To address the unclear technical motivation raised by the reviewer, we have **described the key ideas that motivate the BranchSB problem of Sec 3** of the revised manuscript.
> > >     - **Overall, reviewer 2wLh gave us a borderline score, but we have now addressed each of their concerns in detail in our individual response and revised manuscript.**
> > > - **Reviewer 1Hyc [Score 4]** raised several questions, each of which we have addressed in detail in our response and revised manuscript. Their primary concerns were regarding comparison to unbalanced SB, training complexity **(which we address in Concern 4)**, limited baseline comparisons **(which we address in Concern 3)**, sensitivity to clustering protocol, the importance of each training stage **(which we address in Concern 7)**, and scalability **(which we address in Concern 5).**
> > >     - We emphasize that BranchSBM introduces **three key innovations beyond unbalanced SB/RUOT**:
> > >         - **Multi-modal terminal distributions** with explicit branch structure, avoiding mode collapse inherent to single-terminal unbalanced SB.
> > >         - **A global mass-coupling constraint,** which couples branches and removes the need for growth penalties used in unbalanced SB.
> > >         - **Learnable branching times and mass splits**, enabling inference of multi-modal intermediate marginals **without** intermediate snapshots.
> > >         - Supported by **theoretical guarantees** (Propositions 3 and 4), which establish the optimality of the learned drifts and growth rates.
> > >     - To address their concern about sensitivity to the clustering method, we highlight that **BranchSBM minimizes pipeline bias** by using Leiden clustering, which is fully unsupervised and robust to noise and heterogeneous cluster shapes. Only the minimum cells per cluster needs to be specified, which determines when a small cluster is merged with its nearest cluster.
> > >     - To address their concern regarding sensitivity to state cost, we note that the state cost is defined to naturally align with each task (e.g., altitude in LiDAR; manifold structure in single-cell gene expression). For single-cell data, the data manifold metric is the **most appropriate** and ensures trajectories remain biologically plausible along held-out time points (proven empirically in the $\beta$-cell experiment).
> > >     - We clarify that Wasserstein distance estimation in high dimensions suffers from **curse-of-dimensionality**, making high-D estimates statistically unstable. Evaluating W-distance on top-2 PCs yields a **stable and meaningful** comparison while RBF-MMD is computed on **all PCs** to assess full-dimensional accuracy.
> > >     - **Overall, reviewer 2wLh gave us a borderline score, but we have now addressed each of their concerns in detail in our individual response and revised manuscript.**
> > >
> > > ---
> > >
> > > **Citations:**
> > >
> > > [1] Zhang, Zhenyi, Tiejun Li, and Peijie Zhou. "Learning stochastic dynamics from snapshots through regularized unbalanced optimal transport." (2024).
> > >
> > > [2] Zhang, Zhenyi, et al. "Modeling Cell Dynamics and Interactions with Unbalanced Mean Field Schrödinger Bridge." (2025).
> > >
> > > ---
> > >
> > > We would like to thank our original reviewers again for their time and effort in reviewing our paper. We regret that we can no longer engage in discussion with our reviewers, and would like to assure them that each of their reviews has improved our paper in many aspects.
> > >
> > > Overall, all reviewers gave us borderline scores **(6, 4, 4, 4)**, but we have now **addressed each of their concerns in detail.** We believe these clarifications and new experiments would **likely have resolved their remaining concerns and resulted in an overall positive evaluation** if the review period had not been cut short.
> > >
> > > We kindly ask that our new AC acknowledge the contributions of our paper and our efforts to thoroughly address each of our reviewer comments and support our paper for acceptance to ICLR.
> > >
> > > Thank you,
> > >
> > > The Authors

---

### Meta-Review · Area_Chair_NToJ · 2026-01-04

**Summary:**

This work proposes a distribution transport method based on Schrödinger Bridge that matches one source distribution to multiple branches. This method is motivated by the single-cell evolution process where in different stages branching occurs.

The reviewers acknowledged that the problem is well motivated in single-cell analysis. They also found the theoretical decomposition of this complex problem to stages sound and valid. They also noticed that the method works well on biological data.

The reviewers had concerns regarding the sequential 4-stage training approach proposed in the paper, pointing out that it might be costly and sensitive to parameter tuning.  The reviewers also asked if increasing K (i.e., the number of branches) could slow the runtime. One reviewer raised concerns that the proposed approach might be similar to the multi-marginal transport problem.  A reviewer is additionally critical of the reliance of clustering method, the similar formulation as in (Neklyudov et al., ICML 2024), the lack of baselines, and choice of task specific V_t.

Overall, this work deals with an important problem that often occurs in single-cell evolution analysis. The considered problem is fundamentally different to similar settings in the literature. The proposed 4-stage method may be a bit complex, but the overall learning process is streamlined and the cost is moderate (as addressed in the rebuttal). The concerns on the lack of baselines is addressed in the rebuttal. The concern on only small K was used in simulations was also addressed.

**Reviewer Concerns:**

The authors responded by showing the overall training time being acceptable. In terms of K, the authors added an experiment where K=11 is used.

The authors also pointed out the branching problem is fundamentally different from the multi-marginal problem, which the AC agrees to.

The authors also pointed out that although clustering is used, the pipeline is automated.

The related works using neural interpolants are also added into discussion.

Some more baselines are added.

**Reviewer Scores:**

Reviewer 3tYX (rating 6): main concerns were on clarity. These were discussed in the rebuttal.

Reviewer 7Why (rating 4): had concerns on the complexity of the 4-stage approach, the complexity of hyperparameter tuning, limited baselines, and small K used in experiments. The rebuttal explained that the 4-stage approach is streamlined and not costly to run with a runtime table, that the parameters are separately tuned for each stage and thus tuning is manageable, that two baselines are added, and that a K=11 experiment is added. The AC finds that these answers could have resolved the concerns and lead to an increased score.

Reviewer 2wLh (rating 4): asked what is the difference between multimarginal bridges and is a single cell experiment sufficient? The reviewer also mentioned that no theory to support fundamentally better than others. The authors responded by explaining the fundamental difference from multimarginal bridges and asserted that this work is uniquely positioned for single-cell experiments. More experiments are added. Some theoretical advantages (e.g., proposition 4) were discussed. These may lead to an increase of score.

Reviewer 1Hyc (rating 4) has shared concerns of other reviewers, e.g., on complexity, limited experiments, scalability. They also mentioned missed references. But more experiments were added. Maintaining or increasing the score is expected.

---

### Decision · Program_Chairs · 2026-01-26

Accept (Poster)